# CONSTRAINED DIFFUSION POLICY OPTIMIZATION FOR OFFLINE REINFORCEMENT LEARNING

## ABSTRACT

In this paper, we propose the two-fold improved diffusion policy (TDP) for offline reinforcement learning. We first propose the constrained diffusion policy optimization (CDPO) framework, which unifies existing diffusion-based policy constraint methods. TDP harnesses the full potential of CDPO by initializing with the closed-form solution of a constrained optimization problem and then applying another constrained policy optimization for further refinement. We establish the theoretical properties of TDP, including expected policy improvement, in-distribution property, and approximate gains over existing diffusion policies. We also propose a design method for estimating the desired policy in the TDP loss function to achieve the aforementioned performance improvements. Empirical results on the D4RL benchmark show that TDP outperforms most existing offline reinforcement learning methods.

## 1 INTRODUCTION

Offline reinforcement learning (RL) has gained attention for training agents using precollected datasets, avoiding costly and risky online interactions common in real-world applications like autonomous vehicles and robotics. One of the crucial challenges in offline RL is overestimation in out-of-distribution (OOD) regions due to the limited coverage of the static dataset and the lack of function approximation error correction through online interactions.

Policy constraint methods (Levine et al., 2020) are widely used in offline RL to mitigate overestimation in OOD regions by keeping the learned policy close to the behavior policy, namely constrained policy optimization (CPO) (Achiam et al., 2017). Explicit methods (Wu et al., 2019; Fujimoto & Gu, 2021) combine a Q loss term to encourage high-value actions, with a divergence penalty from the behavior policy, while implicit methods (Nair et al., 2020; Kostrikov et al., 2022) directly regress toward a closed-form optimal policy, with the reverse KL constraint (Peters et al., 2010). Despite their popularity, these methods face key limitations: explicit methods often fail to reliably prevent OOD actions due to indirect guidance, especially on suboptimal datasets, while implicit methods suffer from limited functional diversity and may generate suboptimal actions. Furthermore, the expressiveness of the policy model is crucial to realizing the benefits of these objectives fully.

Recently, diffusion policies (Wang et al., 2023; Hansen-Estruch et al., 2023; Fang et al., 2024) have emerged as highly expressive policies, using diffusion models (Sohl-Dickstein et al., 2015; Ho et al., 2020; Song et al., 2021) to capture complex multimodal structures in data distributions and environment dynamics. Among them, DQL (Wang et al., 2023) is an explicit policy constraint method that balances a Q loss with a diffusion loss, where the latter serves as a divergence-like penalty. However, unlike standard divergence-based regularizers, the diffusion loss lacks rigorous theoretical grounding, and diffusion models trade some stability for expressiveness. Consequently, while DQL achieves strong performance, it also shows training instability in some tasks with a limited theoretical foundation.

In this paper, we introduce *constrained diffusion policy optimization* (CDPO), a framework that generalizes CPO, with high-expressiveness diffusion policies. CDPO relaxes the limitation that the learned policy must remain close to the behavior policy by interpreting implicit methods as mimicking an improved, yet still in-distribution, *anchor policy* without relying on the Q loss. Prior approaches within this framework have not fully exploited its design flexibility, leading to inevitable trade-offs. To address this, we propose the *two-fold improved diffusion policy* (TDP), which fully utilizes the

CDPO design variables. TDP begins with an enhanced version of the behavior policy from implicit methods, and then applies generalized CPO by utilizing *noise-free estimation* for further refinement.

Theoretically, we first show that the diffusion loss corresponds to a divergence from an optimization perspective and further establish a stronger result, the *policy enhancement theorem*. As a corollary, we prove that TDP theoretically outperforms several CDPO policies under mild conditions. We also reformulate the notion of in-distribution policies and demonstrate that CDPO (and, of course, TDP) satisfies this property. To bridge these theoretical insights with practice, we develop a practical TDP algorithm. Empirically, we demonstrate that TDP achieves state-of-the-art performance across diverse tasks in the D4RL benchmark (Fu et al., 2020), and validate its effectiveness through 2D bandit visualizations and ablation studies, including a flow policy variant (Lipman et al., 2023; Park et al., 2025) that highlights the role of policy expressiveness.

Our contributions are summarized as follows:

- We introduce TDP, a simple yet effective offline RL algorithm within the CDPO framework, and its practical implementation to achieve consistent performance improvements.
- We prove several theoretical properties of CDPO and TDP, including policy enhancement, in-distribution behavior, and approximate gains over existing diffusion policies.
- We provide numerical results, visualizations, and ablation studies demonstrating the remarkable performance improvements and effectiveness of the proposed TDP algorithm.

## 2 PRELIMINARY

**Offline RL**    RL is typically modeled as a Markov decision process (MDP) $\mathcal{M} = \langle \mathcal{S}, \mathcal{A}, R, P, d_0, \gamma \rangle$, where $\mathcal{S}, \mathcal{A}$ are the state and action spaces, $R$ the reward, $P$ the dynamics, $d_0$ the initial distribution, and $\gamma$ the discount factor. The goal is to learn a policy $\pi$ that maximizes $J(\pi) = \mathbb{E}_{s_0 \sim d_0, \pi} [\sum_{t=0}^{\infty} \gamma^t R(s_t, a_t)]$. In offline RL, only a fixed dataset $\mathcal{D}$ from a behavior policy $\pi_\beta$ is available, so $(s, a) \sim \mathcal{D}$ means $s \sim \rho^{\pi_\beta}, a \sim \pi_\beta(\cdot|s)$, where $\rho^{\pi_\beta}$ is the state visitation distribution.

**Policy Constraint Methods**    Policy constraint methods (Levine et al., 2020) are the value-based offline RL methods using the following constrained policy optimization (CPO) (Achiam et al., 2017) for actor optimization:

$$\pi^{(t+1)} = \arg\max_{\pi} L_{Q^{\pi^{(t)}}}(\pi) \quad \text{s.t.} \quad \mathbb{E}_{s \sim \rho^{\pi_\beta}} [D(\pi_\beta(\cdot|s) \parallel \pi(\cdot|s))] \leq \varepsilon_b, \quad (1)$$

where $D$ is a divergence and $L_{Q^{\pi^{(t)}}}(\pi) := \mathbb{E}_{s \sim \rho^{\pi_\beta}, a \sim \pi(\cdot|s)}[Q^{\pi^{(t)}}(s, a)]$ is the expected Q value under $\pi$, namely Q loss. This optimizes the policy to maximize the expected Q value, namely Q loss, while remaining close to the behavior policy $\pi_\beta$. If $D$ is an $f$-divergence (Amari, 2016), the problem reduces to the dual form

$$\pi^{(t+1)} = \arg\min_{\pi} \{\mathbb{E}_{s \sim \rho^{\pi_\beta}} [D(\pi_0(\cdot|s) \parallel \pi(\cdot|s))] - \zeta_t L_{Q^{\pi^{(t)}}}(\pi)\}, \quad (2)$$

where $\zeta_t$ matches the divergence constraint $\varepsilon_b$ in the primal form. To implement this update in deep RL, it is customary to convert (2) into the following *loss function form*:

$$L(\theta) := \mathbb{E}_{s \sim \mathcal{D}} [D(\pi_\beta(\cdot|s) \parallel \pi_\theta(\cdot|s))] - \zeta_t L_{Q_\phi}(\theta), \quad (3)$$

where $L_{Q_\phi}(\theta) := \mathbb{E}_{s \sim \mathcal{D}, a \sim \pi_\theta(\cdot|s)} [Q_\phi(s, a)]$, which we also call as Q loss. Explicit policy constraint methods (Wu et al., 2019; Fujimoto & Gu, 2021) directly use (3) with a relaxation by $\zeta_t \equiv \zeta$, while implicit methods (Nair et al., 2020; Fang et al., 2024) regress to $\pi_\eta^*(a|s) = \frac{1}{Z(s)}\pi_\beta(a|s) \exp(\frac{1}{\eta}Q_\phi(s, a))$, a closed-form solution of (3) when $D$ is the reverse KL divergence with $\zeta_t \leftarrow \frac{1}{\eta}$ (Peters et al., 2010).

Although policy constraint methods are originally defined with temporal difference (TD) learning for the critic, in this paper, we use the term more broadly, without restricting the critic learning algorithm.

**Diffusion Policies**    Diffusion policies (Wang et al., 2023; Hansen-Estruch et al., 2023; Fang et al., 2024) are value-based offline RL methods, whose policies are represented by a diffusion model

(Sohl-Dickstein et al., 2015; Ho et al., 2020; Song et al., 2021), defined as

$$\pi_\theta(a^{0:K}|s) := \underbrace{\mathcal{N}(0, I)}_{\sim a^K} \prod_{k=1}^{K} \pi_\theta(a^{k-1}|a^k, s), \tag{4}$$

where $K$ is the number of diffusion steps and the action $a$ is taken as the final output of the denoising process, i.e., $a = a^0$. The reverse sampling distribution $\pi_\theta(a^{k-1}|a^k, s)$ for denoising is realized as $a^{k-1} = \frac{1}{\sqrt{\alpha^k}}(a^k - \frac{1-\alpha^k}{\sqrt{1-\bar\alpha^k}}\epsilon_\theta(a^k, k, s)) + \mathbf{1}_{k>1}\sigma^k z$, where $z \sim \mathcal{N}(0, I)$ and $\epsilon_\theta(a^k, k, s)$ is the noise model. Thus, learning $\pi_\theta$ reduces to learning $\epsilon_\theta$, trained with DDPM diffusion loss (Ho et al., 2020)

$$L_{\text{diff}}(\theta) := \mathbb{E}_{(s,a^0)\sim\mathcal{D}, k\sim U[1,K]}[||\epsilon_\theta(a^k, k, s) - \epsilon||^2], \tag{5}$$

where $\epsilon \sim \mathcal{N}(0, I)$ is the true noise generating $a^k = \sqrt{\bar\alpha^k}a^0 + \sqrt{1-\bar\alpha^k}\epsilon$ and $U[1, K]$ is a uniform distribution on $\{1, \cdots, K\}$. DQL (Wang et al., 2023) mimics an explicit policy constraint method, using TD learning for the actor and treating the diffusion loss as if it were an $f$-divergence, yielding

$$L(\theta) := L_{\text{diff}}(\theta) - \zeta L_{Q_\phi}(\theta), \tag{6}$$

with Q loss actions taken from fully denoised samples. See Appendix A for more about diffusion models and DDPM.

## 3 METHOD

In this section, we introduce constrained diffusion policy optimization (CDPO) and propose two-fold improved diffusion policy (TDP), followed by 2D bandit experiments and practicalization methods.

### 3.1 CONSTRAINED DIFFUSION POLICY OPTIMIZATION (CDPO)

Policy constraint methods, introduced in Sec. 2, use constrained policy optimization (CPO) (Achiam et al., 2017) (1) for actor learning. Explicit methods apply the dual form (3), while implicit methods regress toward the closed-form solution $\pi_\eta^*$. Both are commonly viewed as different realizations of the same principle, namely maximizing the Q loss while keeping $\pi_\theta$ close to the behavior policy $\pi_\beta$.

From another perspective, implicit methods can be viewed as mimicking $\pi_\eta^*$ without the Q loss. Thus, explicit and implicit methods span two complementary axes: explicit methods emphasize Q loss-driven optimization, while implicit methods emphasize flexibility through the choice of an *anchor policy*, i.e., the policy to remain close to. This motivates a more general formulation in which the policy is not restricted to $\pi_\beta$. We therefore define *generalized CPO*, which replaces $\pi_\beta$ with a flexible anchor policy $\pi_0$. Generalized CPO admits a dual form

$$\pi^{(t+1)} = \arg\min_\pi \{\mathbb{E}_{s\sim\rho^{\pi_\beta}}\left[D(\pi_0(\cdot|s) \| \pi(\cdot|s))\right] - \zeta_t L_{Q^{\pi^{(t)}}}(\pi)\}, \tag{7}$$

where $L_{Q^{\pi^{(t)}}}(\pi) := \mathbb{E}_{s\sim\rho^{\pi_\beta}, a\sim\pi(\cdot|s)}[Q^{\pi^{(t)}}(s, a)]$, $\pi^{(0)} = \pi_0$, and $\zeta_t$ matches the divergence constraint $\varepsilon_b$ in the primal form.

Since the full applicability of the above loss requires policies with sufficient expressiveness, we instantiate this framework with diffusion policies. Unlike unimodal policies, diffusion policies are inherently multimodal and can therefore capture a much broader range of distributions. We refer to the resulting framework as *constrained diffusion policy optimization* (CDPO). CDPO unifies and extends existing diffusion policy methods within a single framework. In particular, most explicit policy constraint methods, such as DQL (Wang et al., 2023) and its variants (He et al., 2023; Kang et al., 2023), correspond to the case $\zeta_t \equiv \zeta$ with $\pi_0 = \pi_\beta$, while most implicit policy constraint methods, such as DAC (Fang et al., 2024), correspond to $\zeta_t \equiv 0$ with $\pi_0 = \pi_\eta^*$. We also note that simple BC training, as in IDQL (Hansen-Estruch et al., 2023), falls under the case $\zeta_t \equiv 0$ with $\pi_0 = \pi_\beta$. Beyond these restricted cases, CDPO has full flexibility in the choice of $\zeta_t$ and $\pi_0$.

### 3.2 TWO-FOLD IMPROVED DIFFUSION POLICY (TDP)

Existing policy constraint methods face drawbacks. Explicit methods, such as DQL, risk out-of-distribution (OOD) actions since the constraint only indirectly regularizes the policy. This becomes

especially pronounced on suboptimal datasets, where the behavior policy and the desired policy differ substantially, making indirect regularization prone to generating an OOD policy. Implicit methods, such as DAC, instead suffer from inherently low functional diversity: the policy is restricted to $\pi_\eta^*$, a reweighted version of $\pi_\beta$, which narrows the policy class and limits suppression of suboptimal regions. These limitations are intrinsic to their formulations, motivating the need for a new direction.

To improve upon both classes of methods, we propose to exploit both axes of CDPO by setting $\pi_0 \neq \pi_\beta$ and $\zeta_t \not\equiv 0$. In particular, choosing $\pi_0 = \pi_\eta^*$ provides a strong starting point while preserving in-distribution properties. This motivates our *two-fold improved diffusion policy* (TDP), consisting of: (i) starting with $\pi_\eta^*$, which is stronger than $\pi_\beta$ yet remains in-distribution, to overcome the limit of explicit methods, and (ii) optimizing with (7) using $\zeta_t \not\equiv 0$, enabling further gains through the Q loss beyond the solution obtained by implcit methods. In this way, TDP mitigates the drawbacks of implicit methods by exploring from $\pi_\eta^*$ toward higher-Q policies, and addresses the drawbacks of explicit methods by ensuring the Q loss refines a high-quality policy rather than potentially pushing it OOD. Thus, TDP achieves stronger performance while reliably maintaining in-distribution behavior, especially in challenging tasks, as visualized in Sec. 3.3 and as shown in the main experiments.

The CDPO dual form of TDP directly follows as

$$\pi^{(t+1)} = \arg\min_\pi \{\mathbb{E}_{s\sim\rho^{\pi_\beta}} \left[ D(\pi_\eta^*(\cdot|s) \parallel \pi(\cdot|s)) \right] - \zeta_t L_{Q^{\pi^{(t)}}}(\pi)\}. \tag{8}$$

For practical implementation, we convert (8) into a loss function form analogous to (3). This yields the TDP loss

$$L(\theta) := L_\eta^*(\theta) - \zeta_t L_{Q_\phi}(\theta), \tag{9}$$

where

$$L_\eta^*(\theta) := \mathbb{E}_{s\sim\mathcal{D}} \left[ D(\pi_\eta^*(\cdot|s) \parallel \pi_\theta(\cdot|s)) \right]. \tag{10}$$

For divergence $D$, the natural choice is the divergence induced by the diffusion loss $L_{\text{diff}}(\theta)$, which we refer to as the *diffusion divergence* $D_{\text{diff}}$. Under this choice, the term $L_\eta^*(\theta)$ can be written in the same form as the diffusion loss, except that actions are sampled from $\pi_\eta^*$:

$$L_\eta^*(\theta) := \mathbb{E}_{s\sim\mathcal{D},a^0\sim\pi_\eta^*(a^0|s),k\sim U[1,K]} \left[ ||\epsilon_\theta(a^k,k,s) - \epsilon||^2 \right]. \tag{11}$$

This is infeasible since we cannot directly sample from $\pi_\eta^*$. For this, we aim to construct a surrogate of (11) that *preserves its minimizer*. By the minimum mean square error (MMSE) estimator property, the minimizer of (11) is $\mathbb{E}_{a^0\sim\pi_\eta^*(a^0|a^k,s)}[\epsilon]$, which we denote to $\epsilon_\eta^*(a^k,k,s)$. Analogously, the minimizer of (5) is $\epsilon_\beta(a^k,k,s) := \mathbb{E}_{a^0\sim\pi_\beta(a^0|a^k,s)}[\epsilon]$.

To obtain a tractable surrogate that preserves the minimizer of (11), the simplest approach is to sample state–action pairs $(s, a^0)$ directly from the dataset $\mathcal{D}$. As a result, natural surrogate loss becomes

$$L_\eta^*(\theta) := \mathbb{E}_{(s,a)\sim\mathcal{D},k\sim U[1,K]} \left[ ||\epsilon_\theta(a^k,k,s) - \hat\epsilon||^2 \right]. \tag{12}$$

By the MMSE estimator property, the minimizer of (12) is $\mathbb{E}_{a^0\sim\pi_\beta(a^0|a^k,s)}[\hat\epsilon]$. Since $\epsilon_\beta(a^k,k,s)$ is defined as $\epsilon_\beta(a^k,k,s) := \mathbb{E}_{a^0\sim\pi_\beta(a^0|a^k,s)}[\epsilon]$, our choice is $\hat\epsilon := \epsilon + \epsilon_\eta^*(a^k,k,s) - \epsilon_\beta(a^k,k,s)$. With this construction, the conditional expectation of $\hat\epsilon$ becomes exactly $\epsilon_\eta^*(a^k,k,s)$. Therefore, the minimizer of the surrogate loss (12) coincides with the minimizer of the original objective (11).

Though $\epsilon_\eta^*(a^k,k,s)$ nor $\epsilon_\beta(a^k,k,s)$ is directly computable, we only need their difference to construct $\hat\epsilon$. To make this computable, we first express them by the score functions of the noisy action distributions, which are induced by the noise-free action policies. For $\pi_\eta^*(a^0|s)$, the corresponding distribution over the noisy action $a^k$ is obtained by marginalizing the noise-free action $a^0$ through the forward diffusion kernel $q(a^k|a^0)$, i.e., $\pi_\eta^*(a^k|s) := \int \pi_\eta^*(a^0|s)q(a^k|a^0)da^0$. Similarly, the noisy action distribution induced by $\pi_\beta(a^0|s)$ is $\pi_\beta(a^k|s) := \int \pi_\beta(a^0|s)q(a^k|a^0)da^0$. These noisy distributions admit a score function relationship with the corresponding noise estimates. As shown in prior works (Luo, 2022; Fang et al., 2024), along with full derivation in Appendix D.7, we get $\epsilon_\eta^*(a^k,k,s) = -\sqrt{1-\bar\alpha^k}\nabla_{a^k}\log\pi_\eta^*(a^k|s)$ and $\epsilon_\beta(a^k,k,s)$ satisfies $\epsilon_\beta(a^k,k,s) = -\sqrt{1-\bar\alpha^k}\nabla_{a^k}\log\pi_\beta(a^k|s)$.

From the relationships above, we have

$$\epsilon_\eta^*(a^k, k, s) - \epsilon_\beta(a^k, k, s) = -\sqrt{1 - \bar{\alpha}^k} \nabla_{a^k} (\log \pi_\eta^*(a^k|s) - \log \pi_\beta(a^k|s)). \quad (13)$$

Instead of directly using $g(a^k, k, s) := \log \pi_\eta^*(a^k|s) - \log \pi_\beta(a^k|s)$, Lu et al. (2023) has shown that

$$f(a^k, k, s) := \log \mathbb{E}_{a^0 \sim \pi_\beta(a^0|a^k,s)} \left[ \exp\left( \frac{1}{\eta} Q_\phi(s, a^0) \right) \right] = g(a^k, k, s) + \log Z(s), \quad (14)$$

where $Z(s)$ is the partition function in $\pi_\eta^*(a^0|s) = \frac{1}{Z(s)} \pi_\beta(a^0|s) \exp(\frac{1}{\eta} Q_\phi(s, a))$. Since $Z(s)$ does not depend on $a^k$, it disappears when taking the gradient with respect to $a^k$. Therefore, $f(a^k, k, s)$ can be used equivalently to $g(a^k, k, s)$ for constructing the noise update.

However, this formulation would require training auxiliary models for both $\pi_\beta$ and $f$, which makes the procedure computationally expensive and unnecessarily complex. Our goal is to avoid introducing any additional models. To achieve this, we approximate $f(a^k, k, s)$ using a first-order Taylor expansion, which gives

$$\tilde{f}(a^k, k, s) := \frac{1}{\eta} \mathbb{E}_{a^0 \sim \pi_\beta(a^0|a^k,s)}[Q_\phi(s, a^0)]. \quad (15)$$

One can easily show that using the following one-sample estimate is equivalent to using $\hat{f}(a^k, k, s)$ by MMSE estimator property:

$$\hat{\epsilon} \approx \epsilon - \frac{1}{\eta} \sqrt{1 - \bar{\alpha}^k} \nabla_{a^k} Q_\phi(s, a^0), \quad (16)$$

where $a^0$ is a *true* noise-free action generating $a^k$ during training with $a^k = \sqrt{\bar{\alpha}^k} a^0 + \sqrt{1 - \bar{\alpha}^k} \epsilon$ and $\epsilon \sim \mathcal{N}(0, I)$. We call this *noise-free estimation*, which works in the noise domain but relies only on noise-free actions. Though slightly biased, it is simple, reduces variance, and is highly effective in practice. Note that $\nabla_{a^k} Q_\phi(s, a^0) = \frac{\partial a^0}{\partial a^k} \cdot \nabla_{a^0} Q_\phi(s, a^0)$ is well-defined with $\frac{\partial a^0}{\partial a^k} = \sqrt{\frac{1}{\bar{\alpha}^k}}$.

Next, we examine the bias introduced by this approximation. The bias of $\hat{f}(a^k, k, s)$ relative to $f(a^k, k, s)$ is (see Appendix D.7 for the full proof)

$$\hat{f}(a^k, k, s) - f(a^k, k, s) = -D_{\text{KL}}(\pi_\beta(a^0|a^k, s) \parallel \pi_\eta^*(a^0|a^k, s)). \quad (17)$$

When $\pi_\eta^*(a^0|a^k, s)$ strongly favors high Q actions, the discrepancy between the two posteriors grows, and the negative KL term counteracts such shifts. This conservativeness suppresses movement away from the dataset distribution, thereby mitigating OOD action generation.

Alternative strategies for estimating $f$ include: (i) generating multiple $a^0$ samples from a pretrained diffusion BC model to compute (14), and (ii) *noisy estimation* (Fang et al., 2024), which replaces $\nabla_{a^k} Q_\phi(s, a^0)$ with $\nabla_{a^k} Q_\phi(s, a^k)$, relying on a naive extension of $Q_\phi$ to the noise domain. The first option adds cost without gains, while the second is unreliable since $Q_\phi$ is not trained on $a^k$, potentially making such inputs OOD. We therefore regard noise-free estimation as important to the success of TDP. Results for (i) appear in Appendix F.3, (ii) are illustrated in Sec. 3.3.

### 3.3 2D BANDIT EXPERIMENTS

To further validate both noise-free estimation and the two-fold improvement, we consider the 2D bandit environment. Following Wang et al. (2023), we adopt their setup and use the original DQL implementation (as an explicit method) to avoid bias, while reimplementing the other methods to reflect our design choices for consistency. In the 2D bandit experiments, each algorithm reflects the actor loss design of its offline RL counterpart, while all other components are aligned, enabling a controlled comparison of actor loss designs.

**Settings** The 2D bandit experiment uses a single state with a 2D box action space containing four clusters. The objective is to imitate the highest-reward cluster (bottom right, red) while remaining within the in-distribution regions (purple). To compare noise-free and noisy estimation, we evaluate four variants: TDP itself ("TDP"), the noisy estimate ("Noisy"), TDP without the Q loss ("Implicit"), and their combination ("Noisy+Implicit"). In addition, we compare TDP with explicit ("Explicit") and implicit ("Implicit") policy constraint methods. The results are visualized in Fig. 1. Additional details are provided in Appendix E.1 and Wang et al. (2023).

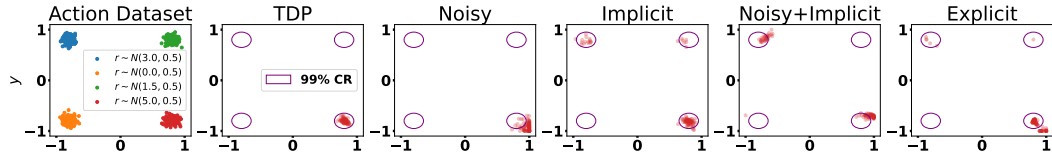

Figure 1: A visualization of sampled actions from various algorithms in the 2D bandit experiments.

**Noise-Free vs. Noisy Estimation**    TDP matches the desired distribution without producing OOD actions, while the noisy estimate variant yields biased OOD actions with incorrect centers. Removing the Q loss makes both suboptimal, but the noise-free version stays in-distribution, while the noisy one fails and produces OOD actions. Overall, only TDP successfully recovers the optimal region.

**Two-Fold vs. Explicit vs. Implicit**    As mentioned in Sec. 3.1, explicit methods tend to generate OOD samples, while implicit methods fail to eliminate suboptimal ones. Both effects appear in the 2D bandit experiments, while TDP succeeds, demonstrating the strength of the two-fold improvement.

### 3.4 Practical Implementation

We now present additional implementation details of TDP, including critic and actor updates, together with computation-saving techniques for the actor to reduce computations. These components ensure robust and efficient training while realizing the Q function–oriented two-fold improvement.

**Critic Update**    We use temporal difference (TD) learning for critic updates and additionally apply the lower confidence bound (LCB) of the Q ensemble (Ghasemipour et al., 2022; Fang et al., 2024) to enhance robustness. The LCB is defined as $Q_{\bar{\phi}} = \mathbb{E}[Q_{\bar{\phi}}] - \rho\sigma_{\bar{\phi}}$, where $\rho$ controls the confidence interval and $\sigma_{\bar{\phi}}$ is the ensemble standard deviation. As in Fang et al. (2024), we also sample $N_a$ actions from the diffusion policy and compute mean LCB targets over them.

**Actor Update with Computation-Saving Techniques**    Recall the actor loss (9). For theoretical analysis, we assume a fixed target divergence value $\varepsilon_b$ across timesteps (Sec. 4). In practice, however, adaptively changing $\zeta_t$ according to $\varepsilon_b$ is difficult. Following prior works, we therefore fix $\zeta_t$ to a constant $\zeta$, justified by the empirical observation that $L_\eta^*(\theta)$ converges rapidly.

Since we employ Q ensembles to improve critic quality, the actor update must be made more efficient to remain computationally feasible. We focus on three parts of the actor loss training for cost reduction: (i) the actor update itself, (ii) Q gradient computation in $L_\eta^*(\theta)$, and (iii) the Q loss term.

For the actor update itself, we apply a delayed policy update, which updates the actor every $d > 1$ steps (typically $d = 2$) instead of after each Q update, following He et al. (2023). For Q gradient computation and Q loss, we use only half of the batches, which we found to have no noticeable effect on performance. Additionally for the Q loss, we adopt the EDP (Kang et al., 2023)-style formulation, replacing full-step denoising (from $a^K \sim \mathcal{N}(0, I)$) with a one-step estimate,

$$\hat{a}^0 := \frac{a^k - \sqrt{1 - \bar{\alpha}^k}\epsilon_\theta(a^k, k, s)}{\sqrt{\bar{\alpha}^k}},\tag{18}$$

where $a^k = \sqrt{\bar{\alpha}^k}a^0 + \sqrt{1 - \bar{\alpha}^k}\epsilon$, $(s, a^0) \sim \mathcal{D}$, and $\epsilon \sim \mathcal{N}(0, I)$. This reduces computation by using a one-step gradient, while reusing the same $\epsilon_\theta$ values as in $L_\eta^*(\theta)$. Overall, these techniques make TDP both efficient and stable in practice.

We provide the additional algorithm details and algorithm pseudocodes in Appendix C.

**Remark** Our two-fold approach, optimizing $\pi_\theta$ to be close to $\pi_\eta^*$ and to maximize the Q value in addition, can be applied to other expressive policies like recent flow-based policies. Our two-fold approach increases performance again when applied to flow-based policies, as shown in Sec. 6.3.

## 4 THEORETICAL ANALYSIS

We now turn to the theoretical properties of CDPO and TDP. As the diffusion divergence is not an $f$-divergence (with respect to $a^0$), we first establish strong duality to ensure the equivalence of the primal and the dual form of CDPO, as mentioned in Appendix D.1.3. This serves as the foundation for our main result, the *policy enhancement theorem*, which provides a stronger guarantee than strong duality. As a corollary, we derive an *approximate improvement* result. Finally, we present an approximate in-distribution property for CDPO, based on a novel formulation of in-distribution. Before we start

**Policy Enhancement Theorem**   One challenge is that the *diffusion divergence*, induced by the diffusion loss, is not an $f$-divergence, raising concerns about the validity of the CDPO dual form. Fortunately, it can be viewed as a *pathwise forward KL divergence* (See Appendix D.2), resolving this issue. Utilizing this with assuming a fixed divergence constraint $\varepsilon_b$ across timesteps, we establish the policy enhancement theorem, which guarantees monotone policy improvement for CDPO.

**Theorem 4.1** (Policy enhancement theorem). *For CDPO, we can achieve **policy improvement** for the first iteration step, i.e.*

$$J(\pi^{(1)}) \geq J(\pi^{(0)}) = J(\pi_0), \tag{19}$$

*where $J(\pi)$ is the expected return of policy $\pi$. Furthermore, we can achieve **expected Q value improvement** for all the other iteration steps, i.e. for $t \geq 1$,*

$$\mathbb{E}_{s\sim\rho^{\pi_\beta}, a\sim\pi^{(t+1)}}[Q^{\pi^{(t)}}(s,a)] \geq \mathbb{E}_{s\sim\rho^{\pi_\beta}, a\sim\pi^{(t)}}[Q^{\pi^{(t)}}(s,a)]. \tag{20}$$

Due to the offline nature, we could not achieve a stronger result, and Theorem 4.1 can be considered as an offline version of the policy improvement theorem.

**Approximate Improvement**   Based on Theorem 4.1, it is straightforward to compare several CDPO algorithms summarized in the following corollaries:

**Corollary 4.2.** *1) $\pi_{DQL} \gtrsim \pi_\beta$, 2) $\pi_\eta^* \gtrsim \pi_\beta$, and 3) $\pi_{TDP} \gtrsim \pi_\eta^*$. Here, $\pi_1 \gtrsim \pi_2$ means that policy $\pi_1$ is approximately better than policy $\pi_2$ in the sense of (19) and (20).*

**Corollary 4.3.** *$\pi_\eta^*$ is approximately better than $\pi_{DQL}$ in the following sense. For $\pi_\beta$ satisfying some mild conditions, for any $\zeta_t$ there exists properly chosen $\eta$ satisfying*

$$\mathbb{E}_{s\sim\rho^{\pi_\beta}, a\sim\pi_\eta^*}[Q^{\pi_\beta}(s,a)] \geq \mathbb{E}_{s\sim\rho^{\pi_\beta}, a\sim\pi_{DQL}^{(1)}}[Q^{\pi_\beta}(s,a)] \tag{21}$$

In other words, TDP is theoretically guaranteed to achieve approximate improvements over existing diffusion policies such as DQL (Wang et al., 2023) and DAC (Fang et al., 2024), and this advantage is further supported by the empirical results in Fig. 1 and Sec. 6.

**In-Distribution Property**   We now examine the in-distribution property of CDPO. To this end, we first reformulate the notions of OOD actions and the in-distribution policy. While the natural definition of an OOD action is $\pi_\beta(a|s) = 0$, this condition rarely occurs in continuous control. Instead, we adopt $\pi_\beta(a|s) < \delta$ for some small $\delta$ as the criterion for OOD actions. Moreover, since completely preventing OOD actions is infeasible, we relax this requirement by considering averages. In this line, we define a $(\delta, \varepsilon)$-in-distribution policy as follows:

**Definition 4.4.** *For $(\delta, \varepsilon) \in (0,1)^2$, the policy $\pi$ is a $(\delta, \varepsilon)$-in-distribution policy if and only if $\mathbb{E}_{s\sim\rho^{\pi_\beta}, a\sim\pi(\cdot|s)}[\mathbb{P}[\pi_\beta(a|s) < \delta]] < \varepsilon$.*

We can ensure a $(\delta, \varepsilon)$-in-distribution CDPO policy by controlling the expected diffusion divergence.

**Theorem 4.5.** *For any $\pi_0$, proper $\varepsilon, \delta > 0$, with a box action space $\mathcal{A} = [-1,1]^d$, there exists $D_{max} > 0$ such that any policy $\pi$ with $\mathbb{E}_{s\sim\rho^{\pi_\beta}}[D_{diff}(\pi_0(\cdot|s) \parallel \pi(\cdot|s))] \leq D_{max}$ is an $(\delta, \epsilon)$-in-distribution policy.*

This implies that the OOD measure of TDP can be bounded through the expected diffusion divergence, ensuring both a $(\delta, \varepsilon)$-in-distribution policy and the correctness of the Q function.

See Appendices D.3, D.4, D.5, and D.6 for proofs of Theorem 4.1, Corollaries 4.2 and 4.3, and Theorem 4.5, respectively.

**Remark** Theorem 4.1 is first proven for the general $f$-divergence case and Theorem 4.5 for forward and reverse KL divergences, before both are specialized to CDPO. As a result, these theorems extend naturally to generalized CPO and ordinary policy constraint methods, thereby unifying CDPO with the classical theory and providing a solid foundation for theoretical analyses.

## 5 RELATED WORKS

**Policy Constraint Methods for Offline RL** Policy constraint methods are widely used in offline RL, with explicit or implicit forms. Explicit methods use BC-based divergences (Fujimoto & Gu, 2021; Wang et al., 2023) or reverse KL by policy log-likelihood (An et al., 2021) or convex conjugate formulation (Wu et al., 2019). Implicit methods mimic the reverse KL-constrained optimal policy through weighted losses (Nair et al., 2020; Kostrikov et al., 2022) or energy guidance (Lu et al., 2023; Fang et al., 2024). Some approaches use non-Gaussian BC models to more effectively compute forward KL–like losses (Wu et al., 2022; Zhang et al., 2023). Variants include BC with Q loss under Decision Transformer (Chen et al., 2021) backbones (Hu et al., 2024; Kim et al., 2024). Other approaches (Kumar et al., 2020; Mao et al., 2023; Xu et al., 2023; Tarasov et al., 2023) propose alternative actor losses, which can more broadly be interpreted as policy constraint methods.

**Diffusion Models for Offline RL** Building on the success of diffusion models (Sohl-Dickstein et al., 2015; Ho et al., 2020; Song et al., 2021), early offline RL works applied them to return-guided trajectory planning (Janner et al., 2022; Ajay et al., 2023). Wang et al. (2023) introduced diffusion policies by state-conditioned action denoising with a linear combination of diffusion and Q losses. Extensions include Q value-based action selection (Hansen-Estruch et al., 2023), external energy guidance (Lu et al., 2023), integrated guidance (Fang et al., 2024), one-step Q loss estimation (Kang et al., 2023), and adaptive Q loss weighting (He et al., 2023). Recent works apply reverse KL–based formulations for improved optimization (Zhang et al., 2024; Gao et al., 2025), while others distill diffusion BC policies into deterministic or Gaussian models (Chen et al., 2024a;b).

**Diffusion Models for Online RL** Diffusion models have also been extended to online RL, taking advantage of their policy-based structure for direct interaction with the environment. Early work applies action gradient updates to dataset actions with the diffusion loss (Yang et al., 2023). Subsequent approaches introduce energy-based formulations with noisy estimation (Psenka et al., 2024), TD3-style training with Gaussian mixture models for noise scale adjustment (Wang et al., 2024), or mixing diffusion losses from uniformly sampled actions combined with rejection sampling (Ding et al., 2024). Recent work further employs energy-based Q weighting with reverse sampling to improve sample efficiency (Ma et al., 2025).

## 6 EXPERIMENTS

In this section, we evaluate TDP on the D4RL benchmark (Fu et al., 2020) and perform ablations on noise-free approximation and policy expressiveness with flow policies.

### 6.1 EXPERIMENTAL SETUP

**Environments** We evaluated TDP on D4RL MuJoCo (Todorov et al., 2012) locomotion, Antmaze, and Kitchen tasks. The benchmark includes nine locomotion tasks (three agents × three optimality levels), six standard and two ultra Antmaze tasks (Jiang et al., 2023), and Kitchen partial and mixed. Final scores are reported, and implementation and hyperparameter details are in Appendix E.2.

**Baselines** TDP inherits the following features: two-fold improvement, noise-free estimation. diffusion policy, and Q ensemble. We compared it against policy constraint methods (Kumar et al., 2020; Kostrikov et al., 2022; Tarasov et al., 2023; Mao et al., 2023), diffusion policies (Wang et al., 2023; Lu et al., 2023; He et al., 2023; Fang et al., 2024), and Q ensemble methods (An et al., 2021) for locomotion. Note that Fang et al. (2024) also corresponds to an implicit method with noisy estimation. For Antmaze, we used goal-conditioned baselines (Kostrikov et al., 2022; Xu et al., 2022; Zeng et al., 2023; Park et al., 2023), and for Kitchen, DT (Chen et al., 2021). Details are in Appendix E.4.

Table 1: The normalized scores in locomotion tasks. The best score for each task is in bold. Here, "m" denotes medium, "m-r" denotes medium-replay, and "m-e" denotes medium-expert.

| Dataset | Policy Constraint | | | Diffusion Policy | | | | Q Ensemble | | |
|---|---|---|---|---|---|---|---|---|---|---|
| | IQL | ReBRAC | SVR | DQL | QGPO | DiffCPS | DAC | SAC-N | EDAC | TDP |
| Halfcheetah m | 47.4 | 65.6 | 60.5 | 51.1 | 54.1 | **71.0** | 59.1 | 67.5 | 65.9 | 69.7 $\pm$ 0.9 |
| Hopper m | 66.3 | 102.0 | **103.5** | 90.5 | 98.0 | 100.1 | 101.2 | 100.3 | 101.6 | 103.0 $\pm$ 0.1 |
| Walker2d m | 78.3 | 82.5 | 92.4 | 87.0 | 86.0 | 90.9 | 96.8 | 87.9 | 92.5 | **108.0** $\pm$ 2.0 |
| Halfcheetah m-r | 44.2 | 51.0 | 52.5 | 47.8 | 47.6 | 50.5 | 55.0 | **63.9** | 61.3 | 58.5 $\pm$ 0.6 |
| Hopper m-r | 94.7 | 98.1 | 103.7 | 101.3 | 96.9 | 101.1 | 103.1 | 101.8 | 101.0 | **104.4** $\pm$ 0.2 |
| Walker2d m-r | 73.9 | 77.3 | 95.6 | 95.5 | 84.4 | 91.3 | 96.8 | 78.7 | 87.1 | **109.2** $\pm$ 3.1 |
| Halfcheetah m-e | 86.7 | 101.1 | 94.2 | 96.8 | 93.5 | 100.3 | 99.1 | **107.1** | 106.3 | 100.3 $\pm$ 0.2 |
| Hopper m-e | 91.5 | 107.0 | 111.2 | 111.1 | 108.0 | **112.1** | 111.7 | 110.1 | 110.7 | 111.4 $\pm$ 0.6 |
| Walker2d m-e | 109.6 | 111.6 | 109.3 | 110.1 | 110.7 | 113.1 | 113.6 | **116.7** | 114.7 | 113.6 $\pm$ 0.6 |
| Average | 77.0 | 88.5 | 91.4 | 87.9 | 86.6 | 92.3 | 92.9 | 92.7 | 93.5 | **97.6** |

Table 2: The normalized scores in Antmaze tasks. The best score for each task is in bold. Here, "p" denotes play and "d" denotes diverse.

| Dataset | Policy Constraint | Diffusion Policy | | | | Goal-Conditioned | | | | |
|---|---|---|---|---|---|---|---|---|---|---|
| | IQL | DQL | QGPO | DiffCPS | DAC | GC-IQL | GC-POR | HIQL | GCPC | TDP |
| Umaze | 92.8 | 93.4 | 96.4 | 97.4 | **99.5** | 91.6 | 90.6 | 86.5 | 71.2 | 98.5 $\pm$ 1.7 |
| Umaze d | 71.2 | 66.2 | 74.4 | 87.4 | 85.0 | **88.8** | 71.3 | 83.5 | 71.2 | 73.0 $\pm$ 4.1 |
| Medium p | 75.8 | 76.6 | 83.6 | 88.2 | 85.8 | 82.6 | 71.4 | 84.1 | 70.8 | **92.5** $\pm$ 4.3 |
| Medium d | 76.6 | 78.6 | 83.8 | 87.8 | 84.0 | 76.2 | 74.8 | 86.8 | 72.2 | **90.5** $\pm$ 3.3 |
| Large p | 50.0 | 46.4 | 66.6 | 65.6 | 50.3 | 40.0 | 63.2 | **86.1** | 78.2 | 81.0 $\pm$ 8.7 |
| Large d | 52.6 | 56.6 | 64.8 | 63.6 | 55.3 | 29.8 | 49.0 | **88.2** | 80.6 | 78.0 $\pm$ 3.2 |
| Ultra p | 21.2 | 22.0 | 7.0 | 14.5 | 39.0 | 20.6 | 31.0 | 39.2 | 56.6 | **69.5** $\pm$ 6.4 |
| Ultra d | 17.8 | 26.0 | 12.0 | 42.5 | 53.0 | 28.4 | 29.8 | 52.9 | 54.6 | **62.5** $\pm$ 3.0 |
| Average | 57.2 | 58.2 | 61.1 | 68.4 | 69.0 | 57.2 | 60.1 | 75.9 | 69.4 | **80.7** |

Table 3: The normalized scores in Kitchen tasks. The best score for each task is in bold. Here, "p" denotes partial and "m" denotes mixed.

| Dataset | CQL | IQL | DQL | DAC | DT | GC-IQL | GCPC | TDP |
|---|---|---|---|---|---|---|---|---|
| Kitchen p | 49.8 | 46.3 | 60.5 | 50.0 | 48.6 | 74.7 | 90.2 | **99.0** $\pm$ 1.7 |
| Kitchen m | 51.0 | 51.0 | 62.6 | 60.2 | 50.0 | 74.6 | 75.6 | **100.0** $\pm$ 0.0 |

## 6.2 RESULTS

As shown in Table 1, TDP achieves strong performance across all locomotion tasks, attaining the highest average score. In particular, it yields large gains on suboptimal datasets, surpassing the best baseline by 11.2 and 12.4 points on Walker2d medium and medium-replay, respectively. Given that locomotion performance is already highly saturated, even a margin of 4-5 points on average over strong baselines represents a significant improvement.

As shown in Table 2, TDP outperforms all baselines by at least 4.8 points on average in Antmaze, including goal-conditioned methods, and achieves notable gains on the ultra tasks. This stems from its mechanism: since TD targets rely on $a' \sim \pi_{\bar{\theta}}(\cdot|s')$, TD-based methods need a strong policy to improve the Q function, forming a positive feedback loop. With its two-fold improvement, TDP enables progress even with suboptimal Q functions, which is critical for long-horizon, sparse-reward tasks such as Antmaze ultra.

As shown in Table 3, TDP surpasses all baselines by a wide margin in Kitchen environments and is the first to achieve near-perfect completions on partial and mixed tasks. Given the sparse-reward and multi-turn nature of Kitchen, these results underscore the effectiveness of two-fold improvements.

The key strength of TDP lies in its robustness across diverse environments. Unlike prior methods that excel only in specific domains, TDP consistently shows strong results. It achieves substantial

Table 4: The normalized final scores of TDP and its noisy estimation variant in Antmaze large and ultra tasks. The best score for each task is in bold.

Table 5: The average normalized scores of TFP in locomotion and Antmaze tasks. The best score is in bold. Locomotion tasks include all 9 tasks in main experiments, and Antmaze tasks include 6 tasks, from umaze to large diverse.

| Algorithm | Large p | Large d | Ultra p | Ultra d |
|---|---|---|---|---|
| TDP | **81.0** | **78.0** | **69.5** | **62.5** |
| Noisy Estimation | 78.0 | 69.0 | 66.5 | 50.0 |

| Dataset | FQL | FBRAC | IFQL | FlowQ | TDP | TFP |
|---|---|---|---|---|---|---|
| Locomotion | 79.3 | 82.0 | 74.2 | 85.6 | **96.7** | 93.1 |
| Antmaze | 83.5 | 63.7 | 64.8 | 69.3 | 85.6 | **87.5** |

gains on suboptimal datasets and on challenging sparse-reward, long-horizon, or multi-turn tasks, where leveraging the Q function is critical, even surpassing goal-conditioned methods tailored to such settings. At the same time, TDP remains competitive on high-optimality datasets, which rely less on Q function-based optimization, demonstrating both versatility and reliability.

We provide the learning curves of TDP in Appendix G.

## 6.3 Ablations

In this section, we empirically verify two aspects of our approach: (i) the effectiveness of noise-free estimation by comparing TDP with its noisy estimation variant on Antmaze large and ultra, and (ii) the role of policy expressiveness by implementing two-fold improvement in the flow policy framework (Park et al., 2025; Alles et al., 2025), which uses flow matching (Lipman et al., 2023) with a vector field model.

**Noise-Free vs. Noisy Estimation in Complex Tasks**  As discussed in Sec. 3.2, TDP employs noise-free estimation, $f(a^k, k, s) \approx \frac{1}{\eta} Q_\phi(s, a^0)$, while an alternative noisy estimation, $f(a^k, k, s) \approx \frac{1}{\eta} Q_\phi(s, a^k)$, is also possible. We compared the two estimations on Antmaze large and ultra. As shown in Table 4, noise-free estimation consistently outperforms noisy estimation, confirming the effectiveness of noise-free estimation. Additional results are provided in Appendix F.1.

**Flow Policy Variant**  As mentioned in Sec. 3.1, policy expressiveness is a key to realizing optimization benefits. Flow policies (Park et al., 2025; Alles et al., 2025) are fast and expressive, but generally fall slightly behind diffusion policies on locomotion tasks and lack the theoretical guarantees of Sec. 4. To test the universality of expressiveness, we implemented the *two-fold improved flow policy* (TFP) with noise-free estimation, building on FlowQ (Alles et al., 2025), without extra techniques such as Q ensembles. We evaluated TFP on nine locomotion and six Antmaze tasks against TDP and flow policy baselines. Each flow policy baseline corresponds to a diffusion policy: FBRAC corresponds to DQL, IFQL to IDQL Hansen-Estruch et al. (2023), FlowQ to DAC, and FQL to DQL with distillation. As shown in Table 5, TFP consistently outperforms flow policy baselines and performs comparably to TDP, showing that policy expressiveness is important and that two-fold improvement with noise-free estimation is effective across expressive policies. Further details are in Appendices B, E.3, and E.4.

Ablations on various hyperparameters are provided in Appendix F.1, and Appendix F.2 shows training and evaluation time comparisons.

## 7 Conclusion

In this work, we organized CDPO, a generalized framework for policy constraint methods with diffusion policies that unifies explicit and implicit approaches. Within this framework, we proposed TDP, which aims to stay close to the reverse KL-constrained optimal policy and incorporates a nonzero Q loss. A key ingredient is noise-free estimation, reducing variance with a single unnoised action sample. We provided theoretical guarantees, including policy enhancement, approximate performance lower bound, and in-distribution properties, and showed empirically that TDP consistently outperforms most offline RL algorithms. Future work includes exploring stronger anchor policies, possibly the optimal one, and improving critic learning beyond TD methods to further enhance TDP, especially on high-optimality datasets.

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

# A  DETAILS OF DIFFUSION MODELS AND DDPM

Diffusion models (Sohl-Dickstein et al., 2015; Ho et al., 2020; Song et al., 2021) were originally proposed for image generation by learning the data distribution through a progressive noising and denoising process. Given an image sample $x^0$, the model gradually transforms it into a standard Gaussian variable $x^K$. By learning the reverse denoising dynamics, the model can generate realistic images from Gaussian noise.

The $k$-th step of the forward process, which adds independent noise to a scaled version of the previous output, follows a Gauss-Markov process:

$$q(x^k|x^{k-1}) := \mathcal{N}(\sqrt{1-\beta^k}x^{k-1}, \beta^k I) \tag{22}$$

for $k = 1, \cdots, K$. Combining the steps up to $k$, we have

$$q(x^k|x^0) := \mathcal{N}(\sqrt{\bar{\alpha}^k}x^0, (1-\bar{\alpha}^k)I), \tag{23}$$

where $\alpha^k := 1 - \beta^k$ and $\bar{\alpha}^k := \prod_{l=1}^{k} \alpha^l$, or equivalently,

$$x^k = \sqrt{\bar{\alpha}^k}x^0 + \sqrt{1-\bar{\alpha}^k}\epsilon, \quad \text{with } \epsilon \sim \mathcal{N}(0, I). \tag{24}$$

The reverse process in the denoising diffusion probabilistic model (DDPM) consists of $K$ steps, where each step is derived using Bayes' rule. Specifically, given $x^0$, the $k$-th reverse step can be obtained from equations (22) and (23) as follows:

$$q(x^{k-1}|x^k, x^0) = \frac{q(x^k|x^{k-1})q(x^{k-1}|x^0)}{q(x^k|x^0)} = \mathcal{N}(\sqrt{1-\beta^k}x^{k-1}, \beta^k\mathbf{I})\frac{\mathcal{N}(\sqrt{\bar{\alpha}^{t-1}}x^0, (1-\bar{\alpha}^{k-1})I)}{\mathcal{N}(\sqrt{\bar{\alpha}^k}\,x^k, (1-\bar{\alpha}^k)I)}, \tag{25}$$

which is Gaussian. Our goal is to model this reverse process without knowing $x^0$. For this, we use the following formulation:

$$p_\theta(x^{0:K}) = \underbrace{\mathcal{N}(0, I)}_{\sim x^K} \prod_{k=1}^{K} p_\theta(x^{k-1}|x^k). \tag{26}$$

Since we do not know $x^0$ in the reverse process, we use its estimate $\hat{x}^0$ by using a noise prediction model $\epsilon_\theta(x^k, k)$, which predicts the noise $\epsilon$ added to $x^0$ to generate $x^k$ in (24). Then, from (24) we have $\hat{x}^0 = \frac{x^k - \sqrt{1-\bar{\alpha}^k}\epsilon_\theta(x^k, k)}{\sqrt{\bar{\alpha}^k}}$. By plugging this to (25), we obtain distribution $p_\theta(x^{k-1}|x^k)$, which is an approximate of $q(x^{k-1}|x^k, x^0)$, as follows:

$$p_\theta(x^{k-1}|x^k) = \mathcal{N}\left(\frac{1}{\sqrt{\alpha^k}}(x^k - \frac{1-\alpha^k}{\sqrt{1-\bar{\alpha}^k}}\epsilon_\theta(x^k, k)), (\beta^k)^2\frac{1-\bar{\alpha}^{k-1}}{1-\bar{\alpha}^k}\mathbf{I}\right).$$

Then, the $k$-th reverse sampling step is given as

$$x^{k-1} = \frac{1}{\sqrt{\alpha^k}}\left(x^k - \frac{1-\alpha^k}{\sqrt{1-\bar{\alpha}^k}}\epsilon_\theta(x^k, k)\right) + \mathbf{1}_{k>1}\beta^k\sqrt{\frac{1-\bar{\alpha}^{k-1}}{1-\bar{\alpha}^k}}z, \tag{27}$$

where $z \sim \mathcal{N}(0, \mathbf{I})$. With $p_\theta$ and $x^K \sim \mathcal{N}(0, \mathbf{I})$, we can get a generated image $x^0$ by iteratively applying the reverse step (27) for $k = K, K-1, \cdots, 1$. Note that we can realize the samples of a complex distribution through $K$ iterative applications of the sampling process (27) although we learn the single noise model $\epsilon_\theta(x^k, k)$ only.

In the training phase, we have $x^0$, generate $\epsilon$ and $x^k$ according to (24). Hence, we know $(x^0, x^k, \epsilon)$ for the training dataset. To train the noise model $\epsilon_\theta(x^k, k)$, DDPM uses evidence lower bound (ELBO)-based loss, which is lower bounded by $\mathbb{E}_{x_0 \sim q(x_0)}[-\log p_\theta(x_0)]$, and it is simplified as the following mean squared error (MSE) loss:

$$L_{\text{diff}}^{\text{DDPM}}(\theta) := \mathbb{E}_{x^0 \sim q(x^0), k \sim U[1,K]}\left[w(k)\|\epsilon_\theta(x^k, k) - \epsilon\|^2\right], \tag{28}$$

where $w(k)$ is a proper weight to match the original loss. Practically, DDPM uses $w(k) \equiv 1$, which is used in this paper too, but for theoretical analysis, we use the formulation of (28).

## B FLOW MATCHING AND FM POLICIES

### B.1 FLOW MATCHING

Flow matching (Lipman et al., 2023) is a generative modeling approach for multimodal distributions. Unlike diffusion models, which rely on stochastic differential equations (SDEs), flow matching employs ordinary differential equations (ODEs), simplifying training and often enabling faster inference with competitive or superior sample quality.

The key idea is to learn a vector field $u(x^t, t)$ that transports samples from a noise distribution $p_0 = \mathcal{N}(0, I)$ to the target distribution $p_1$, whose probability density is unknown but from which we can draw samples. Formally, the flow $\psi(x^0, t)$ satisfies

$$\frac{d}{dt}\psi(x^0, t) = u(\psi(x^0, t), t), \tag{29}$$

so that for $x^0 \sim p_0$, the trajectory $x^t = \psi(x^0, t)$ induces an intermediate distribution $p_t := \psi(\cdot, t)\sharp p_0$.

Since the probability density of $p_1$ is unknown, we cannot compute $u(x^t, t)$ directly. Instead, flow matching trains a parameterized vector field $u_\theta(x^t, t)$ to approximate it with the conditional flow matching (CFM) loss

$$L(\theta) := \mathbb{E}_{x^0 \sim p_0, x^1 \sim p_1, t \sim U([0,1])} \left[ ||u_\theta(x^t, t) - u(x^t, t|x^1)||^2 \right], \tag{30}$$

where $u(x^t, t|x^1)$ is a conditional vector field.

This gives rise to *conditional flow matching*, which defines an interpolation distribution

$$p_t(\cdot|x^1) = \mathcal{N}(\mu(x^1, t), \sigma(t)^2 I),$$

with interpolated samples $x^t = \psi(x^0|x^1) = \mu(x^1, t)x^1 + \sigma(t)x^0$ and target vector field

$$u(x^t, t|x^1) = \mu'(t)x^1 - \frac{\sigma'(t)}{\sigma(t)}(x^t - \mu(x^1, t)x^1).$$

In practice, $\mu$ and $\sigma$ are often chosen according to optimal transport (OT), i.e., $\mu(x^1, t) = t$ and $\sigma(t) = 1 - t$, yielding the simple interpolation

$$x^t = tx^1 + (1 - t)x^0, \quad u(x^t, t|x^1) = x^1 - x^0,$$

which is widely used in flow matching algorithms.

For sampling, we first draw $x^0 \sim p_0 = \mathcal{N}(0, I)$ and then integrate the learned vector field $u_\theta(x^t, t)$ using an ODE solver (Euler method in our case). Importantly, the loss in (30) is valid because it induces, in expectation, the same gradient as training against the true field $u(x^t, t)$, ensuring theoretical correctness (Lipman et al., 2023).

Unlike diffusion models, flow models allow direct computation of log probabilities using the continuity equation (Lipman et al., 2023). However, the CFM loss is not related to any $f$-divergence, making theoretical analysis of flow models and flow policies more challenging.

### B.2 FLOW POLICIES

Analogous to diffusion policies, we can define *flow policies* using flow matching, with the CFM loss

$$L_{\text{CFM}}(\theta) := \mathbb{E}_{a^0 \sim \mathcal{N}(0,I), (s,a^1) \sim \mathcal{D}, t \sim U([0,1])} \left[ ||u_\theta(a^t, t|s) - u(a^t, t|a^1, s)||^2 \right]. \tag{31}$$

Park et al. (2025) introduced algorithms that use this formulation.

First, FBRAC corresponds to DQL (Wang et al., 2023), combining TD learning with the actor loss

$$L(\theta) := \beta L_{\text{CFM}}(\theta) - L_{Q_\phi}(\theta), \tag{32}$$

where $L_{Q_\phi}(\theta)$ evaluates full-step denoised actions from $x^0 \sim \mathcal{N}(0, I)$ using $u_\theta(x^t, t|s)$ with Euler method.

Second, IFQL follows an IDQL-style approach (Hansen-Estruch et al., 2023), training Q and V functions and applying rejection sampling based on the Q function, while the flow policy itself is trained with $L_{\text{CFM}}(\theta)$.

Third, following the argument of Kang et al. (2023), Park et al. (2025) observed that the full-step denoising in FBRAC is computationally expensive. To reduce this cost, they proposed a *one-step policy* $v_\theta$ that maps $(s, a^0)$ directly to $a^1$ by distilling the flow BC model $u_\phi$. Let $a_\psi(a^0, s)$ denote the output of the flow BC policy (with Euler method) and $a_\theta(a^0, s)$ the one-step policy. The distillation is trained with

$$L_{\text{distill}}(\theta) := \mathbb{E}_{a^0 \sim \mathcal{N}(0,I), s \sim \mathcal{D}, t \sim U([0,1])} \left[ ||a_\theta(a^0, s) - a_\psi(a^0, s)|| \right], \tag{33}$$

ensuring that the one-step policy mimics the flow BC policy, given the same noise input.

Building on this, FQL jointly trains the diffusion BC policy with $L_{\text{CFM}}(\psi)$ and the one-step policy with

$$L(\theta) := \beta L_{\text{distill}}(\theta) - L_{Q_\phi}(\theta), \tag{34}$$

where $L_{Q_\phi}(\theta)$ (together with the TD loss) relies on the one-step policy, thereby accelerating training and evaluation while preserving multimodality.

Finally, Alles et al. (2025) proposes FlowQ, an implicit policy constraint method using flow policy and noisy estimation, built on FBRAC and incorporating energy guidance (Sohl-Dickstein et al., 2015; Lu et al., 2023). Starting from $\pi_\eta^*(a^t|s) = \frac{1}{\tilde{Z}^t(s)} \pi_\beta(a^t|s) \exp\left(f(a^t, t, s)\right)$, they set

$$f(a^t, t, s) = \alpha h(t) \mathcal{Q}(a^t, t, s),$$

where $\alpha = \frac{1}{\eta}$, $\mathcal{Q}(a^t, t, s)$ is the noisy-domain counterpart of $Q_\phi$, and $h(t)$ is a function with $h(0) = 0$. To ensure that $\pi_\eta^*(a^t|a^1, s)$ remains Gaussian, they applied a first-order Taylor expansion,

$$\mathcal{Q}(a^t, t, s) \approx Q(s, ta^1) + \left\langle \nabla_{ta^1} Q(s, ta^1), (a^t - ta^1) \right\rangle, \tag{35}$$

This yields

$$a^t = ta^1 + (1-t)a^0 + \alpha(1-t)^2 h(t) \nabla_{ta^1} Q(s, ta^1), \tag{36}$$

and

$$\hat{u}(a^t, t|a^1, s) = \frac{a^1 - a^t}{1-t} - \alpha(1-t)(h(t) - (1-t)h'(t)) \nabla_{ta^1} Q(s, ta^1) + \alpha(1-t)^2 h(t) \left\langle \nabla_{ta^1}^2 Q(s, ta^1), a^1 \right\rangle. \tag{37}$$

Among various choices of $h$, $h(t) = \frac{t^2}{1-t}$ empirically performs the best, giving simplified expressions:

$$a^t = ta^1 + (1-t)a^0 + \alpha t^2(1-t) \nabla_{ta^1} Q(s, ta^1), \tag{38}$$

and

$$\hat{u}(a^t, t|a^1, s) = a^1 - a^0 + \alpha t(2 - 3t) \nabla_{ta^1} Q(s, ta^1) + \alpha t^2(1-t) \left\langle \nabla_{ta^1}^2 Q(s, ta^1), a^1 \right\rangle. \tag{39}$$

The resulting actor loss for FlowQ is

$$L(\theta) := \mathbb{E}_{a^0 \sim \mathcal{N}(0,I), (s,a^1) \sim \mathcal{D}, t \sim U([0,1])} \left[ \|u_\theta(a^t, t|s) - \hat{u}(a^t, t|a^1, s)\|^2 \right], \tag{40}$$

while the critic is trained with a standard TD loss, as in FBRAC.

### B.3 Two-Fold Improved Flow Policy (TFP)

We now introduce the two-fold improved flow policy (TFP), a flow-policy analogue of TDP that improves FlowQ with the same core arguments.

For noise-free estimation, we use

$$\mathcal{Q}(a^t, t, s) \approx Q(s, a^1), \tag{41}$$

which yields

$$a^t = ta^1 + (1-t)a^0 + \alpha(1-t)^2 h(t) \nabla_{a^1} Q(s, a^1), \tag{42}$$

and

$$\hat{u}(a^t, t|a^1, s) = \frac{a^1 - a^t}{1-t} - \alpha(1-t)(h(t) - (1-t)h'(t)) \nabla_{a^1} Q(s, a^1). \tag{43}$$

Using $h(t) = \frac{t^2}{1-t}$ (Alles et al., 2025), these simplify to

$$a^t = ta^1 + (1-t)a^0 + \alpha t^2(1-t)\nabla_{a^1}Q(s, a^1), \tag{44}$$

and

$$\hat{u}(a^t, t|a^1, s) = a^1 - a^0 + \alpha t(2 - 3t)\nabla_{a^1}Q(s, a^1). \tag{45}$$

By defining

$$L_\alpha^*(\theta) := \mathbb{E}_{a^0 \sim \mathcal{N}(0,I),(s,a^1) \sim \mathcal{D}, t \sim U([0,1])}\left[||u_\theta(a^t, t|s) - \hat{u}(a^t, t|a^1, s)||^2\right], \tag{46}$$

the TFP actor loss becomes

$$L(\theta) := \beta L_\alpha^*(\theta) - L_{Q_\phi}(\theta), \tag{47}$$

where $L_{Q_\phi}(\theta)$ and critic loss follows FBRAC.

To isolate the effect of our core idea, we intentionally omit practical enhancements such as Q ensemble LCB regularization (Ghasemipour et al., 2022; Fang et al., 2024) (instead we use double Q-learning (Hasselt, 2010)), delayed policy updates, EDP (Kang et al., 2023)-style Q loss, and adaptive $\eta$ (here $\alpha$).

The main limitation of TFP is that, due to the properties of the CFM loss, it cannot be directly linked to the theoretical analyses in Sec. 4.

### B.4  FULL RESULTS

Here, we present the complete results of TFP and the baselines from Sec. 6.3. As shown in Tables 6 and 7, TFP consistently outperforms other flow policy baselines. We highlight comparisons with FBRAC and FlowQ, which represent explicit and implicit policy-constraint approaches with flow policies, making them particularly relevant baselines for TFP. Furthermore, despite the inherent limitations of flow policies relative to diffusion policies and the omission of certain practical implementation techniques, TFP achieves performance competitive with TDP.

Table 6: The normalized scores in locomotion tasks. The best score for each task is in bold. Here, "m" denotes medium, "m-r" denotes medium-replay, and "m-e" denotes medium-expert.

| Dataset | FQL | FBRAC | IFQL | FlowQ | TDP | TFP |
|---------|-----|-------|------|-------|-----|-----|
| Halfcheetah m | 62.8 | 47.6 | 53.6 | 56.8 | 69.7 | **72.6** $\pm$ 1.3 |
| Hopper m | 81.9 | 65.7 | 86.7 | 88.9 | **103.0** | 101.9 $\pm$ 0.5 |
| Walker2d m | 79.8 | 83.7 | 78.2 | 84.9 | **108.0** | 87.0 $\pm$ 0.5 |
| Halfcheetah m-r | 52.3 | 44.6 | 45.5 | 50.3 | 58.5 | **61.2** $\pm$ 2.6 |
| Hopper m-r | 65.0 | 98.3 | 79.4 | 99.2 | **104.4** | 102.2 $\pm$ 0.2 |
| Walker2d m-r | 80.8 | 86.2 | 68.0 | 88.5 | 109.2 | 93.9 $\pm$ 0.5 |
| Halfcheetah m-e | **105.9** | 93.0 | 86.4 | 89.6 | 100.3 | 99.5 $\pm$ 2.0 |
| Hopper m-e | 75.7 | 109.6 | 61.8 | 103.5 | **111.4** | 108.9 $\pm$ 3.8 |
| Walker2d m-e | 109.1 | 109.0 | 108.5 | 109.1 | **113.6** | 111.0 $\pm$ 0.5 |
| Average | 79.3 | 82.0 | 74.2 | 85.6 | **97.6** | 93.1 |

Table 7: The normalized scores in Antmaze tasks. The best score for each task is in bold. Here, "p" denotes play and "d" denotes diverse.

| Dataset | FQL | FBRAC | IFQL | FlowQ | TDP | TFP |
|---|---|---|---|---|---|---|
| Umaze | 96.0 | 94.0 | 92.0 | 92.8 | **98.5** | **98.5** $\pm$ 1.7 |
| Umaze d | 89.0 | 82.0 | 62.0 | 65.2 | 73.0 | **91.5** $\pm$ 3.8 |
| Medium p | 78.0 | 77.0 | 56.0 | 84.8 | **92.5** | 89.0 $\pm$ 5.4 |
| Medium d | 71.0 | 77.0 | 60.0 | 44.7 | 90.5 | 78.5 $\pm$ 11.1 |
| Large p | **84.0** | 32.0 | 55.0 | 72.4 | 81.0 | 82.0 $\pm$ 2.8 |
| Large d | 83.0 | 20.0 | 64.0 | 55.8 | 78.0 | **85.5** $\pm$ 0.9 |
| Average | 83.5 | 63.7 | 64.8 | 69.3 | 85.6 | **87.5** |

## C  ALGORITHM DETAILS AND PSEUDOCODE

In this section, we first provide additional techniques applied to the TDP actor update. Then, we provide pseudocode for the training of TDP in Algorithm 1, along with loss functions per minibatch.

### C.1  ADDITIONAL ACTOR UPDATE TECHNIQUES

Although $L_\eta^*(\theta)$ in (12) with (16) can be used directly, it is more practical to reformulate the loss so that the constituent terms are more interpretable and the $\eta$-related hyperparameters are easier to handle. This can be achieved by expanding the expression, discarding constants, and rescaling (multiplying by $\eta/2$ and then substituting $\eta/2 \mapsto \eta$) (Fang et al., 2024). The resulting form of $L_\eta^*$ is

$$L_\eta^*(\theta) = \eta L_{\text{diff}}(\theta) + L_{\text{guide}}(\theta), \tag{48}$$

where $L_{\text{diff}}(\theta)$ is the diffusion loss in (5) and $L_{\text{guide}}(\theta)$ captures gradient alignment between the policy and critic, defined as

$$L_{\text{guide}}(\theta) := \mathbb{E}_{(s,a^0)\sim\mathcal{D}, k\sim U[1,K]} \left[ \sqrt{1 - \bar{\alpha}^k} \left\langle \nabla_{a^k} Q_\phi(s, a^0), \epsilon_\theta(a^k, k, s) \right\rangle \right], \tag{49}$$

with $\langle \cdot, \cdot \rangle$ denoting the inner product. The full TDP loss is then

$$L_{\text{TDP}}(\theta) = \eta L_{\text{diff}}(\theta) + L_{\text{guide}}(\theta) - \zeta L_{Q_\phi}(\theta). \tag{50}$$

We further introduce practical modifications to improve stability and simplify tuning. First, instead of fixing $\eta$ directly, we set a target diffusion divergence $\varepsilon_b^\eta$ between $\pi_\beta$ and $\pi_\eta^*$, which provides more intuitive control and often improves stability (He et al., 2023; Fang et al., 2024):

$$\eta_t = \eta_{t-1} + \alpha_\eta(L_{\text{diff}}(\theta) - \varepsilon_b^\eta), \tag{51}$$

where $\alpha_\eta$ is a learning rate and $\varepsilon_b^\eta$ the target divergence. This update is applied in some environments as needed.

Second, for convenience, we adopt a slightly modified actor loss:

$$L'_{\text{TDP}}(\theta) = \eta L_{\text{diff}}(\theta) + (1 - \lambda)L_{\text{guide}}(\theta) - \lambda L_{Q_\phi}(\theta), \tag{52}$$

where a balance coefficient $0 \leq \lambda \leq 1$ balances $L_{\text{guide}}$ and $L_{Q_\phi}$ with a single hyperparameter, tuned in place of $\zeta_t$.

### C.2  CRITIC UPDATE

For each sample $(s_b, a_b, r_b, s'_b, T_b)$ in the minibatch, we use the TD loss for each $Q_{\phi_n}$ ($n = 1, \cdots, N_q$) as the critic loss $L(\phi_n)$ defined as:

$$L(\phi_n) = \frac{1}{B} \sum_{b=1}^{B} \left( Q_{\phi_n}(s_b, a_b) - y_b \right)^2, \tag{53}$$

---

**Algorithm 1** Training of Two-Fold Improved Diffusion Policy (TDP)

---

**Hyperparameters:** Number of steps $M$, batch size $B$, target diffusion divergence value $\varepsilon_b^\eta$, LCB confidence interval $\rho$, balance coefficient $\lambda$, policy update interval $d$, number of Q ensembles $N_q$, number of actions for critic learning $N_a$, actor learning rate $\alpha_\theta$, critic learning rate $\alpha_\phi$, $\eta$ learning rate $\alpha_\eta$, EMA update rate $\tau$.

1: **Initialize:** Actor noise model $\epsilon_\theta$, target actor noise model $\epsilon_{\bar\theta} \leftarrow \epsilon_\theta$, ensemble critics $\{Q_\phi^n\}_{n=1}^{N_q}$, target ensemble critics $\{Q_{\bar\phi_n}\}_{n=1}^{N_q} \leftarrow \{Q_{\phi_n}\}_{n=1}^{N_q}$
2: **for** $m = 1$ **to** $M$ **do**
3:     Sample size $B$ minibatch $\{(s_b, a_b, r_b, s_b', T_b)\}_{b=1}^B$ from dataset $\mathcal{D}$
4:     Update each critic using TD loss with $a' \sim \pi_{\bar\theta}(\cdot|s')$: $\phi_n \leftarrow \phi_n - \alpha_\phi \nabla_{\phi_n} L(\phi_n)$ for $n = 1, \ldots, N_q$     (53)
5:     **if** $m \bmod d \equiv 0$ **then**
6:         Update actor using TDP loss: $\theta \leftarrow \theta - \alpha_\theta \nabla_\theta L(\theta)$     (56)
7:         $\eta$ update: $\eta \leftarrow \eta + \alpha_\eta(L_{\text{diff}}(\theta) - \varepsilon_b^\eta)$
8:     **end if**
9:     EMA target update: $\bar\theta \leftarrow \tau\bar\theta + (1-\tau)\theta$, $\bar\phi_n \leftarrow \tau\bar\phi_n + (1-\tau)\phi_n$ for $n = 1, \ldots, N_q$
10: **end for**

---

where $B$ is the minibatch size, $y_b = r_b + \gamma(1 - T_b) \cdot Q_{\text{targ},b}$ ($T_b = 1$ if terminal, $T_b = 0$ otherwise) and

$$Q_{\text{targ},b} = \frac{1}{N_q N_a} \sum_{n=1}^{N_q} \sum_{l=1}^{N_a} Q_{\bar\phi_n}(s_b', a_{l,b}') \text{ with } a_{l,b}' \sim \pi_{\bar\theta}(\cdot|s_b'), \tag{54}$$

for the ordinary case, and

$$Q_{\text{targ},b} = \frac{1}{N_q} \sum_{n=1}^{N_q} \max_l Q_{\bar\phi_n}(s_b', a_{l,b}') \text{ with } a_{l,b}' \sim \pi_{\bar\theta}(\cdot|s_b'), \tag{55}$$

for the max Q backup case (Kumar et al., 2020), typically used in environments like Antmaze.

C.3   ACTOR UPDATE

For each sample $(s_b, a_b, r_b, s_b', T_b)$ in the minibatch, we first sample noise steps $k_b \sim U[1, K]$ and compute the noised action as $a_b^{k_b} = \sqrt{\bar\alpha^k}a_b^0 + \sqrt{1 - \bar\alpha^k}\epsilon_b$, where $\epsilon_b \sim \mathcal{N}(0, I)$, for $b = 1, \ldots, B$. The actor loss $L(\theta)$ is defined as:

$$L(\theta) = \eta L_{\text{diff}}(\theta) + (1 - \lambda)L_{\text{guide}}(\theta) - \lambda L_{Q_\phi}(\theta). \tag{56}$$

where the diffusion loss $L_{\text{diff}}(\theta)$, the guidance loss $L_{\text{guide}}(\theta)$, and the Q value regularization term $L_{Q_\phi}(\theta)$ are defined as

$$L_{\text{diff}}(\theta) = \frac{1}{B} \sum_{b=1}^B \left\| \epsilon_\theta(a_b^{k_b}, k_b, s_b) - \epsilon_b \right\|^2, \tag{57}$$

$$L_{\text{guide}}(\theta) = \frac{1}{BC} \sum_{b=1}^B \sqrt{1 - \bar\alpha_b^{k_b}} \nabla_{a_b^{k_b}} Q_{\text{ref}}(s_b, a_b^0) \circ \epsilon_\theta(a_b^{k_b}, k_b, s_b), \tag{58}$$

$$L_{Q_\phi}(\theta) = \frac{1}{BC'N_q} \sum_{b=1}^B \sum_{n=1}^{N_q} Q_{\phi_n}(s_b, \hat a_b^0), \tag{59}$$

where $Q_{\text{ref}}(s, a) = \frac{1}{N_q} \sum_{n=1}^{N_q} Q_{\phi_n}(s, a)$ is the average of $N_q$ ensemble Q values, the reconstructed action $\hat a_b^0$ from $a_b^k$ is defined as

$$\hat a_b^0 = \frac{a_b^k - \sqrt{1 - \bar\alpha^k}\epsilon_\theta(a_b^k, k, s)}{\sqrt{\bar\alpha^k}}, \tag{60}$$

and the normalization constant $C, C'$ are defined as:

$$C = \frac{1}{BN_q} \sum_{b=1}^{B} \sum_{n=1}^{N_q} \left| Q_{\bar{\phi}_n}(s_b, a_b^0) \right|, \tag{61}$$

$$C' = \frac{1}{BN_q} \sum_{b=1}^{B} \sum_{n=1}^{N_q} \left| Q_{\bar{\phi}_n}(s_b, \hat{a}_b^0) \right|. \tag{62}$$

The gradient $\nabla_{a_b^{k_b}} Q_{\text{ref}}(s_b, a_b^0)$ in (58) is computed via the chain rule. When using only half of the batch for Q gradient and Q loss calculation, as mentioned in Sec. 3.4, the batch size for $L_{\text{guide}}(\theta)$, $L_{Q_\phi}(\theta)$, $C$, and $C'$ is reduced from $B$ to $B/2$. In this case, the indices are restricted to a randomly chosen subset $\{b_1, \ldots, b_{B/2}\} \subset \{1, \ldots, B\}$.

# D  PROOFS

## D.1  NOTATIONS AND DEFINITIONS

### D.1.1  $f$-DIVERGENCE

Consider the strictly convex function $f : [0, \infty) \to (-\infty, \infty]$ satisfying the following conditions:

*Condition 1.* $f(1) = 0$,

*Condition 2.* $f(0) = \lim_{x \to 0^+} f(x)$, and

*Condition 3.* $f(x) < \infty \; \forall x > 0$.

Let $p$ and $q$ be two probability distributions. Then the $f$-divergence between $p$ and $q$ (denoted by $D_f(p \parallel q)$) induced by $f$ is (Rényi, 1961):

$$D_f(p \parallel q) := \mathbb{E}_{x \sim q(x)} \left[ f \left( \frac{p(x)}{q(x)} \right) \right]. \tag{63}$$

For $f$-divergences, we have the following useful properties:

(i) For any $f$-divergence $D_f$ generated by $f$, $f$ is strictly convex with respect to any $p(x) \geq 0$, even with violation of $p$ being probability distribution.

(ii) For any valid (piecewise) differentiable $f$ for $f$-divergence, $(f')^{-1}$ is well-defined and strictly increasing because of strict convexity of $f$.

(iii) For any $f$-divergence $D_f$ generated by $f$, for $g(u) := u f(\frac{1}{u})$, $D_g$ generated by $g$ is also an $f$-divergence and satisfies the following (Amari, 2016):

$$D_f(p \parallel q) = D_g(q \parallel p). \tag{64}$$

We use the notation $D_f$ for Appendix D.3. For special cases, $D_f$ includes the forward KL divergence, the reverse KL divergence, or the diffusion divergence, each of which will be defined below. For $f(t) = t \log t$, we get KL divergence:

$$D_{\text{KL}}(p \parallel q) := \mathbb{E}_{x \sim p(x)} \left[ \log \frac{p(x)}{q(x)} \right] \tag{65}$$

(i) If $D(p \parallel q)$ refers to the forward KL divergence between $p$ and $q$ (denoted by $D_{\text{FKL}}(p, q)$), then

$$D(p \parallel q) = D_{\text{FKL}}(p \parallel q) := D_{\text{KL}}(p \parallel q) = \mathbb{E}_{x \sim p(x)} \left[ \log \frac{p(x)}{q(x)} \right] \tag{66}$$

(ii) If $D(p \parallel q)$ refers to the reverse KL divergence between $p$ and $q$ (denoted by $D_{\text{RKL}}(p \parallel q)$), then

$$D(p \parallel q) = D_{\text{RKL}}(p \parallel q) := D_{\text{KL}}(q \parallel p) = \mathbb{E}_{x \sim q(x)} \left[ \log \frac{q(x)}{p(x)} \right] \tag{67}$$

(iii) If $D(p \parallel q)$ refers to the diffusion divergence (Proposition D.1) between $p$ and $q$ (denoted by $D_{\text{diff}}(p \parallel q)$), then

$$D(p \parallel q) = D_{\text{diff}}(p \parallel q) := D_{\text{KL}}(p(x^{0:K}) \parallel q(x^{0:K})) \tag{68}$$

We define the diffusion divergence as an extension of the KL divergence, applied to the probability distributions of a sequence of variables $x^{0:K} := (x^0, x^1, \cdots, x^K)$ in a diffusion process with $K$ steps, rather than a single variable $x$.

### D.1.2 DIFFUSION DIVERGENCE

We define the *diffusion divergence* as the divergence induced by $L_{\text{diff}}^{\text{DDPM}}(\theta)$ in (28).

**Lemma D.1.** *Consider a diffusion model on $x^{0:K} = (x^0, \ldots, x^K)$ with $x^k = \sqrt{1 - \bar{\alpha}^k}x^0 + \sqrt{\bar{\alpha}^k}\epsilon$, $\epsilon \sim \mathcal{N}(0, I)$. Let $q(x^{0:K})$ be the forward distribution with initial $q(x^0)$, and $p_\theta(x^{0:K})$ the reverse denoising process parameterized by $\epsilon_\theta(x^k, k)$ with $p(x^K) = \mathcal{N}(0, I)$. The diffusion divergence $D_{diff}(q \parallel p_\theta)$, induced by the DDPM loss $L_{diff}^{DDPM}(\theta) = \mathbb{E}_{x^0 \sim q(x^0), k \sim U[1,K]}[w(k)||\epsilon_\theta(x^k, k) - \epsilon||^2]$, coincides with the KL divergence $D_{KL}(q(x^{0:K}) \parallel p_\theta(x^{0:K}))$ up to a constant.*

In other words, diffusion divergence is simply a **pathwise forward KL divergence** up to a constant. From this result, we can use the properties of the forward KL divergence in analyzing the diffusion divergence. Specifically, we may first establish inequalities using the forward KL divergence and subsequently apply marginalization. As a direct consequence, convexity follows immediately. Moreover, by the data processing inequality for the KL divergence, we obtain:

$$D_{\text{KL}}(q(x^0) \parallel p_\theta(x^0)) \leq D_{\text{diff}}(q \parallel p_\theta), \tag{69}$$

which we utilize in the proof of Theorem 4.5. Note that the loss weight simplification (Ho et al., 2020), i.e., $w(k) \equiv 1$ is applied, as mentioned in Appendix A.

Proof of Lemma D.1 is provided in Appendix D.2.

### D.1.3 Constrained Diffusion Policy Optimization (CDPO)

As mentioned in Sec. 3.1, CDPO is a generalized constrained policy optimization (CPO) (Achiam et al., 2017) with diffusion policy and the associated diffusion divergence. To analyze this connection in more detail, we first formalize the generalized CPO framework, with the primal form:

$$\pi^{(t+1)} = \arg\max_{\pi \in \Pi} \mathbb{E}_{s \sim \rho^{\pi_\beta}, a \sim \pi(\cdot|s)}[Q^{(\pi^{(t)})}(s,a)]: \tag{70}$$

$$\mathbb{E}_{s \sim \rho^{\pi_\beta}}[D(\pi_0(\cdot|s) \| \pi(\cdot|s))] \leq \varepsilon_b, \tag{71}$$

where $\pi^{(t)}$ is the policy at iteration $t$, $\Pi$ denotes the set of policies $\pi : \mathcal{S} \to P(\mathcal{A})$, $\rho^{\pi_\beta}$ is the state visitation distribution of the behavior policy $\pi_\beta$, $D$ is a divergence, $\pi_0 := \pi^{(0)}$ is the anchor policy, and $\varepsilon_b$ is a divergence threshold such that

$$0 < \varepsilon_b < \varepsilon_b^{\text{upper}} := \mathbb{E}_{s \sim \rho^{\pi_\beta}}[D(\pi_0(\cdot|s) \| \pi^*(\cdot|s))], \tag{72}$$

where $\pi^* := \arg\max_\pi \mathbb{E}_{s \sim \rho^{\pi_\beta}, a \sim \pi(\cdot|s)}[Q^{\pi^{(t)}}(s,a)]$. We assume $\rho^{\pi_\beta}(s) > 0$ for all $s \in \mathcal{S}$. This constraint ensures that the updated policy remains sufficiently close to the anchor policy $\pi_0$.

Our main concern is whether the dual form (7) is equivalent to the primal form, i.e., whether strong duality holds. When $D$ is an $f$-divergence, this holds because the objective is linear, the equality constraints are convex, and Slater's condition is satisfied by setting $\pi \equiv \pi_\beta$ (see Step 1 in Appendix D.3). In this case, (7) follows from the dual formulation after rescaling by a Lagrange multiplier, with $\zeta_t$ depending on $\varepsilon_b$, $\rho^{\pi_\beta}$, and $Q_\phi$.

Since the diffusion divergence can be expressed as a pathwise forward KL divergence (Lemma D.1), which is an $f$-divergence, CDPO inherits the same property. This holds whether $\pi$ is defined over the full path $(a^0, \ldots, a^K)$ or marginalized, since the integral operator is linear. Hence, strong duality is guaranteed for CDPO.

### D.1.4 Special Cases of CDPO Policies

As mentioned in Section 3.1, CDPO includes several algorithms and their corresponding solutions as special cases. For this, first recall (7):

$$\pi^{(t+1)} = \arg\min_\pi \{\mathbb{E}_{s \sim \rho^{\pi_\beta}}[D(\pi_0(\cdot|s) \| \pi(\cdot|s))] - \zeta_t L_{Q^{\pi^{(t)}}}(\pi)\} \tag{73}$$

For convenience, define $L(\pi; \pi_0, D, \zeta_t) := \mathbb{E}_{s \sim \rho^{\pi_\beta}}[D(\pi_0(\cdot|s) \| \pi(\cdot|s))] - \zeta_t L_{Q^{\pi^{(t)}}}(\pi)$. In what follows, we focus on four representative policies: the behavior policy $\pi_\beta$, the reverse KL constrained optimal policy and implicit method $\pi_\eta^*$, the explicit method $\pi_{\text{DQL}}$, and our proposed $\pi_{\text{TDP}}$, which are defined as follows.

1. $\pi_\beta$ denotes a behavior policy. Note that $\pi_\beta$ is a solution of CDPO with $\pi_0 = \pi_\beta$ and $\zeta_t = 0$.

2. We use two different definitions of $\pi_\eta^*$:

    (i) $\pi_\eta^*(a|s) = \pi_\beta(a|s) \cdot \exp\left(\frac{1}{\eta} Q^{\pi'}(s,a)\right)/Z(s)$ for some policy $\pi'$. $Z(s)$ is the partition function.

    (ii) $\pi_\eta^*$ denotes a set of $\pi_L^{(t)}$ with $L = L(\pi; \pi_\beta, D_{\text{RKL}}, \zeta_t)$ defined in (7).

3. $\pi_{\text{DQL}}$ denotes a set of $\pi_L^{(t)}$ with $L = L(\pi; \pi_\beta, D_{\text{diff}}, \zeta_t)$ defined in (7).
   Note that $\pi_{\text{DQL}}$ refers to the policy derived from DQL (Wang et al., 2023).

4. $\pi_{\text{TDP}}$ denotes a set of $\pi_L^{(t)}$ with $L = L(\pi; \pi_\eta^*, D_{\text{diff}}, \zeta_t)$ in (7). TDP refers to the proposed algorithm.

Note that the formulation of $\pi_\eta^*$ in 2-(ii) does not strictly correspond to CDPO; however, it is necessary for establishing the approximate relations in Appendix D.5. To support this, we prove the policy enhancement theorem (Appendix D.3) for the CPO setting defined above, assuming an $f$-divergence as the constraint measure.

### D.1.5 APPROXIMATE RELATION

**Definition D.2.** Let $\pi_1$ and $\pi_2$ be the policies within the CDPO formulation defined by Section D.1.3 with $L = L_1$ and $L = L_2$, respectively and $\pi_1^{(0)} = \pi_2^{(0)} = \pi_0$. Define the approximate relation denoted by $\gtrsim$ as follows: $\pi_1 \gtrsim \pi_2$ if

$$\mathbb{E}_{s \sim \rho^{\pi_\beta}, a \sim \pi_1^{(1)}(\cdot|s)}[Q^{\pi_0}(s, a)] \geq \mathbb{E}_{s \sim \rho^{\pi_\beta}, a \sim \pi_2^{(1)}(\cdot|s)}[Q^{\pi_0}(s, a)]. \tag{74}$$

where $J(\pi) := \mathbb{E}_{s_0 \sim d_0, \pi}[\sum_{t=0}^{\infty} \gamma^t R(s_t, a_t)]$ is the expected return of policy $\pi$.

Using Definition D.2, we consider the relation between two CDPO policies $\pi_1$ and $\pi_2$ sharing the same $\pi_0$ with different CPO losses $L = L_1$ and $L = L_2$, respectively. We assume that the leap of the *first* step is highly related to the subsequent steps. Therefore, we compare $\mathbb{E}_{s \sim \rho^{\pi_\beta}, a \sim \pi_1^{(1)}(\cdot|s)}[Q^{\pi_0}(s, a)]$ and $\mathbb{E}_{s \sim \rho^{\pi_\beta}, a \sim \pi_2^{(1)}(\cdot|s)}[Q^{\pi_0}(s, a)]$ to determine which policy approximately achieves a higher expected return.

Observe that this statement depends on $\zeta_t$. Therefore, if we want to say that $\pi_1 \gtrsim \pi_2$ with $L_1$ and $L_2$ without fixing certain $\zeta_t$, such as $\pi_{\text{DQL}}$ or $\pi_\eta^*$, we either confine $\zeta_t$ of $L_2$ or some other components and find $\zeta_t$ of $L_1$ which satisfy (74).

As a special case, when the CDPO policy $\pi$ is initialized with $\pi^{(0)} = \pi_0$, including cases where the loss is not diffusion-based, we have $\pi \gtrsim \pi_0$. This guarantee holds regardless of the choice of $\zeta_t$.

## D.2 PROOF OF LEMMA D.1

**Lemma D.1.** *Consider a diffusion model on $x^{0:K} = (x^0, \ldots, x^K)$ with $x^k = \sqrt{1 - \bar{\alpha}^k}x^0 + \sqrt{\bar{\alpha}^k}\epsilon$, $\epsilon \sim \mathcal{N}(0, I)$. Let $q(x^{0:K})$ be the forward distribution with initial $q(x^0)$, and $p_\theta(x^{0:K})$ the reverse denoising process parameterized by $\epsilon_\theta(x^k, k)$ with $p(x^K) = \mathcal{N}(0, I)$. The diffusion divergence $D_{diff}(q \parallel p_\theta)$, induced by the DDPM loss $L_{diff}^{DDPM}(\theta) = \mathbb{E}_{x^0 \sim q(x^0), k \sim U[1,K]}[w(k)||\epsilon_\theta(x^k, k) - \epsilon||^2]$, coincides with the KL divergence $D_{KL}(q(x^{0:K}) \parallel p_\theta(x^{0:K}))$ up to a constant.*

*Proof.* Our goal is to show that for $D_{\text{diff}}(q \parallel p_\theta) := D_{\text{KL}}(q(x^{0:K}) \parallel p_\theta(x^{0:K}))$, there exists a proper choice of $w(k)$ and a constant $C$ independent of $p_\theta$ such that the following holds:

$$\mathbb{E}_{x^0 \sim q(x^0), k \sim U[1,K]}[w(k)||\epsilon - \epsilon_\theta(x^k, k)||^2] + C = D_{\text{diff}}(q \parallel p_\theta) = D_{\text{KL}}(q(x^{0:K}) \parallel p_\theta(x^{0:K})). \tag{75}$$

First, observe that the following identity holds:

$$D_{\text{diff}}(q \parallel p_\theta) := D_{\text{KL}}(q(x^{0:K}) \parallel p_\theta(x^{0:K})) = \mathbb{E}_{x^{0:K} \sim q(x^{0:K})}\left[-\log \frac{p_\theta(x^{0:K})}{q(x^{0:K})}\right] \tag{76}$$

$$= \mathbb{E}_{x^{0:K} \sim q(x^{0:K})}\left[-\log \frac{p_\theta(x^{0:K})}{q(x^{1:K}|x^0)} + \log q(x^0)\right] \tag{77}$$

$$= \mathbb{E}_{x^{0:K} \sim q(x^{0:K})}\left[-\log \frac{p_\theta(x^{0:K})}{q(x^{1:K}|x^0)}\right] - H(q(x^0)). \tag{78}$$

The second term is a constant, and according to the DDPM paper (Ho et al., 2020), the first term is given by:

$$\mathbb{E}_{x^{0:K} \sim q(x^{0:K})}\left[D_{\text{KL}}(q(x^K|x^0) \parallel p(x^K)) + \sum_{k=2}^{K} D_{\text{KL}}(q(x^{k-1}|x^k, x^0) \parallel p_\theta(x^{k-1}|x^k)) - \log p_\theta(x^0|x^1)\right]. \tag{79}$$

Since $q(x^K|x^0)$, $p(x^K)$, and $q(x^{k-1}|x^k, x^0)$ ($k = 2, \cdots, K$) are all Gaussian, and $p_\theta(x^{k-1}|x^k)$ is also Gaussian with a mean parameterized by the noise model $\epsilon_\theta(x^k, k)$ as:

$$p_\theta(x^{k-1}|x^k) = \mathcal{N}\left(\frac{1}{\sqrt{\alpha^k}}\left(x^k - \frac{1-\alpha^k}{\sqrt{1-\bar{\alpha}^k}}\epsilon_\theta(x^k, k)\right), (\sigma^k)^2 I\right). \tag{80}$$

Therefore, we conclude that the expression in (79) coincides with the KL divergence $D_{\text{KL}}(q(x^{0:K}) \parallel p(x^{0:K}))$ up to an additive constant, with $w(k) = \frac{(1-\alpha^k)^2}{2(\sigma^k)^2\alpha^k(1-\bar{\alpha}^k)}$, following the derivation in the DDPM paper (Ho et al., 2020). $\qquad\square$

### D.3 Proof of Theorem 4.1

**Theorem 4.1.** *For CDPO, we can achieve **policy improvement** for the first iteration step, i.e.,*

$$J(\pi^{(1)}) \geq J(\pi^{(0)}) = J(\pi_0), \tag{81}$$

*where $J(\pi)$ is the expected return of policy $\pi$. Furthermore, we can achieve **expected Q value improvement** for all the other iteration steps, i.e. for $t \geq 1$,*

$$\mathbb{E}_{s \sim \rho^{\pi_\beta}, a \sim \pi^{(t+1)}}[Q^{\pi^{(t)}}(s,a)] \geq \mathbb{E}_{s \sim \rho^{\pi_\beta}, a \sim \pi^{(t)}}[Q^{\pi^{(t)}}(s,a)]. \tag{82}$$

We first prove this for generalized CPO with diffusion divergence or $f$-divergences for Corollary 4.2 (Steps 1-3), and then smoothly extend to CDPO cases (Step 4).

*Proof of (81).* For simplicity, we prove (81) in the tabular setting, where $\mathcal{S} = \{s_1, \ldots, s_m\}$ and $\mathcal{A} = \{a_1, \ldots, a_n\}$. Nonetheless, the proof can be readily extended to continuous control settings. We assume $\rho^{\pi_\beta}(s) > 0$ for all $s \in \mathcal{S}$, which is standard in continuous control and can be generalized to cases involving the *support* of $\rho^{\pi_\beta}$. The result is proven for general $f$-divergences as needed in Appendix D.5. Our argument is based on the constrained form of CDPO (or generalized CPO with $f$-divergences) described in Appendix D.1.3, which is equivalent to its unconstrained version by the transformation discussed in Appendix D.1.3.

*Proof of (82).* We first prove this for generiazed CPO with $f$-divergences.

***(Step 1.)*** First, we verify whether strong duality holds for the hard formulation of generalized CPO introduced in Appendix D.1.3. To this end, we restate the hard formulation of generalized CPO in the tabular case, explicitly incorporating the normalization constraints. We also express the divergence as $D_g(\pi(\cdot|s) \parallel \pi_0(\cdot|s))$ instead of $D_f(\pi_0(\cdot|s) \parallel \pi(\cdot|s))$ by applying the transformation $g(u) := uf(\frac{1}{u})$.

$$\pi^{(t+1)} = \arg\max_{\pi \geq 0} \sum_{i=1}^{m} \rho^{\pi_\beta}(s_i) \sum_{j=1}^{n} \pi(a_j|s_i) Q^{(t)}(s_i, a_j) : \tag{83}$$

$$\sum_{i=1}^{m} \rho^{\pi_\beta}(s_i) D_g(\pi(\cdot|s_i) \parallel \pi_0(\cdot|s_i)) \leq \varepsilon_b, \tag{84}$$

$$\sum_{j=1}^{n} \pi(a_j|s_i) = 1 \ \forall 1 \leq i \leq m. \tag{85}$$

Since $\{\pi : \pi \geq 0\}$ is a convex set, the objective is linear, for each $\pi(a|s)$, the inequality constraint is convex as mentioned in Appendix D.1.1, and the equality constraint is affine, we conclude that the optimization problem is convex. Also, by using $D_g(\pi_0(\cdot|s_i) \parallel \pi_0(\cdot|s_i)) = 0 \ \forall s_i \in \mathcal{S}$ and that $\pi_0 \geq 0$, we get $\sum_{i=1}^{m} \rho^{\pi_\beta}(s_i) D_g(\pi_0(\cdot|s_i) \parallel \pi_0(\cdot|s_i)) = 0 < \varepsilon_b$, satisfying Slater's condition. Therefore, the KKT condition holds, and the strong duality holds.

***(Step 2.)*** Next, we aim to express $\pi^{(t+1)}$ in terms of $\pi_0$ and $Q^{(t)}$ for each $(s_i, a_j)$. We write the dual problem for timestep $t$:

$$\min_{\mu_t \geq 0, \nu_t(s)} \max_{\pi \geq 0} \mathcal{L}(\pi, \mu_t, \nu_t(s)) := \sum_{i=1}^{m} \rho^{\pi_\beta}(s_i) \sum_{j=1}^{n} \pi(a_j|s_i) Q^{(t)}(s_i, a_j)$$

$$+ \mu_t \left( \varepsilon_b - \sum_{i=1}^{m} \rho^{\pi_\beta}(s_i) D_g(\pi(\cdot|s_i) \parallel \pi_0(\cdot|s_i)) \right) + \sum_{i=1}^{m} \nu_t(s_i) \left( 1 - \sum_{j=1}^{n} \pi(a_j|s_i) \right). \tag{86}$$

By the strong duality, we can swap the order of min and max operators:

$$\min_{\mu_t \geq 0, \nu_t(s)} \max_{\pi \geq 0} \mathcal{L}(\pi, \mu_t, \nu_t(s)) = \max_{\pi \geq 0} \min_{\mu_t \geq 0, \nu_t(s)} \mathcal{L}(\pi, \mu_t, \nu_t(s)) = \max_{\pi \geq 0} \mathcal{L}(\pi, \mu_t^*, \nu_t^*(s)), \tag{87}$$

where $(\mu_t^*, \nu_t^*(s))$ are dual optimal points. Note that by Lemma D.3 (which will be stated and proved later), $\mu_t^* > 0$. By taking the partial derivative of $\mathcal{L}(\pi, \mu_t^*, \nu_t^*(s))$ with respect to $\pi(a_j|s_i)$ for each $(s_i, a_j)$, we get the following: for each $1 \le i \le m, 1 \le j \le n$,

$$\left.\frac{\partial \mathcal{L}(\pi, \mu_t^*, \nu_t^*(s))}{\partial \pi(a_j|s_i)}\right|_{\pi \equiv \pi^{(t+1)}} = \rho^{\pi_\beta}(s_i)\left(Q^{(t)}(s_i, a_j) - \mu_t^* \cdot \left.\frac{\partial D_g(\pi_0(\cdot|s_i)\|\pi(\cdot|s_i))}{\partial \pi(a_j|s_i)}\right|_{\pi \equiv \pi^{(t+1)}}\right) - \nu_t^*(s_i). \tag{88}$$

By the strong duality, for all $1 \le i \le m, 1 \le j \le n$, we have either $(i)$ (88) is zero or $(ii)$ $\pi^{(t+1)}(a_j|s_i) = 0$ if (88) is zero at $\pi^{(t+1)}(a_j|s_i) < 0$. By the definition of $f$-divergence, $\rho^{\pi_\beta}(s_i) > 0$ $\forall s_i \in \mathcal{S}$, and $\mu_t^* > 0$, we rearrange (88):

$$g'\left(\frac{\pi^{(t+1)}(a_j|s_i)}{\pi_0(a_j|s_i)}\right) = \left.\frac{\partial D_g(\pi(\cdot|s_i)\|\pi_0(\cdot|s_i))}{\partial \pi(a_j|s_i)}\right|_{\pi \equiv \pi^{(t+1)}} = \frac{Q^{(t)}(s_i, a_j) - \nu_t^*(s_i)/\rho^{\pi_\beta}(s_i)}{\mu_t^*} \tag{89}$$

or $\pi^{(t+1)}(a_j|s_i) = 0$ if (88) is zero at $\pi^{(t+1)}(a_j|s_i) < 0$. Thus, we can express $\pi^{(t+1)}(a_j|s_i)$ for each $(s_i, a_j)$ as:

$$\pi^{(t+1)}(a_j|s_i) = \pi_0(a_j|s_i) \cdot \max\left\{(g')^{-1}\left(\frac{Q^{(t)}(s_i, a_j) - \nu_t^*(s_i)/\rho^{\pi_\beta}(s_i)}{\mu_t^*}\right), 0\right\}, \tag{90}$$

Note that $g'$ is strictly increasing, and so is $(g')^{-1}$.

*(Step 3.)* Finally, we are ready to prove (81) for generalized CPO with $f$-divergence. Let

$$\hat{Q}^{(t)}(s_i, a_j) := \frac{Q^{(t)}(s_i, a_j) - \nu_t^*(s_i)/\rho^{\pi_\beta}(s_i)}{\mu_t^*} - g'(1) \tag{91}$$

Since $\mu_t^*(s_i), \rho^{\pi_\beta}(s_i), \mu_t^*$, and $g'(1)$ do not depend on actions $a_j$, we have

$$\mathbb{E}_{a \sim \pi^{(t+1)}(\cdot|s_i)}[Q^{\pi^{(t)}}(s_i, a)] \ge \mathbb{E}_{a \sim \pi_0(\cdot|s_i)}[Q^{\pi^{(t)}}(s_i, a)] \quad \forall s_i \in \mathcal{S}. \tag{92}$$

$$\iff \sum_{j=1}^n \pi^{(t+1)}(a_j|s_i)Q^{(t)}(s_i, a_j) \ge \sum_{j=1}^n \pi_0(a_j|s_i)Q^{(t)}(s_i, a_j) \quad \forall s_i \in \mathcal{S}. \tag{93}$$

$$\iff \sum_{j=1}^n \pi^{(t+1)}(a_j|s_i)\hat{Q}^{(t)}(s_i, a_j) \ge \sum_{j=1}^n \pi_0(a_j|s_i)\hat{Q}^{(t)}(s_i, a_j) \quad \forall s_i \in \mathcal{S}. \tag{94}$$

where $A \iff B$ denotes "$A$ if and only if $B$". For each fixed $s_i \in \mathcal{S}$ and $a_j \in \mathcal{A}$, $\pi^{(t+1)}(a_j|s_i)$ is reweighted as follows:

Case 1. for $a_j \in \mathcal{A}$ with $\hat{Q}^{(t)}(s_i, a_j) = 0$, we have $\pi^{(t+1)}(a_j|s_i) = \pi_0(a_j|s_i)$ by (90-91).

Case 2. for $a_j \in \mathcal{A}$ with $\hat{Q}^{(t)}(s_i, a_j) > 0$, we have $\pi^{(t+1)}(a_j|s_i) > \pi_0(a_j|s_i)$ by (90-91) and strictly increasing $g'$.

Case 3. for $a_j \in \mathcal{A}$ with $\hat{Q}^{(t)}(s_i, a_j) < 0$, we have $\pi^{(t+1)}(a_j|s_i) < \pi_0(a_j|s_i)$ by (90-91) and strictly increasing $g'$.

Therefore, (92-94) holds.

By plugging $t = 0$ and $\pi_0 = \pi^{(0)}$, we conclude:

$$\mathbb{E}_{a \sim \pi^{(1)}(\cdot|s_i)}[Q^{\pi^{(0)}}(s_i, a)] \ge \mathbb{E}_{a \sim \pi^{(0)}(\cdot|s_i)}[Q^{\pi^{(0)}}(s_i, a)] \quad \forall s_i \in \mathcal{S}, \tag{95}$$

satisfying the policy improvement theorem (Sutton & Barto, 2018).

*(Step 4.)* To extend this to the diffusion divergence, we can apply this with respect to the paths, namely $\pi(a^{0:K}|s)$ and $\pi_0(a^{0:K}|s)$, instead of $\pi(a|s)$ (i.e., $\pi(a^0|s)$) and $\pi_0(a|s)$ (i.e., $\pi_0(a^0|s)$). Following the same argument, we arrive at the same conclusion by marginalizing over the paths, using the relation

$$\int \cdots \int \pi(a^{0:K}|s)\, da^{1:K} = \pi(a^0|s) \int \cdots \int \pi(a^{1:K} \mid a^0, s)\, da^{1:K} = \pi(a^0|s).$$

$\square$

Note that for forward and reverse KL divergences (and thus diffusion divergence), $\pi^{(t+1)}(a|s) = 0$ never happens because $g'(t) = -\frac{1}{t}$ and $1 + \log t$, respectively, where $\lim_{t \to 0+} g'(t) = -\infty$.

*Proof of (82).* Since $\pi^{(t+1)} = \arg\max_{\pi \in \Pi} \mathbb{E}_{s \sim \rho^{\pi_\beta}, a \sim \pi(\cdot|s)}[Q^{(t)}(s, a)]$, (82) holds if the constrained variant of CDPO in Section D.1.3 is feasible for all $t$. (81) implies that the problem is feasible at $t = 0$. Also, if the problem is feasible at $t$, then it is also feasible at $t + 1$ because $\pi^{(t+1)} = \pi^{(t)}$ is feasible. Thus, we conclude that (82) holds for all $t$. $\square$

**Lemma D.3.** *For (87), $\mu_t^* > 0$.*

*Proof.* It is given that $\mu_t^* \geq 0$, and we claim that $\mu_t^* \neq 0$. Suppose $\mu_t^* = 0$. We already showed that Slater's condition holds. Let $\tilde{\pi}^{(t+1)}$ be a point that satisfies Slater's condition. Then, $\tilde{\pi}^{(t+1)} \in \Pi$ and the dual problem (87) becomes

$$\pi^{(t+1)} = \arg\max_{\pi \in \Pi} \mathcal{L}(\pi, 0, \nu_t^*(s)) = \arg\max_{\pi \in \Pi} \mathbb{E}_{s \sim \rho^{\pi_\beta}, a \sim \pi(\cdot|s)}[Q^{(t)}(s, a)]. \tag{96}$$

Therefore, $\pi^{(t+1)} = \pi^*$ by definition, and $\mathbb{E}_{s \sim \rho^{\pi_\beta}}[D_g(\pi_0(\cdot|s) \| \pi^*(\cdot|s))] \leq \varepsilon_b$. However, in (72), we assumed that $\varepsilon_b < \varepsilon_b^{upper} := \mathbb{E}_{s \sim \rho^{\pi_\beta}}[D_g(\pi_0(\cdot|s) \| \pi^*(\cdot|s))]$, which is a contradiction. $\square$

### D.4 Proof of Corollary 4.2

**Corollary 4.2.** *Let $\pi_1 \gtrsim \pi_2$ denote that policy $\pi_1$ is is approximately better than policy $\pi_2$ in the sense of (19) and (20). Then,*

1) $\pi_{DQL} \gtrsim \pi_\beta$,

2) $\pi_\eta^* \gtrsim \pi_\beta$, *and*

3) $\pi_{TDP} \gtrsim \pi_\eta^*$.

*Proof.* Observe that in all three cases, the latter policy coincides with the anchor policy $\pi_0$ of the preceding one, thereby the approximate relationship holds in each case. $\square$

## D.5 PROOF OF COROLLARY 4.3

**Corollary 4.3.** $\pi_\eta^*$ *is approximately better than* $\pi_{DQL}$ *in the following sense. For* $\pi_\beta$ *satisfying some mild conditions, for any* $\zeta_t$ *there exists properly chosen* $\eta$ *satisfying*

$$\mathbb{E}_{s \sim \rho^{\pi_\beta}, a \sim \pi_\eta^*}[Q^{\pi_\beta}(s, a)] \geq \mathbb{E}_{s \sim \rho^{\pi_\beta}, a \sim \pi_{DQL}^{(1)}}[Q^{\pi_\beta}(s, a)] \tag{97}$$

We first define the terms and conditions necessary for the proof. Assume that $Q^{\pi_\beta}(s, a)$ is bounded $\forall s \in \mathcal{S}, a \in \mathcal{A}$. For states $s$ with a normalization factor $N(s) := \sup_{a' \in \mathcal{A}} Q^{\pi_\beta}(s, a') - V^{\pi_\beta}(s) > 0$, we first define the normalized advantage $\bar{A}^{\pi_\beta}(s, a)$ as follows:

$$\bar{A}^{\pi_\beta}(s, a) := \frac{Q^{\pi_\beta}(s, a) - V^{\pi_\beta}(s)}{N(s)}, \tag{98}$$

where $\bar{A}^{\pi_\beta}(s, a) \leq 1$ with equality when $Q^{\pi_\beta}(s, a) = \sup_{a' \in \mathcal{A}} Q^{\pi_\beta}(s, a')$. For states $s$ with $N(s) = 0$, we regard that all actions achieve $\bar{A}^{\pi_\beta}(s, a) = 1$ for generality.

We also define the cumulative distribution function (CDF) $F_{\bar{A}}(r|\pi, s)$ and the probability density function (PDF) $f_{\bar{A}}(r|\pi, s)$ of $\bar{A}^{\pi_\beta}(s, a)$ with respect to policy $\pi$ as

$$F_{\bar{A}}(r|\pi, s) := \mathbb{P}_{a \sim \pi(\cdot|s)}[\bar{A}^{\pi_\beta}(s, a) \leq r], \tag{99}$$

$$f_{\bar{A}}(r|\pi, s) := \frac{\partial}{\partial r} F_{\bar{A}}(r|\pi, s). \tag{100}$$

For typical suboptimal behavior policy $\pi_\beta$, since the policy struggles to select the best action, $f_{\bar{A}}(r|\pi_\beta, s)$ would be small for $r$ close to 1, and even with a large decay. In this line, we define a set of such states, $\mathcal{S}(c, n, t)$, as:

$$\mathcal{S}(c, n, t) := \Big\{ s \in \mathcal{S} : f_{\bar{A}}(r|\pi_\beta, s) \leq c(1 - r^n) \,\forall t \leq r < 1$$
$$\text{and } \pi_\beta(a|s) = 0 \,\forall a \text{ s.t. } Q^{\pi_\beta}(s, a) = \sup_{a' \in \mathcal{A}} Q^{\pi_\beta}(s, a') \Big\} \tag{101}$$

for $c > 0$, $n \in \mathbb{N}$, $0 \leq t < 1$ with $cn(1 - t) < 1$. Note that for $s$ with $N(s) = 0$, $f_{\bar{A}}(r|\pi_\beta, s) = \delta(r - 1)$, where $\delta(\cdot)$ is a Dirac delta function, by our formulation, we conclude that such $s$ cannot belong to $\mathcal{S}(c, n, t)$.

For the proof, we have the following assumptions on $\pi_\beta$:

Assumption 1. $Q^{\pi_\beta}(s, a)$ is bounded.

Assumption 2. $\pi_\beta(a|s) > 0 \,\forall s \in \mathcal{S}, \forall a \in \mathcal{A} : Q^{\pi_\beta}(s, a) < \sup_{a' \in \mathcal{A}} Q^{\pi_\beta}(s, a')$

Assumption 3. $\int_{\mathcal{S}(c,n,t)} \rho^{\pi_\beta}(s) N(s) ds > 0$ for some $c > 0$, $n \in \mathbb{N}$, $0 \leq t < 1$ with $cn(1-t) < 1$.

Assumption 4. for each $s$ with $N(s) > 0$, there exists $l(s) < 1$ such that $f_{\bar{A}}(r|\pi_\beta, s) > 0$ for all $l(s) \leq r < 1$.

*proof.* Let $M(\eta)$ and $M$ be defined by

$$M(\eta) := \mathbb{E}_{s\sim\rho^{\pi_\beta}, a\sim\pi_\eta^*(\cdot|s)}[Q^{\pi_\beta}(s,a)], \tag{102}$$

$$M := \mathbb{E}_{s\sim\rho^{\pi_\beta}}[\sup_{a'\in\mathcal{A}} Q^{\pi_\beta}(s,a')]. \tag{103}$$

Note that $M(\eta) \le M$ for $\eta > 0$. Also, since $\pi_\eta^*(a|s) = \pi_\beta(a|s)\frac{\exp(\frac{1}{\eta}Q^{\pi_\beta}(s,a))}{Z(s)}$ and by Assumptions 2 and 4,

$$\lim_{\eta\to 0+} \mathbb{E}_{s\sim\rho^{\pi_\beta}, a\sim\pi_\eta^*(\cdot|s)}[Q^{\pi_\beta}(s,a)] = \lim_{\eta\to 0+} M(\eta) = M. \tag{104}$$

We aim to show that $M - \mathbb{E}_{s\sim\rho^{\pi_\beta}, a\sim\pi_{\text{DQL}}^{(1)}(\cdot|s)}[Q^{\pi_\beta}(s,a)] > 0$.

By definition of $\bar{A}^{\pi_\beta}(s,a)$, for $s$ with $N(s) > 0$, we have $1 - \mathbb{E}_{a\sim\pi(\cdot|s)}[\bar{A}^{\pi_\beta}(s,a)] \ge 0$ and

$$1 - \mathbb{E}_{a\sim\pi(\cdot|s)}[\bar{A}^{\pi_\beta}(s,a)] = \frac{\sup_{a'\in\mathcal{A}} Q^{\pi_\beta}(s,a') - V^{\pi_\beta}(s)}{N(s)} - \frac{\mathbb{E}_{a\sim\pi(\cdot|s)}[Q^{\pi_\beta}(s,a)] - V^{\pi_\beta}(s)}{N(s)} \tag{105}$$

$$= \frac{\sup_{a'\in\mathcal{A}} Q^{\pi_\beta}(s,a') - \mathbb{E}_{a\sim\pi(\cdot|s)}[Q^{\pi_\beta}(s,a)]}{N(s)}. \tag{106}$$

By the fact that $f_{\bar{A}}(r|\pi,s)$ is a probability density function, we also have

$$1 - \mathbb{E}_{a\sim\pi(\cdot|s)}[\bar{A}^{\pi_\beta}(s,a)] = \int_{-\infty}^{1} f_{\bar{A}}(r|\pi,s)dr - \int_{-\infty}^{1} r f_{\bar{A}}(r|\pi,s)dr = \int_{-\infty}^{1} (1-r)f_{\bar{A}}(r|\pi,s)dr. \tag{107}$$

By (106) and (107), we have

$$\sup_{a'\in\mathcal{A}} Q^{\pi_\beta}(s,a') - \mathbb{E}_{a\sim\pi(\cdot|s)}[Q^{\pi_\beta}(s,a)] = N(s)\int_{-\infty}^{1}(1-r)f_{\bar{A}}(r|\pi,s)dr \ge 0. \tag{108}$$

Note that (108) also holds for $s$ with $N(s) = 0$. Then, taking the expectations with respect to $s \sim \rho^{\pi_\beta}$ on (108) yields

$$M - \mathbb{E}_{s\sim\rho^{\pi_\beta}, a\sim\pi(\cdot|s)}[Q^{\pi_\beta}(s,a)] = \int_{\mathcal{S}} \rho^{\pi_\beta}(s)N(s)\int_{-\infty}^{1}(1-r)f_{\bar{A}}(r|\pi,s)drds \ge 0. \tag{109}$$

Therefore, by Lemma D.4 (which will be stated and proved later), (109) with $\pi = \pi_{\text{DQL}}^{(1)}$ becomes

$$M - \mathbb{E}_{s\sim\rho^{\pi_\beta}, a\sim\pi_{\text{DQL}}^{(1)}(\cdot|s)}[Q^{\pi_\beta}(s,a)] \tag{110}$$

$$= \int_{\mathcal{S}} \rho^{\pi_\beta}(s)N(s)\int_{-\infty}^{1}(1-r)f_{\bar{A}}(r|\pi_{\text{DQL}}^{(1)},s)drds \qquad \text{(by (109))} \tag{111}$$

$$\ge \int_{\mathcal{S}(c,n,t)} \rho^{\pi_\beta}(s)N(s)\int_{-\infty}^{1}(1-r)f_{\bar{A}}(r|\pi_{\text{DQL}}^{(1)},s)drds \qquad (\mathcal{S} \supseteq \mathcal{S}(c,n,t)) \tag{112}$$

$$\ge (1-t)\int_{\mathcal{S}(c,n,t)} \rho^{\pi_\beta}(s)N(s)F_{\bar{A}}(t|\pi_{\text{DQL}}^{(1)},s)ds \tag{113}$$

$$\ge (1-t)(1-cn(1-t))\int_{\mathcal{S}(c,n,t)} \rho^{\pi_\beta}(s)N(s)ds \qquad \text{(by Lemma D.4)} \tag{114}$$

$$> 0. \qquad \text{(by Assumption 3)} \tag{115}$$

Since $M > \mathbb{E}_{s\sim\rho^{\pi_\beta}, a\sim\pi_{\text{DQL}}^{(1)}(\cdot|s)}[Q^{\pi_\beta}(s,a)]$, there exists $\eta > 0$ that satisfies $M(\eta) \ge \mathbb{E}_{s\sim\rho^{\pi_\beta}, a\sim\pi_{\text{DQL}}^{(1)}(\cdot|s)}[Q^{\pi_\beta}(s,a)]$. $\qquad\square$

**Lemma D.4.** *Assuming four assumptions in the main proof, $\forall s \in \mathcal{S}(c, n, t)$ the following holds:*

$$F_{\bar{A}}(t|\pi_{DQL}^{(1)}, s) \geq 1 - cn(1 - t) > 0. \tag{116}$$

*Proof.* We start from (90) in Section D.3 (the proof of Theorem 4.1). DQL uses diffusion divergence $D_{\text{diff}}(q \parallel p_\theta) := D_{\text{FKL}}(q(x^{0:K}) \parallel p_\theta(x^{0:K}))$ and for the forward KL divergence, $(g')^{-1}(u) = -\frac{1}{u}$. As in the proof of Theorem 4.1, we assume that $\rho^{\pi_\beta}(s) > 0 \; \forall s \in \mathcal{S}$. Then we get $\pi_{\text{DQL}}^{(1)}(a^{0:K}|s)$ for each $s \in \mathcal{S}, a \in \mathcal{A}$ as

$$\pi_{\text{DQL}}^{(1)}(a^{0:K}|s) = \pi_\beta(a^{0:K}|s) \frac{\mu_t^*}{\nu_t^*(s)/\rho^{\pi_\beta}(s) - Q^{\pi_\beta}(s, a^0)}. \tag{117}$$

By integrating both sides over $a^{1:K}$, and taking $a^0 = a$, we get

$$\pi_{\text{DQL}}^{(1)}(a|s) = \pi_\beta(a|s) \frac{\mu_t^*}{\nu_t^*(s)/\rho^{\pi_\beta}(s) - Q^{\pi_\beta}(s, a)}. \tag{118}$$

*(Step 1.)* We will show that for $s \in \mathcal{S}$ with $N(s) := \sup_{a' \in \mathcal{A}} Q^{\pi_\beta}(s, a') - V^{\pi_\beta}(s) > 0$, $a \in \mathcal{A}$, and $0 \leq r < 1$ with $r = \bar{A}^{\pi_\beta}(s, a)$:

$$\nu_t^*(s)/\rho^{\pi_\beta}(s) - Q^{\pi_\beta}(s, a) \geq (1 - r)\mu_t^*. \tag{119}$$

Note that $\pi_{\text{DQL}}^{(1)}$ in (118) is well-defined, i.e., $\pi_{\text{DQL}}^{(1)}(a|s) \geq 0$ for all $s \in \mathcal{S}, a \in \mathcal{A}$. By Lemma D.3 $\mu_t^* > 0$. Therefore, the denominator must be positive, i.e.,

$$\nu_t^*(s)/\rho^{\pi_\beta}(s) - Q^{\pi_\beta}(s, a) > 0 \text{ for all } s \in \mathcal{S}, a \in \mathcal{A}. \tag{120}$$

Next, by integrating both sides of (118) with respect to $a \in \mathcal{A}$,

$$1 = \int_{\mathcal{A}} \pi_{\text{DQL}}^{(1)}(a|s)da = \int_{\mathcal{A}} \pi_\beta(a|s) \frac{\mu_t^*}{\nu_t^*(s)/\rho^{\pi_\beta}(s) - Q^{\pi_\beta}(s, a)} da = \mathbb{E}_{a \sim \pi_\beta(\cdot|s)} \left[ \frac{\mu_t^*}{\nu_t^*(s)/\rho^{\pi_\beta}(s) - Q^{\pi_\beta}(s, a)} \right] \tag{121}$$

Since $\mu_t^* > 0$ is a constant, this implies that

$$(\mu_t^*)^{-1} = \mathbb{E}_{a \sim \pi_\beta(\cdot|s)}[(\nu_t^*(s)/\rho^{\pi_\beta}(s) - Q^{\pi_\beta}(s, a))^{-1}] \tag{122}$$

$$\geq \left( \mathbb{E}_{a \sim \pi_\beta(\cdot|s)}[(\nu_t^*(s)/\rho^{\pi_\beta}(s) - Q^{\pi_\beta}(s, a))] \right)^{-1} \tag{123}$$

$$= (\nu_t^*(s)/\rho^{\pi_\beta}(s) - V^{\pi_\beta}(s))^{-1} \tag{124}$$

where the inequality holds by Jensen's inequality $\mathbb{E}[f(x)] \geq f(\mathbb{E}[x])$ for a convex function $f(x) = x^{-1}$ on $x > 0$, which is valid due to (120). Therefore,

$$\nu_t^*(s)/\rho^{\pi_\beta}(s) - V^{\pi_\beta}(s) \geq \mu_t^*. \tag{125}$$

Also, due to (120), $\nu_t^*(s)/\rho^{\pi_\beta}(s) - \sup_{a' \in \mathcal{A}} Q^{\pi_\beta}(s, a') \geq 0$, hence

$$N(s) := \sup_{a' \in \mathcal{A}} Q^{\pi_\beta}(s, a') - V^{\pi_\beta}(s) \leq \nu_t^*(s)/\rho^{\pi_\beta}(s) - V^{\pi_\beta}(s) \tag{126}$$

Therefore, for $(s, a)$ with $\bar{A}^{\pi_\beta}(s, a) = r$, we have $Q^{\pi_\beta}(s, a) = V^{\pi_\beta}(s, a) + rN(s)$ and:

$$\nu_t^*(s)/\rho^{\pi_\beta}(s) - Q^{\pi_\beta}(s, a) = (\nu_t^*(s)/\rho^{\pi_\beta}(s) - V^{\pi_\beta}(s)) - rN(s) \tag{127}$$

$$\geq (1 - r)(\nu_t^*(s)/\rho^{\pi_\beta}(s) - V^{\pi_\beta}(s)) \quad \text{(by (125))} \tag{128}$$

$$\geq (1 - r)\mu_t^*. \quad \text{(by (126))} \tag{129}$$

*(Step 2.)* Plugging (119) into (118), we get

$$\pi_{\text{DQL}}^{(1)}(a|s) \leq \frac{1}{1-r}\pi_\beta(a|s). \tag{130}$$

Note that for $s$ with $N(s) = 0$, (130) also holds because $f_{\bar{A}}(r|\pi_\beta, s) = \delta(r - 1)$ for $s$ with $N(s) = 0$, which implies $\pi_{\text{DQL}}^{(1)}(a|s) = 0 \; \forall a$ with $0 \leq r < 1$. By integrating the both sides of (130) over $\{a \in \mathcal{A} : \bar{A}^{\pi_\beta}(s, a) \leq r\}$ and by definition of $F_A$ and $f_A$ in (99-100), we obtain

$$f_{\bar{A}}(r|\pi_{\text{DQL}}^{(1)}, s) \leq \frac{1}{1-r} f_{\bar{A}}(r|\pi_\beta, s) \; \forall s \in \mathcal{S}, 0 \leq r < 1. \tag{131}$$

**(Step 3.)** Now we are ready to prove (D.4). For all $s \in \mathcal{S}(c, n, t)$,

$$F_{\bar{A}}(t | \pi_{\text{DQL}}^{(1)}, s) = \int_{-\infty}^{t} f_{\bar{A}}(r | \pi_{\text{DQL}}^{(1)}, s) dr \tag{132}$$

$$= 1 - \int_{t}^{1} f_{\bar{A}}(r | \pi_{\text{DQL}}^{(1)}, s) dr \tag{133}$$

$$\geq 1 - \int_{t}^{1} \tfrac{1}{1-r} f_{\bar{A}}(r | \pi_{\beta}, s) dr \qquad \text{(by (131))} \tag{134}$$

$$\geq 1 - \int_{t}^{1} \tfrac{1}{1-r} c(1 - r^n) dr \qquad \text{(by definition of } \mathcal{S}(c, n, t) \tag{135}$$

$$= 1 - \int_{t}^{1} c(1 + r + \cdots + r^{n-1}) dr \quad \text{(by factor theorem and definition of } \mathcal{S}(c, n, t))) \tag{136}$$

$$\geq 1 - \int_{t}^{1} cn \, dr \qquad (\because 0 \leq r < 1) \tag{137}$$

$$= 1 - cn(1 - t) \tag{138}$$

$$> 0. \qquad \text{(by Assumption 3)} \tag{139}$$

$\square$

### D.6 PROOF OF THEOREM 4.5

For this proof, we concretize the statements of Theorem 4.5.

**Theorem 3.6.** *Let $\mathcal{A} = [-1,1]^d$ be an action space. For any $\delta > 0$, with $\varepsilon > 2^d\delta + 2\mathbb{E}_{s\sim\rho^{\pi_\beta}}[D_{TV}^\delta(\pi_\beta(\cdot|s), \pi_0(\cdot|s))]$, where*

$$D_{TV}^\delta(\pi_\beta(\cdot|s), \pi_0(\cdot|s)) := \frac{1}{2}\int_{a\in\mathcal{A}:\pi_\beta(a|s)<\delta} |\pi_\beta(a|s) - \pi_0(a|s)|da, \tag{140}$$

*any policy $\pi$ satisfying*

$$\mathbb{E}_{s\sim\rho^{\pi_\beta}}[D_{diff}(\pi_0(\cdot|s) \parallel \pi(\cdot|s))] < \frac{1}{2}(\varepsilon - 2^d\delta - 2\mathbb{E}_{s\sim\rho^{\pi_\beta}}[D_{TV}^\delta(\pi_\beta(\cdot|s), \pi_0(\cdot|s))])^2 \tag{141}$$

*is an $(\epsilon, \delta)$-in-distribution policy.*
*Note that $\mathbb{E}_{s\sim\rho^{\pi_\beta}}[D_{TV}^\delta(\pi_\beta(\cdot|s), \pi_\beta(\cdot|s))] = 0$ and $\mathbb{E}_{s\sim\rho^{\pi_\beta}}[D_{TV}^\delta(\pi_\beta(\cdot|s), \pi_\eta^*(\cdot|s))]$ is small when $\eta$ is large enough.*

---

We first introduce lemmas to simplify the mathematical derivations, followed by the main proof.

**Lemma D.5.** *For any $\delta > 0$ and $\mathcal{A} = [-1,1]^d$, the following holds:*

$$\mathbb{E}_{s\sim\rho^{\pi_\beta}}[2D_{TV}(\pi(\cdot|s), \pi_0(\cdot|s))] > \mathbb{E}_{s\sim\rho^{\pi_\beta}, a\sim\pi(\cdot|s)}[\mathbb{P}[\pi_\beta(a|s) < \delta]] - 2^d\delta - 2\mathbb{E}_{s\sim\rho^{\pi_\beta}}[D_{TV}^\delta(\pi_\beta(\cdot|s), \pi_0(\cdot|s))]. \tag{142}$$

*Proof.* For each $s \in \mathcal{S}$, define

$$\mathcal{A}(s,\delta) := \{a \in \mathcal{A} = [-1,1]^d : \pi_\beta(a|s) < \delta\} \tag{143}$$

By the definition of $\mathcal{A}(s,\delta)$ and $m(\mathcal{A}(s,\delta)) \le m(\mathcal{A}) = 2^d$ for Lebesgue measure $m(\cdot)$, we get

$$\int_{\mathcal{A}(s,\delta)} \pi_\beta(a|s)da < \int_{\mathcal{A}(s,\delta)} \delta\, da \le 2^d\delta, \tag{144}$$

and by definition of $\mathcal{A}(s,\delta)$, we get

$$\int_{\mathcal{A}(s,\delta)} \pi(a|s)da = \int_{\mathcal{A}} \pi(a|s)\mathbf{1}_{[\pi_\beta(a|s)<\delta]}da = \mathbb{E}_{a\sim\pi(\cdot|s)}[\mathbb{P}[\pi_\beta(a|s) < \delta]]. \tag{145}$$

Then, we have

$$2D_{\text{TV}}(\pi(\cdot|s), \pi_0(\cdot|s))$$

$$:= \int_{\mathcal{A}} |\pi(a|s) - \pi_0(a|s)|da \qquad\qquad \text{(by definition of TV distance)}$$

$$\ge \int_{\mathcal{A}(s,\delta)} |\pi(a|s) - \pi_0(a|s)|da \qquad\qquad (\because \mathcal{A}(s,\delta) \subseteq \mathcal{A}, |\pi(a|s) - \pi_0(a|s)| \ge 0)$$

$$\ge \int_{\mathcal{A}(s,\delta)} |\pi(a|s) - \pi_\beta(a|s)|da - \int_{\mathcal{A}(s,\delta)} |\pi_\beta(a|s) - \pi_0(a|s)|da \qquad \text{(by triangular inequality)}$$

$$\ge \int_{\mathcal{A}(s,\delta)} \pi(a|s)da - \int_{\mathcal{A}(s,\delta)} \pi_\beta(a|s)da - \int_{\mathcal{A}(s,\delta)} |\pi_\beta(a|s) - \pi_0(a|s)|da \qquad \text{(by triangular inequality)}$$

$$= \int_{\mathcal{A}(s,\delta)} \pi(a|s)da - \int_{\mathcal{A}(s,\delta)} \pi_\beta(a|s)da - 2D_{\text{TV}}^\delta(\pi_\beta(\cdot|s), \pi_0(\cdot|s)) \qquad \text{(by def. of } D_{\text{TV}}^\delta(\pi_\beta(\cdot|s), \pi_0(\cdot|s)))$$

$$> \int_{\mathcal{A}(s,\delta)} \pi(a|s)da - 2^d\delta - 2D_{\text{TV}}^\delta(\pi_\beta(\cdot|s), \pi_0(\cdot|s)) \qquad\qquad \text{(by (144))}$$

$$\ge \mathbb{E}_{a\sim\pi(\cdot|s)}[\mathbb{P}[\pi_\beta(a|s) < \delta]] - 2^d\delta - 2D_{\text{TV}}^\delta(\pi_\beta(\cdot|s), \pi_0(\cdot|s)) \qquad\qquad \text{(by (145))}.$$

We get the conclusion by taking expectations along $\rho^{\pi_\beta}$ on both sides. □

**Lemma D.6.** *The following holds:*

$$\mathbb{E}_{s \sim \rho^{\pi_\beta}}[2D_{TV}(\pi_0(\cdot|s), \pi(\cdot|s))] \leq \mathbb{E}_{s \sim \rho^{\pi_\beta}}[\sqrt{2D_{diff}(\pi_0(\cdot|s) \parallel \pi(\cdot|s))}]. \tag{146}$$

*where $D_{TV}(\cdot, \cdot)$ denotes the total variation distance.*

*Proof.* By Pinsker's inequality and the property that $D_{\mathrm{FKL}}(\pi_0(\cdot|s) \parallel \pi(\cdot|s)) \leq D_{\mathrm{diff}}(\pi_0(\cdot|s) \parallel \pi(\cdot|s))'$, we get:

$$\mathbb{E}_{s \sim \rho^{\pi_\beta}}[2D_{\mathrm{TV}}(\pi_0(\cdot|s), \pi(\cdot|s))] \leq \mathbb{E}_{s \sim \rho^{\pi_\beta}}[\sqrt{2D_{\mathrm{FKL}}(\pi_0(\cdot|s) \parallel \pi(\cdot|s))}]$$
$$\leq \mathbb{E}_{s \sim \rho^{\pi_\beta}}[\sqrt{2D_{\mathrm{diff}}(\pi_0(\cdot|s) \parallel \pi(\cdot|s))}]. \tag{147}$$

$\square$

*Proof of Theorem 4.5.* Here we write $C(\delta, \pi_\beta, \pi_0) = 2^d\delta + \mathbb{E}_{s \sim \rho^{\pi_\beta}}[D_{\mathrm{TV}}^\delta(\pi_\beta(\cdot|s), \pi_0(\cdot|s))]$ for simplicity.

$$\mathbb{E}_{s \sim \rho^{\pi_\beta}, a \sim \pi(\cdot|s)}[\mathbb{P}[\pi_\beta(a|s) < \delta]] \tag{148}$$
$$< \mathbb{E}_{s \sim \rho^{\pi_\beta}}[\sqrt{2D_{\mathrm{TV}}(\pi(\cdot|s), \pi_0(\cdot|s))}] + C(\delta, \pi_\beta, \pi_0) \quad \text{(by Lemma } D.5) \tag{149}$$
$$\leq \mathbb{E}_{s \sim \rho^{\pi_\beta}}[\sqrt{2D_{\mathrm{diff}}(\pi_0(\cdot|s) \parallel \pi(\cdot|s))}] + C(\delta, \pi_\beta, \pi_0) \quad \text{(by Lemma } D.6) \tag{150}$$
$$\leq \sqrt{2\mathbb{E}_{s \sim \rho^{\pi_\beta}}[D_{\mathrm{diff}}(\pi_0(\cdot|s) \parallel \pi(\cdot|s))]} + C(\delta, \pi_\beta, \pi_0) \quad \text{(by Jensen's inequality)} \tag{151}$$
$$\leq \sqrt{2 \cdot \tfrac{1}{2}(\varepsilon - C(\delta, \pi_\beta, \pi_0))^2} + C(\delta, \pi_\beta, \pi_0) \quad \text{(by (141))} \tag{152}$$
$$= (\varepsilon - C(\delta, \pi_\beta, \pi_0)) + C(\delta, \pi_\beta, \pi_0) \quad \text{(by the condition for } \varepsilon), \tag{153}$$
$$= \varepsilon, \tag{154}$$

Therefore, by Definition 4.4 in Section 4, $\pi$ is an $(\varepsilon, \delta)$-in-distribution policy. $\square$

**Remark** Similarly, Theorem 4.5 can also be established for forward and reverse KL divergences in place of diffusion divergence. For these divergences, observe that the key requirement is the form (146) from Lemma D.6.

For the forward KL divergence, the result follows directly from (147). For the reverse KL divergence, we use the symmetry of the total variation distance. Specifically, observe that

$$\mathbb{E}_{s \sim \rho^{\pi_\beta}}[2D_{\mathrm{TV}}(\pi_0(\cdot|s), \pi(\cdot|s))] = \mathbb{E}_{s \sim \rho^{\pi_\beta}}[2D_{\mathrm{TV}}(\pi(\cdot|s), \pi_0(\cdot|s))], \tag{155}$$

and

$$\mathbb{E}_{s \sim \rho^{\pi_\beta}}[2D_{\mathrm{TV}}(\pi(\cdot|s), \pi_0(\cdot|s))] \leq \mathbb{E}_{s \sim \rho^{\pi_\beta}}[\sqrt{2D_{\mathrm{FKL}}(\pi(\cdot|s) \parallel \pi_0(\cdot|s))}]$$
$$= \mathbb{E}_{s \sim \rho^{\pi_\beta}}[\sqrt{2D_{\mathrm{RKL}}(\pi_0(\cdot|s) \parallel \pi(\cdot|s))}], \tag{156}$$

which gives the desired result.

## D.7 PROOFS FOR SEC. 3.2

We begin with the formal definition of the MMSE estimator property.

**Definition D.7** (MMSE estimator property). Let $y$ denote the measurements, and let $\hat{x}(y)$ be an estimator of $x$. The mean squared error (MSE) loss is defined as

$$\mathbb{E}_{x\sim p(x|y)}\left[||\hat{x}(y) - x||^2\right].$$

The estimator that minimizes this loss is unique and is given by

$$\hat{x}_{\text{MMSE}}(y) := \mathbb{E}_{x\sim p(x|y)}[x].$$

With this, by choosing $y = (a^k, k, s)$ and $x = \epsilon$ and noticing that $\epsilon$ is a function of $(a^0, a^k, k)$, we get

$$\epsilon^*_\eta(a^k, k, s) = \mathbb{E}_{a^0\sim\pi^*_\eta(a^0|a^k,s)}[\epsilon] = \mathbb{E}_{\epsilon\sim\pi^*_\eta(\epsilon|a^k,s)}[\epsilon]$$

as the minimizer of (11), which is written again as

$$L^*_\eta(\theta) = \mathbb{E}_{s\sim\mathcal{D},a^0\sim\pi^*_\eta(a^0|s),k\sim U[1,K]}\left[||\epsilon_\theta(a^k, k, s) - \epsilon||^2\right]. \tag{157}$$

Note that the minimizer is defined for each $y = (a^k, k, s)$ and the similar thing holds for $\epsilon_\beta(a^k, k, s) = \mathbb{E}_{a^0\sim\pi_\beta(a^0|a^k,s)}[\epsilon]$.

*Proof of* $\epsilon^*_\eta(a^k, k, s) = -\sqrt{1-\bar{\alpha}^k}\nabla_{a^k}\log\pi^*_\eta(a^k|s)$. Consider the following score matching loss Song et al. (2021):

$$\mathbb{E}_{s\sim\mathcal{D},a^0\sim\pi^*_\eta,k\sim U[1,K]}\left[||s_\theta(a^k, k, s) - \nabla_{a^k}\log q(a^k|a^0)||^2\right]. \tag{158}$$

By Lemma 1 of Fang et al. (2024), one has

$$\nabla_{a^k}\log q(a^k|a^0) = -\frac{\epsilon}{\sqrt{1-\bar{\alpha}^k}}$$

when $a^k = \sqrt{\bar{\alpha}^k}a^0 + \sqrt{1-\bar{\alpha}^k}\epsilon$. Therefore, (158) is equivalent to

$$\mathbb{E}_{s\sim\mathcal{D},a^0\sim\pi^*_\eta,k\sim U[1,K]}\left[\left\|s_\theta(a^k, k, s) + \frac{\epsilon}{\sqrt{1-\bar{\alpha}^k}}\right\|^2\right]. \tag{159}$$

By the MMSE estimator property, the minimizer of (159) is

$$-\frac{\epsilon^*_\eta(a^k, k, s)}{\sqrt{1-\bar{\alpha}^k}}.$$

Alternatively, applying the MMSE property directly to (158) gives

$$\mathbb{E}_{a^0\sim\pi^*_\eta(a^0|a^k,s)}\left[\nabla_{a^k}\log q(a^k|a^0)\right],$$

noting that $\nabla_{a^k}\log q(a^k|a^0)$ is a function of $(a^0, a^k, k)$. We directly compute this conditional expectation:

$$\mathbb{E}_{a^0\sim\pi^*_\eta(a^0|a^k,s)}[\nabla_{a^k}\log q(a^k|a^0)] = \int \pi^*_\eta(a^0|a^k, s)\nabla_{a^k}\log q(a^k|a^0)da^0$$

$$= \int \pi^*_\eta(a^0|a^k, s)\frac{\nabla_{a^k}q(a^k|a^0)}{q(a^k|a^0)}da^0 \qquad \text{(by log-derivative trick)}$$

$$= \int \frac{\pi^*_\eta(a^0|s)q(a^k|a^0)}{\pi^*_\eta(a^k|s)}\frac{\nabla_{a^k}q(a^k|a^0)}{q(a^k|a^0)}da^0 \qquad \text{(by Bayes' rule)}$$

$$= \frac{1}{\pi^*_\eta(a^k|s)}\nabla_{a^k}\left(\int \pi^*_\eta(a^0|s)q(a^k|a^0)da^0\right) \qquad \text{(by Leibniz rule)}$$

$$= \frac{1}{\pi^*_\eta(a^k|s)}\nabla_{a^k}\pi^*_\eta(a^k|s) \qquad \text{(by definition of } \pi^*_\eta(a^k|s))$$

$$= \nabla_{a^k}\log\pi^*_\eta(a^k|s). \qquad \text{(by log-derivative trick)}$$

Thus,

$$\nabla_{a^k} \log \pi_\eta^*(a^k|s) = \mathbb{E}_{a^0 \sim \pi_\eta^*(a^0|a^k,s)}[\nabla_{a^k} \log q(a^k|a^0)] = -\frac{\epsilon_\eta^*(a^k, k, s)}{\sqrt{1 - \bar\alpha^k}},$$

or equivalently,

$$\epsilon_\eta^*(a^k, k, s) = \sqrt{1 - \bar\alpha^k} \nabla_{a^k} \log \pi_\eta^*(a^k|s).$$

completing the proof. $\square$

Note that $\epsilon_\beta(a^k, k, s) = -\sqrt{1 - \bar\alpha^k} \nabla_{a^k} \log \pi_\beta(a^k|s)$ can be proven in the similar way.

*Proof of (17).* Recall that

$$f(a^k, k, s) = \log \pi_\eta^*(a^k|s) - \log \pi_\beta(a^k|s) + \log Z(s) \tag{160}$$

and

$$\frac{1}{\eta} Q_\phi(s, a^0) = \log \pi_\eta^*(a^0|s) - \log \pi_\beta(a^0|s) + \log Z(s). \tag{161}$$

Then, we get

$$\tilde f(a^k, k, s) - f(a^k, k, s)$$

$$= \frac{1}{\eta} \mathbb{E}_{a^0 \sim \pi_\beta(a^0|a^k,s)} \left[ Q_\phi(s, a^0) \right] - \log \pi_\eta^*(a^k|s) + \log \pi_\beta(a^k|s) - \log Z(s) \qquad \text{(by(15, 160))}$$

$$= \mathbb{E}_{a^0 \sim \pi_\beta(a^0|a^k,s)} \left[ \frac{1}{\eta} Q_\phi(s, a^0) - \log \pi_\eta^*(a^k|s) + \log \pi_\beta(a^k|s) - \log Z(s) \right] \qquad \text{(by definition of expectation)}$$

$$= \mathbb{E}_{a^0 \sim \pi_\beta(a^0|a^k,s)} \left[ \log \pi_\eta^*(a^0|s) - \log \pi_\beta(a^0|s) + \log \pi_\eta^*(a^k|s) + \log \pi_\beta(a^k|s) \right] \qquad \text{(by (161))}$$

$$= \mathbb{E}_{a^0 \sim \pi_\beta(a^0|a^k,s)} \left[ \log \frac{\pi_\eta^*(a^0|s) q(a^k|a^0) / \pi_\eta^*(a^k|s)}{\pi_\beta(a^0|s) q(a^k|a^0) / \pi_\beta(a^k|s)} \right]$$

$$= \mathbb{E}_{a^0 \sim \pi_\beta(a^0|a^k,s)} \left[ \log \frac{\pi_\eta^*(a^0|a^k, s)}{\pi_\beta(a^0|a^k, s)} \right] \qquad \text{(by Bayes' rule)}$$

$$= -D_{\mathrm{KL}}(\pi_\beta(a^0|a^k, s) \parallel \pi_\eta^*, \qquad \text{(by definition of KL divergence)}$$

completing the proof. $\square$

**Remark** It follows that (12) and (11) share the same minimizer. The surrogate proposed by Fang et al. (2024) appears similar to (12) but is derived differently. Their approach relies on the score function relation with a relaxation where the expectation is taken under $\pi_\beta$ instead of $\pi_\eta^*$, without addressing minimizer preservation. It also employs the inner product technique of Vincent (2011), which is less intrinsic than the MMSE estimator property, together with a smooth extension assumption (referred to as the "naive extension" in Sec. 3.2). In contrast, our construction establishes a stronger theoretical correspondence and offers clearer interpretability.

# E EXPERIMENTAL DETAILS

## E.1 2D BANDIT EXPERIMENT

**Settings** We mostly followed the setup from the 2D bandit experiment in DQL codebase (Wang et al., 2023). The environment consists of a single state and a 2D box action space $[-1, 1]^2$. The dataset contains four Gaussian clusters centered at $(\pm 0.8, \pm 0.8)$ with standard deviation 0.05 per dimension, each containing 2500 samples. The reward function $R(a)$ assigns Gaussian rewards with means of 3.0, 1.5, 5.0, and 0.0, starting from the top-left corner and proceeding clockwise, each with standard deviation 0.5.

We regarded the 99% confidence regions (CR) of each cluster as in-distribution, computed using the $\chi^2$ distribution with 2 degrees of freedom centered at $(\pm 0.8, \pm 0.8)$. The threshold radii were set to $0.05 \times 3.03$, where $F_{\chi^2(2)}^{-1}(0.99) \approx 3.03$.

**Algorithm Implementation** We adopted double Q-learning (Hasselt, 2010) and trained the critic with simple MSE regression toward $R(a)$. Noise models were optimized using algorithm-specific actor losses. Except for "Explicit", we employed the formulation in (50) (see Appendix C.1):

$$L_{\text{TDP}}(\theta) = \eta L_{\text{diff}}(\theta) + L_{\text{guide}}(\theta) - \zeta L_{Q_\phi}(\theta), \tag{162}$$

with $L_{\text{guide}}$ defined as

$$L_{\text{guide}}(\theta) := \mathbb{E}_{(s,a^0) \sim \mathcal{D}, k \sim U[1,K]} \left[ \sqrt{1 - \bar{\alpha}^k} \left\langle \nabla_{a^k} Q_\phi(s, a^0), \epsilon_\theta(a^k, k, s) \right\rangle \right], \tag{163}$$

where $\langle \cdot, \cdot \rangle$ denotes the inner product.

For noisy estimation variants, the only difference lies in the definition of $L_{\text{guide}}$, which is replaced by

$$L'_{\text{guide}}(\theta) := \mathbb{E}_{(s,a^0) \sim \mathcal{D}, k \sim U[1,K]} \left[ \sqrt{1 - \bar{\alpha}^k} \left\langle \nabla_{a^k} Q_\phi(s, a^k), \epsilon_\theta(a^k, k, s) \right\rangle \right]. \tag{164}$$

As a result, "TDP" used the actor loss

$$L(\theta) = \eta L_{\text{diff}}(\theta) + L_{\text{guide}}(\theta) - \zeta L_{Q_\phi}(\theta), \tag{165}$$

while "Noisy" used

$$L(\theta) = \eta L_{\text{diff}}(\theta) + L'_{\text{guide}}(\theta) - \zeta L_{Q_\phi}(\theta). \tag{166}$$

The "Implicit" and "Noisy+Implicit" variants correspond to (165) and (166), respectively, with the $\zeta L_{Q_\phi}(\theta)$ term removed. Finally, "Explicit" used

$$L(\theta) = L_{\text{diff}}(\theta) - \zeta L_{Q_\phi}(\theta). \tag{167}$$

**Model Structure** We used the default settings in Wang et al. (2023): each Q network and noise model is a three-layer MLP with 256 hidden units per layer and Mish activation (Misra, 2020). The noise model also incorporates sinusoidal time embeddings.

**Common hyperparameters** We sampled 100 samples for visualization. We used the Adam optimizer (Kingma & Ba, 2015) for training. The learning rate for the actor and critic updates is set to $3 \times 10^{-4}$. We adopted the implementation of DDPM (Ho et al., 2020) for the diffusion model with 50 diffusion steps ($K$) and variance-preserving (VP) scheduling (Song et al., 2021). An exponential moving average (EMA) update is applied with a decay rate $\tau = 0.995$ on actor networks. All models are trained for 1000 epochs with a batch size 100 ($B$), where each epoch consists of a single full sweep of the dataset.

**Per-Algorithm Hyperparameters** We used $\zeta = 2.5$ for "DQL", following the default setting from the original codebase. For "TDP" and "$a^0 \to a^k$", we set $(\eta, \zeta) = (0.05, 2.5)$, while for "No Q Loss" and "Both", we used $(\eta, \zeta) = (0.05, 0)$, which were found to yield the best qualitative performance.

## E.2 IMPLEMENTATION AND HYPERPARAMETERS

**Codes** Our implementation is based on the CleanDiffuser (Dong et al., 2024) codebase, with semantic references to DAC (Fang et al., 2024). We also drew implementation guidance from the codebases of DQL (Wang et al., 2023) and DiffCPS (He et al., 2023).

**Model Structure** We adopted the noise model structure of DQL (Wang et al., 2023), which uses an MLP-based model architecture. We enhanced this with Fourier time embeddings, following DAC. Each Q network is modeled as a three-layer MLP with 256 hidden units per layer, employing layer normalization (Lei Ba et al., 2016) followed by Mish activation (Misra, 2020).

**Common Hyperparameters** We used the Adam optimizer (Kingma & Ba, 2015) with cosine learning rate decay (Loshchilov & Hutter, 2017), setting the minimum learning rate to one-tenth of the initial value. State normalization was applied to all environments. The learning rate for the actor, critic, and $\eta$ updates was set to $3 \times 10^{-4}$ across all tasks. Following DQL (Wang et al., 2023), we implemented the DDPM diffusion model (Ho et al., 2020) with 5 diffusion steps ($K$) and a variance-preserving (VP) noise schedule (Song et al., 2021). Consistent with DAC (Fang et al., 2024), we used 10 Q ensembles ($N_q$) and 10 action samples ($N_a$) for actor learning. For action selection, we sampled 50 candidates ($N_c$) and performed hard maximization. An exponential moving average (EMA) update with decay 0.995 ($\tau$) was applied to both the actor and critic networks. All models were trained for 2 million steps (M) with a batch size 256 (B). We evaluated each environment using four random seeds, with 50 evaluations per model, and reported the final scores. To reduce computational overhead, we computed the guidance and Q losses for actor updates using only half of the batch. Additionally, we adopted low-temperature sampling by initializing the sampling procedure in the evaluation phase from $a^K \sim 0.5 \cdot \mathcal{N}(0, I)$, following DD (Ajay et al., 2023).

**Basic Per-Environment Training Hyperparameters** For MuJoCo (Todorov et al., 2012) locomotion tasks, we adopted the reward tuning scheme from IQL (Kostrikov et al., 2022). For Antmaze, we applied customized scaling by mapping rewards $0 \mapsto -1$ and $1 \mapsto 10$, while Kitchen tasks used the raw rewards. The policy update period was set to 1 for Antmaze and 2 for all other environments. The discount factor was $\gamma = 0.995$ for Antmaze and $\gamma = 0.99$ elsewhere. For Antmaze, we additionally employed max Q backup (Kumar et al., 2020). We primarily followed DAC (Fang et al., 2024) for hyperparameter choices where applicable, including $\varepsilon_b^\eta$ (or $\eta$) and $\rho$, to reduce tuning effort. Additional implementation details are provided in the corresponding paragraphs.

**Locomotion Tasks** In Table 8, we provide per-environment hyperparameters for MuJoCo locomotion environments. For $\varepsilon_b^\eta$, if DAC used $\varepsilon_b^\eta = 1.0$, we tuned within $\{0.8, 1.0\}$ and followed DAC's setting elsewhere. For $\rho$, we adopted values of DAC when available: if $\rho = 1.0$ (in Hopper and Walker2d), we tuned within $\{1.0, 1.25\}$; if $\rho = 0$ (in Halfcheetah), we kept $\rho = 0$. For the balance coefficient $\lambda$, we tuned over $\{0.1, 0.2, 0.5, 0.8, 0.9\}$. Notably, tasks with higher stability, either due to dataset coverage (medium-replay) or physical agent characteristics (Halfcheetah, which has two contact points and a low center of mass), can tolerate higher $\lambda$ values.

**Antmaze Tasks** In Table 9, we provide per-environment hyperparameters for Antmaze environments. For these tasks, we fixed $\eta$ due to its high sensitivity to $\varepsilon_b^\eta$ and for consistency with DAC. Instead of tuning $\varepsilon_b^\eta$, we directly tuned $\eta$ along with $\rho$ and $\lambda$. We adopted $(\eta, \rho, \lambda) = (0.1, 1.0, 0.5)$ as the default configuration from DAC and modified at most one of these values per environment.

**Kitchen Tasks** In Table 10, we provide per-environment hyperparameters for Kitchen environments. For Kitchen tasks, we tuned $(\varepsilon_b^\eta, \rho, \lambda)$ following as for locomotion tasks. Since we do not know the hyperparameters used for Kitchen in DAC, we referred to the extremely low $\eta$ used in DQL (Wang et al., 2023), which corresponds to the reciprocal of $\zeta$ in TDP, and accordingly chose low values for $\varepsilon_b^\eta$ and $\lambda$. Further tuning was guided by keeping at least two of the three hyperparameters consistent across Kitchen partial and mixed.

Table 8: Per-environment hyperparameters in MuJoCo locomotion environments.

| Environment | $\varepsilon_b^\eta$ | $\rho$ | $\lambda$ |
|---|---|---|---|
| Halfcheetah m | 0.8 | 0.0 | 0.9 |
| Halfcheetah m-r | 0.8 | 0.0 | 0.8 |
| Halfcheetah m-e | 0.1 | 0.0 | 0.5 |
| Hopper m | 1.0 | 1.25 | 0.1 |
| Hopper m-r | 1.0 | 1.25 | 0.8 |
| Hopper m-e | 0.05 | 1.25 | 0.1 |
| Walker2d m | 0.8 | 1.25 | 0.2 |
| Walker2d m-r | 1.0 | 1.0 | 0.9 |
| Walker2d m-e | 1.0 | 1.25 | 0.1 |

Table 9: Per-environment hyperparameters in Antmaze environments.

| Environment | $\eta$ | $\rho$ | $\lambda$ |
|---|---|---|---|
| Umaze | 5.0 | 1.0 | 0.5 |
| Umaze d | 0.1 | 1.0 | 0.5 |
| Medium p | 0.1 | 1.0 | 0.2 |
| Medium d | 0.1 | 1.0 | 0.5 |
| Large p | 0.05 | 1.0 | 0.5 |
| Large d | 0.1 | 1.25 | 0.5 |
| Ultra p | 0.1 | 1.0 | 0.5 |
| Ultra d | 0.1 | 1.0 | 0.2 |

Table 10: Per-environment hyperparameters in Kitchen environments.

| Environment | $\varepsilon_b^\eta$ | $\rho$ | $\lambda$ |
|---|---|---|---|
| Kitchen p | 0.005 | 0.5 | 0.1 |
| Kitchen m | 0.01 | 0.5 | 0.1 |

### E.3 IMPLEMENTATION AND HYPERPARAMETERS FOR FLOW MATCHING VARIANTS

**Codes** Our implementation builds on the FQL (Park et al., 2025) codebase, with additional guidance taken from diffusion policy implementations mentioned in Appendix E.2.

**Model Structure** We adopted the vector field architecture of FQL, where the time input (0–1) is concatenated into an MLP. Each Q network is a three-layer MLP with 256 hidden units per layer for locomotion tasks, and a four-layer MLP with 512 units for Antmaze, using layer normalization (Lei Ba et al., 2016) (Antmaze only) and Mish activation (Misra, 2020). The policy network has the same size as the Q network.

**Common Hyperparameters** We used the Adam optimizer (Kingma & Ba, 2015) with a learning rate of $3 \times 10^{-4}$ for both actor and critic. Following FQL (Park et al., 2025), the continuous-time flow vector field was implemented with 10 steps ($N$) using the Euler method. For action selection, we sampled 50 candidates ($N_c$) and applied soft maximization with temperature 1. All models were trained for 2M steps on locomotion tasks and 1M on Antmaze, following Alles et al. (2025), with batch size 256 ($B$). Each environment was evaluated with four random seeds, using 10 rollouts per seed for locomotion and 50 for Antmaze, also following Alles et al. (2025), and we report the final scores.

**Basic Per-Environment Training Hyperparameters**   For Antmaze, we applied IQL (Kostrikov et al., 2022) tuning, i.e., mapping rewards of 0 to $-1$ and 1 to 0. The discount factor was set to $\gamma = 0.995$ for Antmaze and $\gamma = 0.99$ for locomotion. We tuned $\alpha$ and $\beta$, corresponding to $\frac{1}{\eta}$ and $\frac{1}{\zeta}$, respectively. For $\beta$, our tuning strategy for locomotion mostly follows the "Tips for hyperparameter tuning" in FQL codebase website (Park et al., 2025). Further implementation details are given in the corresponding sections.

**Locomotion Tasks**   In Table 11, we provide per-environment hyperparameters for MuJoCo locomotion environments. We searched 1–3 (mostly 2) candidate values of $\alpha$ or $\beta$ per environment by using only one of the two, and selected the final value while avoiding over-diversification.

**Antmaze Tasks**   In Table 9, we provide per-environment hyperparameters for Antmaze environments. For these tasks, we fixed $\beta = 10$ and chose $\alpha \in \{2, 5\}$ per environment.

Table 11: Per-environment hyperparameters in MuJoCo locomotion environments.

| Environment | $\alpha$ | $\beta$ |
|---|---|---|
| Halfcheetah m | 0.05 | 0.001 |
| Halfcheetah m-r | 0.05 | 0.001 |
| Halfcheetah m-e | 0.05 | 0.1 |
| Hopper m | 0.01 | 0.1 |
| Hopper m-r | 0.01 | 0.1 |
| Hopper m-e | 0.01 | 3 |
| Walker2d m | 0.05 | 0.3 |
| Walker2d m-r | 0.05 | 0.3 |
| Walker2d m-e | 0.05 | 0.1 |

Table 12: Per-environment hyperparameters in Antmaze environments.

| Environment | $\alpha$ | $\beta$ |
|---|---|---|
| Umaze | 5 | 10 |
| Umaze d | 2 | 10 |
| Medium p | 2 | 10 |
| Medium d | 2 | 10 |
| Large p | 2 | 10 |
| Large d | 5 | 10 |

### E.4 DETAILS ON BASELINES

The policy constraint method baselines include CQL (Kumar et al., 2020), IQL (Kostrikov et al., 2022), ReBRAC (Tarasov et al., 2023), and QCS (Kim et al., 2024). Diffusion policies include DQL (Wang et al., 2023), QGPO (Lu et al., 2023), DiffCPS (He et al., 2023), and DAC (Fang et al., 2024). Q ensemble-based algorithms include SAC-N and EDAC (An et al., 2021). Goal-conditioned methods consist of GC-IQL (Kostrikov et al., 2022), GC-POR (Xu et al., 2022), HIQL (Park et al., 2023), and GCPC (Zeng et al., 2023). We detail the source of the performance scores below.

**Locomotion**  All baseline scores were obtained from their respective original papers.

**Antmaze**  Performance scores for the baselines were primarily obtained from their original papers, with the following exceptions: scores for IQL, GC-IQL, and GCPC on Antmaze tasks were sourced from (Zeng et al., 2023), and GC-POR results were taken from its original paper for umaze tasks and from (Park et al., 2023) for the remaining Antmaze tasks. Additionally, we conducted experiments for settings not covered in prior work, including diffusion policy baselines on Antmaze ultra tasks and HIQL on umaze tasks.

Several baselines did not report results for specific Antmaze tasks, namely, the ultra tasks for DQL(Wang et al., 2023), QGPO (Lu et al., 2023), DiffCPS (He et al., 2023), and DAC (Fang et al., 2024), and the umaze tasks for HIQL (Park et al., 2023). Therefore, we reproduced those results. For DQL, DiffCPS, and DAC, we tuned hyperparameters for the Antmaze ultra datasets based on those used for the corresponding large tasks: we adopted the Antmaze large-play settings for ultra-play and the large-diverse settings for ultra-diverse. Each method was evaluated using two discount factors, $\gamma \in \{0.99, 0.995\}$, and the better-performing result was reported.

For QGPO, we selected the best performance among 12 configurations by sweeping $\gamma \in \{0.99, 0.995\}$ and guidance scale values from $\{1.0, 1.5, 2.0, 2.5, 3.0, 4.0\}$, where the latter sweep follows the configuration range suggested in the original paper. For HIQL, we applied the hyperparameter configuration from Antmaze large-diverse uniformly across both umaze tasks.

DQL and DiffCPS were evaluated using online model selection (OMS), where the best return per seed was selected across evaluation checkpoints. To improve selection stability, the evaluation interval was increased from the standard 10K steps to 50K steps. For all other algorithms, we reported the final evaluation scores.

All algorithms were evaluated with four random seeds, using 100 evaluation episodes per model.

**Kitchen**  The scores for CQL, IQL, DQL, and DAC were obtained from Fang et al. (2024), and the scores for DT, GC-IQL, and GCPC for Kitchen tasks were obtained from Zeng et al. (2023).

**Flow Policies**  The scores for FlowQ were obtained from their respective original papers. The scores for FBRAC for locomotion tasks were obtained from Alles et al. (2025) (as the name "(FM) DiffusionQL"). The scores for FQL, FBRAC, and IFQL for Antmaze tasks were obtained from Park et al. (2025). The scores for TDP were from Tables 1 and 2.

Since the scores for FQL and IFQL on locomotion tasks are not provided in Park et al. (2025), we reproduced using FQL codebase with almost the same basic hyperparameter settings for TFP, except for the inclusion of layer normalization for Q network, which is the default setting of FQL.

For FQL, we tuned $\beta$ by following "Tips for hyperparameter tuning" in FQL codebase website (Park et al., 2025). For IFQL, we used expectile $\tau = 0.7$, following IQL (Kostrikov et al., 2022) and IDQL (Hansen-Estruch et al., 2023). For action selection, we tested $\{32, 64, 128\}$ action candidates with hard maximization, and found that 32 candidates showed the best result and reported the scores from such a configuration.

# F   ADDITIONAL EXPERIMENTAL RESULTS

## F.1   ABLATIONS

**Additional Experiments on Noise Estimation and Q loss**    In addition to the 2D bandit experiments in Sec. 3.3 and the Antmaze ablations in Sec. 6.3, we conducted further experiments on D4RL locomotion environments to assess the effects of noise-free estimation and the presence of the Q loss. Specifically, we evaluated "TDP", "Noisy", "Implicit", and "Noisy+Implicit" across nine locomotion tasks. The results for "TDP" and "Noisy+Implicit" (corresponding to DAC) were taken directly from Table 1, while "Noisy" and "Implicit" were obtained through reimplementations of TDP.

As shown in Table 13, the inclusion of the Q loss term has a more substantial impact on performance than noise-free estimation. The effect of noise-free estimation is less pronounced in most locomotion tasks, which are generally more tolerant to out-of-distribution (OOD) actions due to their broader and more connected in-distribution regions. In contrast, the 2D bandit environment contains four disconnected clusters with large gaps, making it highly sensitive to OOD behavior.

Nevertheless, noise-free estimation remains important. In environments such as Hopper medium and Walker2d medium-expert, "Noisy" suffered from severe performance degradation, indicating that noisy estimation can increase OOD risk. Consequently, its average performance becomes comparable to "Noisy+Implicit" (DAC), despite achieving high scores in environments such as Halfcheetah medium and Walker2d medium.

Finally, we observe a general trend in which larger $\lambda$ values, corresponding to stronger weighting on the Q loss, consistently improve policy performance with the presence of the Q loss. This pattern holds across most environments, with the exception of Walker2d medium, where smaller $\lambda$ values had a pronounced effect.

Table 13: Comparison of "TDP", "Noisy", "Implicit", and "Noisy+Implicit" (DAC) on locomotion tasks. For each task, the highest score is in bold. Values in parentheses indicate the $\lambda$ used for "TDP" and "Noisy".

| Dataset | Halfcheetah | | | Hopper | | | Walker2d | | | |
| | m (0.9) | m-r (0.8) | m-e (0.5) | m (0.1) | m-r (0.8) | m-e (0.1) | m (0.2) | m-r (0.9) | m-e (0.1) | Mean |
|---|---|---|---|---|---|---|---|---|---|---|
| TDP | 69.7 | **58.5** | **100.3** | **103.0** | **104.4** | 111.4 | **108.0** | **109.2** | 113.6 | **97.6** |
| Noisy | **70.7** | 58.3 | 98.4 | 80.1 | 103.5 | 111.6 | 107.8 | 103.3 | 98.4 | 92.5 |
| Implicit | 58.9 | 51.1 | 97.9 | 99.8 | 102.8 | 111.4 | 94.6 | 97.9 | 113.0 | 92.0 |
| Noisy+Implicit | 59.1 | 55.0 | 99.1 | 101.2 | 103.1 | **111.7** | 96.8 | 96.8 | **113.6** | 92.9 |

**Selection of Target Diffusion Divergence Value** TDP employs the target diffusion value $\varepsilon_b^\eta$ to indirectly regulate $\eta$, by controlling diffusion divergence. For locomotion tasks, we considered $\{0.8, 1.0\}$ for Halfcheetah medium, Halfcheetah medium-replay, Hopper medium, and Hopper medium-replay, and $\{0.05, 0.1\}$ for Halfcheetah medium-expert and Hopper medium-expert. For each task, one candidate value was selected, and the unselected one was used as an ablation. As shown in Tables 14 and 15, the choice of target value had only a minor effect on performance.

Table 14: The normalized scores of TDP with varying $\varepsilon_b^\eta$. The best score is in bold and the main reported score is underlined.

| $\varepsilon_\mathbf{b}^\eta$ | 0.8 | 1.0 |
| --- | --- | --- |
| Halfcheetah m | **69.7** | 68.6 |
| Halfcheetah m-r | **58.5** | 57.6 |
| Hopper m | 101.9 | **103.0** |
| Hopper m-r | 104.3 | **104.4** |

Table 15: The normalized scores of TDP with adaptive or fixed $\eta$. The best score is in bold and the main reported score is underlined.

| $\varepsilon_\mathbf{b}^\eta$ | 0.05 | 0.1 |
| --- | --- | --- |
| Halfcheetah m-e | 100.1 | **100.3** |
| Hopper m-e | **111.4** | 111.2 |

**Adaptive vs. Fixed $\eta$** TDP employs adaptive $\eta$ by setting $\varepsilon_b^\eta$ to achieve more stable results. For comparison, we also evaluated fixed $\eta$, chosen to approximately match the average of the finalized $\eta$ values in each environment. As shown in Table 16, the performance differences are minimal. Nevertheless, we adopted adaptive $\eta$ because it yields consistent target values $(0.05, 0.1, 0.8, 1.0)$, while fixed $\eta$ values vary substantially across environments.

Table 16: The normalized scores of TDP with varying $\varepsilon_b^\eta$. The best score is in bold and the main reported score is underlined.

| $\eta$ | Adaptive | Fixed |
| --- | --- | --- |
| Halfcheetah m | 69.7 | **69.9** |
| Halfcheetah m-r | 58.5 | **58.6** |
| Halfcheetah m-e | **100.3** | 99.8 |
| Hopper m | **103.0** | 97.9 |
| Hopper m-r | **104.4** | 104.2 |
| Hopper m-e | 111.4 | **111.9** |

**Selection of Confidence Interval**   We varied $\rho$ to investigate the effect of the confidence interval of LCB used for critic learning. As shown in Table 17, increasing $\rho$ slightly reduces performance but has a small impact overall, especially on the Halfcheetah medium-expert (m-e) task. These results support both the validity of the choice of $\rho = 0$ and the robustness of TDP to variations in $\rho$.

Table 17: The normalized scores of TDP with varying $\rho$. The best score is in bold and the main reported score is underlined.

| $\rho$ | 0 | 0.5 | 1.0 | 1.25 |
|---|---|---|---|---|
| Halfcheetah m | **_69.7_** | 68.2 | 66.4 | 64.6 |
| Halfcheetah m-r | **_58.5_** | 57.4 | 52.6 | 50.6 |
| Halfcheetah m-e | _100.3_ | 100.3 | **100.9** | 98.9 |

**Selection of Balance Coefficient**   As noted in Appendix C.1, TDP introduces a balance coefficient $\lambda$ to trade off between the Q loss and guidance terms: $\lambda = 0$ applies only the guidance term, while $\lambda = 1$ applies only the Q loss. We varied $\lambda$ on Halfcheetah medium and Hopper medium tasks. As shown in Table 18, though tuning $\lambda$ matters, TDP remains stable under small deviations from the values used in the main experiments.

Table 18: The normalized final scores of TDP with varying $\lambda$. For each task, the best score is in bold and the main reported score is underlined.

| $\lambda$ | 0 | 0.1 | 0.2 | 0.5 | 0.8 | 0.9 | 1.0 |
|---|---|---|---|---|---|---|---|
| Halfcheetah m | 58.9 | 60.0 | 61.1 | 65.9 | 67.4 | _69.7_ | **70.3** |
| Hopper m | 99.8 | **_103.0_** | 99.1 | 97.1 | 71.8 | 37.9 | 8.0 |

**Selection of Number of Qs**   TDP employs a Q ensemble to enhance performance. To evaluate its impact, we varied the number of Q networks, including $N_q = 2$, which corresponds to standard double Q-learning (Hasselt, 2010). As shown in Table 19, increasing the ensemble size provides limited additional benefit. Notably, double Q-learning led to slightly lower scores on Halfcheetah medium (m) and a more pronounced drop on Halfcheetah medium-replay (m-r). In conclusion, while the Q ensemble contributes to the performance of TDP, the results indicate that TDP does not heavily rely on it to achieve strong performance.

Table 19: The normalized scores of TDP with varying $N_q$. The best score is in bold and the main reported score is underlined.

| $N_q$ | 2 (double Q) | 5 | 10 | 20 |
|---|---|---|---|---|
| Halfcheetah m | 66.8 | **70.0** | 69.7 | 69.2 |
| Halfcheetah m-r | 52.5 | 55.9 | 58.5 | **58.6** |
| Halfcheetah m-e | **100.4** | 99.8 | 100.3 | 99.3 |

**Policy Update Intervals**   For the locomotion and Kitchen tasks, we adopted delayed policy updates with an interval of $d = 2$. To evaluate the impact of this choice, we compared performance with $d \in \{1, 2, 4\}$ on the Hopper tasks. Fig. 2 presents the resulting learning curves. As shown, $d = 2$ offers the best training stability, followed by $d = 1$ and $d = 4$. While the overall performance differences across values of $d$ are modest, the Hopper medium task shows a notable advantage with $d = 2$.

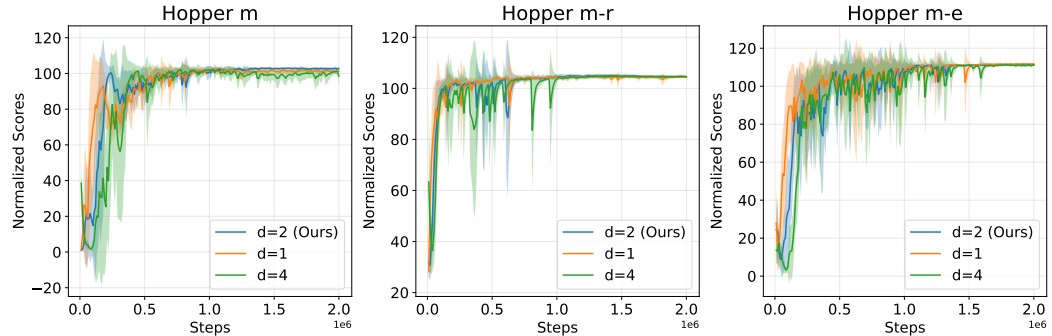

Figure 2: Learning curves of MuJoCo Hopper tasks with varying policy update interval $d$.

**Number of the Action Candidates** As mentioned in Appendix E, we used $N_c = 50$ action candidates for diffusion policy action sampling. Table 20 reports the results for $N_c \in \{10, 50, 100, 200\}$. As shown, using $N_c = 10$ slightly degrades performance, whereas $N_c \in \{50, 100, 200\}$ yield comparable results. Thus, $N_c = 50$ serves as a standard and effective choice, balancing performance and evaluation efficiency.

Table 20: The normalized scores of TDP with varying $N_c$. The best score is in bold and the main reported score is underlined.

| $\mathbf{N_c}$ | 10 | 50 | 100 | 200 |
|---|---|---|---|---|
| Halfcheetah m-r | $58.5 \pm 0.7$ | $\underline{58.5} \pm 0.6$ | $\mathbf{58.6} \pm 0.6$ | $\mathbf{58.6} \pm 0.7$ |
| Hopper m | $100.0 \pm 0.5$ | $\underline{103.0} \pm 0.1$ | $103.0 \pm 0.1$ | $\mathbf{103.1} \pm \mathbf{0.1}$ |
| Walker2d m-e | $113.4 \pm 0.5$ | $\underline{113.6} \pm 0.6$ | $\mathbf{113.7} \pm 0.5$ | $113.6 \pm 0.6$ |

**Temperature** As mentioned in Appendix E, we used a temperature of 0.5, i.e., $a^K \sim 0.5 \cdot \mathcal{N}(0, I)$, following DD (Ajay et al., 2023). Table 21 presents the results of TDP with temperatures 0, 0.5, and 1. As shown, TDP has little sensitivity to the choice of temperature.

Table 21: The normalized scores of TDP with varying temperatures. The best score is in bold and the main reported score is underlined.

| Temperature | 0.0 | 0.5 | 1.0 |
|---|---|---|---|
| Halfcheetah m-r | $58.7 \pm 0.8$ | $\underline{58.5} \pm 0.6$ | $\mathbf{58.8} \pm 0.7$ |
| Hopper m | $102.8 \pm 0.3$ | $\underline{\mathbf{103.0}} \pm 0.1$ | $102.9 \pm 0.1$ |
| Walker2d m-e | $\mathbf{113.6} \pm 0.6$ | $\underline{\mathbf{113.6}} \pm 0.6$ | $\mathbf{113.6} \pm 0.5$ |

**Diffusion Steps** As mentioned in Appendix E, we used $K = 5$ diffusion steps for TDP. Table 22 reports the performance of TDP with $K \in \{5, 10, 20\}$. As shown, longer diffusion steps slightly degrade performance, consistent with prior findings (He et al., 2023), and thus validate the correct implementation of TDP.

Table 22: The normalized scores of TDP with varying Diffusion steps. The best score is in bold and the main reported score is underlined.

| Diffusion Steps | 5 | 10 | 20 |
|---|---|---|---|
| Halfcheetah m | $\underline{\mathbf{69.7}} \pm 0.9$ | $68.6 \pm 0.6$ | $68.7 \pm 1.0$ |
| Halfcheetah m-r | $\underline{\mathbf{58.5}} \pm 0.6$ | $58.1 \pm 0.7$ | $57.4 \pm 0.7$ |
| Halfcheetah m-e | $\underline{100.3} \pm 0.2$ | $100.4 \pm 0.2$ | $\mathbf{100.5} \pm 0.1$ |

**Noise Schedule**    As mentioned in Appendix E, we used a variance-preserving (VP) noise schedule as the default setting, following most diffusion policies (Wang et al., 2023; He et al., 2023; Fang et al., 2024). Table 23 presents the results of TDP with VP and cosine scheduling. As shown, using a cosine schedule leads to a slight performance drop, but the effect is not substantial.

Table 23: The normalized scores of TDP with varying noise schedule. The best score is in bold and the main reported score is underlined.

| Noise Schedule | VP | Cosine |
|---|---|---|
| Halfcheetah m | **69.7** $\pm$ 0.9 | 63.6 $\pm$ 0.6 |
| Halfcheetah m-r | **58.5** $\pm$ 0.6 | 52.5 $\pm$ 0.7 |
| Halfcheetah m-e | **100.3** $\pm$ 0.2 | 97.3 $\pm$ 0.2 |

**Critic Structure**    As mentioned in Appendix E, we used a temporal difference (TD) learning for critic updates. Table 24 presents the results of TDP with TD and IQL critic update. As shown, using TD learning for the critic is essential for achieving strong performance.

Table 24: The normalized scores of original TDP and TDP with IQL critic. The best score is in bold and the main reported score is underlined.

| Critic | TD | IQL |
|---|---|---|
| Halfcheetah m | **69.7** $\pm$ 0.9 | 59.3 $\pm$ 0.7 |
| Halfcheetah m-r | **58.5** $\pm$ 0.6 | 39.1 $\pm$ 3.3 |
| Halfcheetah m-e | **100.3** $\pm$ 0.2 | 84.1 $\pm$ 3.7 |

## F.2 TRAINING AND EVALUATION TIME

Since TDP employs both a Q ensemble and a guidance loss, concerns may arise regarding excessive training and evaluation time. However, as shown in Table 25, both remain within twice that of DQL. This efficiency primarily stems from the use of an EDP-style Q loss (Kang et al., 2023), which reduces training time compared to the full-step Q loss in DQL. Additional time-saving techniques, such as delayed policy updates and computing the guidance loss using only half of the samples, also contribute to the improved efficiency. Evaluation time is almost solely affected by the number of Q ensembles.

Table 25: The relative training and evaluation times of various diffusion policies, normalized to DQL (set as 100%), are reported. Since DAC originally uses a different library, making direct comparison difficult, we re-implemented DAC using the CleanDiffuser (Dong et al., 2024) library, which is also used for TDP. Here, DAC* denotes the version with all TDP time-saving techniques applied, while DAC** refers to the version without these optimizations.

| Algorithm | TDP | DAC* | DAC** | DQL | DiffCPS |
|---|---|---|---|---|---|
| Training time (relative) | 172.1% | 141.1% | 175.2% | 100.0% | 72.6% |
| Evaluation time (relative) | 179.9% | 172.0% | 170.4% | 100.0% | 102.3% |

## F.3 FIRST ALTERNATIVE OF $\epsilon_\eta^*$ ESTIMATION

To test the first alternative, we pretrained a diffusion BC model and used five $a^0$ samples to estimate $f(a^k, k, s)$ according to (14). We trained in the Halfcheetah and Hopper environments. As shown in Table 26, this approach yielded neither meaningful performance improvements nor greater stability. Moreover, it required an additional diffusion BC model, leading to about 20% longer training time (excluding the pretraining phase) and increased training complexity. For these reasons, we decided not to adopt this alternative.

Table 26: The normalized scores of TDP with varying number of noise-free action samples. The best score is in bold.

| # samples | 1 (TDP) | 5 (Alternative 1) |
|---|---|---|
| Halfcheetah m | $69.7 \pm 0.9$ | $\mathbf{70.0} \pm 0.7$ |
| Halfcheetah m-r | $\mathbf{58.5} \pm 0.6$ | $57.4 \pm 0.4$ |
| Halfcheetah m-e | $\mathbf{100.3} \pm 0.2$ | $100.2 \pm 0.3$ |
| Hopper m | $\mathbf{103.0} \pm 0.1$ | $102.7 \pm 0.6$ |
| Hopper m-r | $\mathbf{104.4} \pm 0.2$ | $103.9 \pm 0.7$ |
| Hopper m-e | $\mathbf{111.4} \pm 0.6$ | $111.1 \pm 1.2$ |

# G  LEARNING CURVES

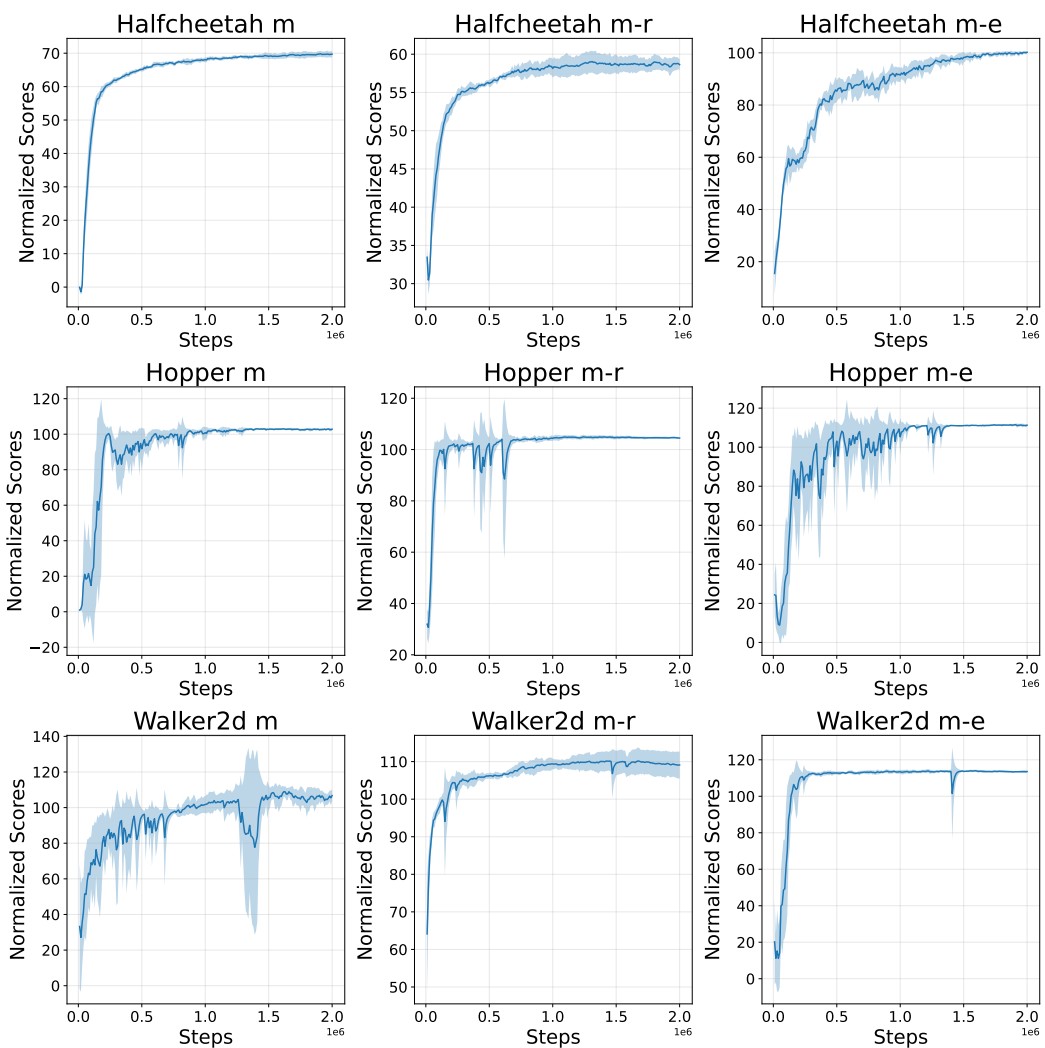

Figure 3: Learning curves of MuJoCo locomotion tasks.

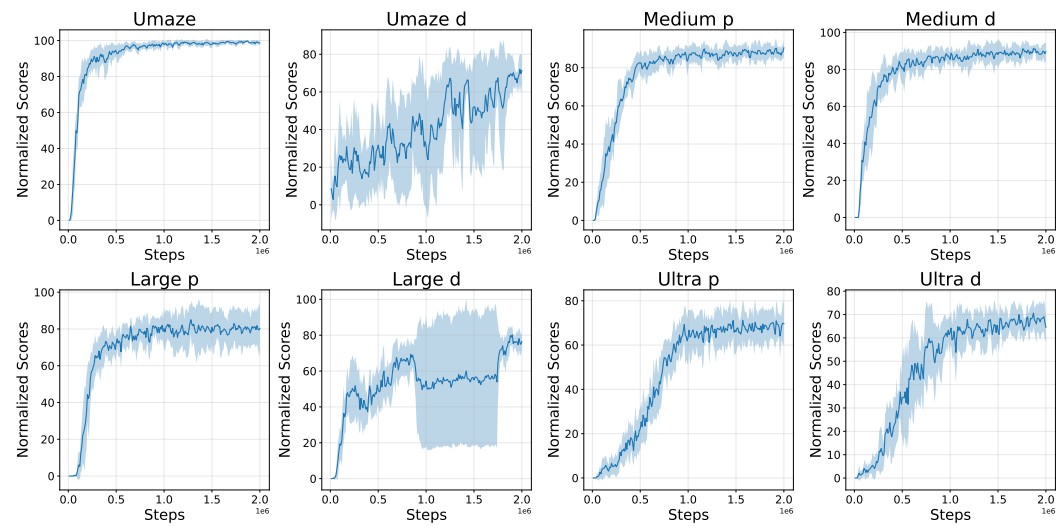

Figure 4: Learning curves of Antmaze tasks.

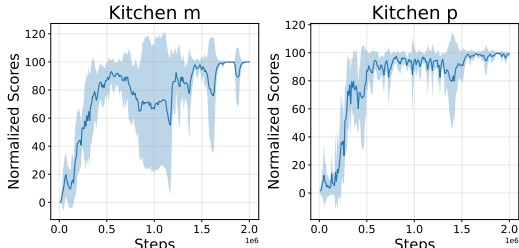

Figure 5: Learning curves of Kitchen tasks.

## H    DECLARATION OF LLM USAGE

The authors used GPT-4o and 5 to correct grammatical errors, to polish texts to improve readability, and to embellish writings at the paragraph level. The outputs were verified and additionally refined by the authors without the assistance of LLM, to ensure coherence across paragraphs and sections and to use preferred terminologies. The authors did not use LLM for generating the core ideas, proofs, and experiments.

