# OpenReview forum: "Constrained Diffusion Policy Optimization for Offline Reinforcement Learning"
_ICLR.cc/2026/Conference — ICLR 2026 Conference Withdrawn Submission_

### Official Review · Reviewer_K1Gw · 2025-10-30

**Soundness:** 3
**Presentation:** 3
**Contribution:** 2
**Rating:** 4
**Confidence:** 4

**Summary:**

This paper introduces TDP, a diffusion policy-based offline RL algorithm that combines the so-called implicit and explicit policy constraint methods in offline RL. During policy improvement, TDP uses 1) a Q value term that encourages the diffusion policy to generate high Q-valued actions; and 2) a regularization term that aligns the score of the diffusion policy with the score of behavior policies and the gradient of Q-values. Several relaxations are made in the practical implementation for tractable and efficient computation. Finally, the proposed TDP is evaluated using tasks from D4RL, and it outperforms baseline methods by a large margin.

**Strengths:**

Empirical performance is strong. Ablation studies with respect to certain design choices, such as the noise-free estimation, are comprehensive.

**Weaknesses:**

The novelty is limited. For the algorithm design, it combines the techniques from DAC and the Q-maximizing objective from Diffusion-QL and EDP for policy improvement, which is straightforward. The theoretical analysis appears to follow [1] and AWAC, with an extension to diffusion divergence.

Many relaxations to the theory are made in practice, e.g., the noise-free estimation, and the approximate inference used in EDP, making the overall algorithms biased from their theoretical characterization.

Besides, TPO seems to require extensive hyperparameter tuning efforts (Tables 8,9,10).

Finally, several related papers should be discussed in the related work section, including DiffCPS [3] and BDPO [4], which all employ the policy constraint framework for diffusion policy optimization.

[1] Offline RL with no ood actions: In-sample learning via implicit value regularization. ICLR 2023.

[2] DiffCPS: Diffusion Model-based Constrained Policy Search for Offline Reinforcement Learning.

[3] Behavior-Regularized Diffusion Policy Optimization for Offline Reinforcement Learning. ICML 2025.

**Questions:**

In lines 156-159, the authors mentioned that implicit methods such as DAC suffer from a restricted function class and limited suppression of suboptimal regions. However, since we are working with policies that are parameterized as diffusion models, the function class is flexible enough and does not suffer from the mode-averaging issue of simple policy classes like Gaussians.  Besides, this claim seems to be related to the bandit experiments in Figure 1, where the implicit method generates points in suboptimal clusters. However, this actually makes sense to me since the generation results of Implicit methods conform to the Boltzmann distribution of the clusters. Adding an extra Q-maximization loss seems to encourage mode-seeking behaviors; however, this can also be achieved by adjusting the temperature for the implicit methods. In short, I would appreciate it if the authors could provide more explanation about this claim and, therefore, better motivate TDP.

---

> ### Author Response · Authors · 2025-11-24
>
> We thank the reviewer for the considerate and constructive comments. Below, we present our detailed replies to all points raised, together with answers to the specific questions.
>
>
>
> **[R1] Novelty of TDP (Weakness 1)**
>
>
>
> Our generalized CPO and CDPO formulations unify the two major policy constraint paradigms, explicit and implicit methods, within a single framework. This perspective reveals that leveraging both axes, **choosing an appropriate anchor policy** and **incorporating a Q function-based improvement term**, naturally leads to what we call the **two-fold improvement**, from which TDP is derived. Although the final algorithm may appear at first glance to be a simple linear combination of components, the underlying design is nontrivial.
>
> Regarding the loss that encourages proximity to $\pi^\ast_\eta$, it may superficially resemble DAC, but the gap between **noise-free** and **noisy** estimation is significantly larger than it appears. Our 2D bandit experiments, additional ablations, and theoretical derivations all support that noise-free estimation is more coherent and better aligned with the intended objective. Even at the formulation level, before estimation, our approach is **minimizer-preserving**, while DAC introduces an additional relaxation that does not guarantee this property. Moreover, we additionally analyzed noise-free estimation and found more meanings of it. See revised Section 3.2 for details.
>
> For these reasons, we believe that both the conceptual development and the resulting algorithm represent meaningful and nontrivial contributions.
>
>
>
> **[R2] Regarding gaps between theory and practice due to relaxations (Weakness 2)**
>
>
>
> We understand the concern, but these relaxations do not compromise the core theoretical viewpoint of the paper, and most are both unavoidable and preferable to available alternatives. As shown in the 2D bandit experiment, noise-free estimation provides almost zero OOD action generation in contrast to the noisy estimation, and for main experiments, noise-free estimation outperforms the unbiased alternative that relies on multiple $a^0$ samples. Moreover, as explained in the revised pdf, the bias suppresses the strong guidance to the high Q actions, thereby reduces OOD action generation.
>
> Regarding the EDP-like Q loss, previous work and our experiments both indicate that its practical bias is small and mitigated by reduced variance. Theoretically, since the action is deterministic once the noise is fixed and there is no backpropagation-through-time (BPTT), which is required for a full-denoising Q loss like in DQL, the EDP-like Q loss significantly reduces variance. As a result, the remaining bias does not disrupt the essential properties of the Q objective.
>
>
>
> **[R3] Hyperparameter tuning of TDP (Weakness 3)**
>
>
>
> The hyperparameters naturally differ across environments, and we made efforts to keep these differences minimal, as detailed in Appendix E.2. This variation is expected: just as in other offline RL methods, environment-specific tuning is required to account for dataset coverage, optimality levels, dynamics, and reward structure. These differences arise from the intrinsic characteristics of the tasks rather than from implementation artifacts. Importantly, the tuning process remains intuitive and generalizable, as it follows from the structural properties of each dataset and environment. Moreover, the performance improvements of TDP cannot be attributed merely to careful tuning, but instead indicate the effectiveness of the underlying algorithm.

---

> > ### Author Response · Authors · 2025-11-24
> >
> > **[R4] Additional related works (Weakness 4)**
> >
> >
> >
> > DiffCPS has already discussed in **Diffusion Models for Offline RL** in related works section. We added BDPO as a related work in **Diffusion Models for Offline RL**.
> >
> >
> >
> > **[R5] Addressing problems of implicit methods (Question 1)**
> >
> >
> >
> > First of all, the term **restricted function class** for implicit methods means that the policy is restricted to the form $\pi^\ast_\eta(a | s) = \frac{1}{Z(s)} \pi_\beta(a | s)\exp(\tfrac{1}{\eta} Q_{\phi}(s,a))$,
> > not the function class of the practical policy model. Although $\pi^\ast_\eta$ is a strong candidate that improves expected Q while remaining in-distribution, it is reasonable to expect that an even better policy lies near it, rather than another $\pi^\ast_\eta$ with a larger temperature that simply moves toward a greedy policy.
> >
> >
> > TDP aims to uncover such an improved policy by optimizing
> > $L(\theta) := L^\ast_\eta(\theta) - \zeta_t L_{Q_\phi}(\theta)$,
> > where $L^\ast_\eta(\theta)$ keeps the policy close to $\pi^\ast_\eta$ and $L_{Q_\phi}(\theta)$ encourages higher expected Q. This balance is the central idea of TDP. In contrast, increasing the temperature only pushes the policy farther from the dataset, which tends to induce undesirable OOD actions.
> >
> > It is also important to note that, in our formulation, the Q loss influences the policy more effectively than when it is combined with a naive BC loss that simply matches $\pi_\beta$. Since $\pi^\ast_\eta$ already has high expected Q, the updates from this reference are small but accurate, enabling gradual improvement while staying in-distribution. In addition, capturing these refined improvements, as well as accurately approximating $\pi^\ast_\eta$ itself, requires an expressive policy class so that such behaviors can be represented in practice.

---

### Official Review · Reviewer_5MGM · 2025-11-01

**Soundness:** 2
**Presentation:** 1
**Contribution:** 1
**Rating:** 2
**Confidence:** 4

**Summary:**

The paper introduces CDPO, a framework that unifies explicit and implicit policy-constraint methods for offline RL under diffusion policies, and proposes TDP (Two-fold Improved Diffusion Policy): (i) initialize from the reverse-KL-constrained closed-form “anchor” policy πη∗ (implicit step), then (ii) further improve it via a constrained optimization with a nonzero Q-loss (explicit step). Theoretically, the authors argue that the diffusion loss induces a pathwise forward KL, enabling strong duality and a policy enhancement theorem (monotone improvement for the first step and expected-Q improvement thereafter); they also state an approximate in-distribution property. Practically, they introduce a noise-free estimate to implement the anchor loss without pushing the critic into the noise domain, and add stabilizers (LCB Q-ensembles,
delayed actor updates, and an EDP one-step approximation). Experiments on D4RL locomotion, AntMaze, Kitchen, plus a 2D bandit visualization and a flow-policy variant (TFP), suggest TDP outperforms strong baselines and generalizes to flow policies.

**Strengths:**

The paper provides a well-motivated unified view (CDPO) that generalizes both explicit and implicit policy constraint methods.

**Weaknesses:**

1. The notions are complicated, and the writing, especially the methodological expression, is rather chaotic.
2. In my opinion, this paper has limited innovation. Compared to DAC, it merely uses Tweedie's formula to estimate the action a in order to mitigate the extrapolation error in calculating the gradient of the Q function in x_t.
3. This article's explanation of the two-fold policy for TDP is unclear, which greatly diminishes its importance.
4. The experimental baselines are limited, and in recent years, many new methods for diffusion policy and flow policy have emerged in the offline RL field.
5. Do all the baselines in Tables 1–3 use the same critic structure? While the paper emphasizes the importance of the LCB objective in stable Q-value estimation, it lacks persuasiveness. I believe more ablation studies should be conducted, such as experiments combining IQL and TDP, or LCB and IDQL.
6. f in (9) requires further explanation and analysis.

**Questions:**

See weakness.

---

> ### Author Response · Authors · 2025-11-24
>
> We thank the reviewer for the  valuable and thoughtful reviews. Below, we provide detailed responses to each concern raised.
>
>
>
> **[R1] Regarding the notions and writing (Weakness 1)**
>
>
>
> To address the reviewer's  concerns, we conducted several revisions to improve readability and strengthening the coherence of the presentation. These include:
>
> i) reorganizing and rewriting the derivation of the TDP loss in Section 3.2, and
> ii) revising the notation in Section 2 to match the formulation in (7) (formerly (6)) and the related theoretical analysis.
>
>
> Because the revisions to Section 3.2 constitute the core of our response, we provided a detailed summary of those modifications in the common comment above.
>
> Notation alignment also provides a sense of coherence across sections, preventing confusion caused by inconsistent terminology.
>
> We would welcome any concrete critiques on writing style or particular explanations that could be further refined.
>
>
>
>
> **[R2] Innovation of this paper (Weakness 2)**
>
>
>
> The main contribution of this paper is  the two-fold improvement, an idea of addressing the core limitations of the two major types of policy-constraint methods: explicit and implicit methods. Explicit methods often struggle to optimize the policy when the behavior policy differs significantly from the target policy, and their reliance on the Q loss guidance introduces ambiguity in the update direction, increasing the risk of generating OOD actions. Implicit methods, on the other hand, are limited by the fixed functional form of $\pi^\ast_\eta$. Although $\pi^\ast_\eta$ is a strong candidate, it is highly plausible that an even better policy exists in its neighborhood. This distinction underscores the central innovation of our work compared to DAC, which is an implicit method and therefore only imitates $\pi^\ast_\eta$.
>
> For the noise-free estimation, we acknowledge that the previous explanation was somewhat simplistic. In the revised version, we clarified its deeper significance, showing that noise-free estimation corresponds to a first-order Taylor expansion and that its bias effectively suppresses excessive guidance of the reverse process toward high Q regions.
> Moreover, majority of existing approaches involving $\exp(\frac{1}{\eta}Q_\phi(s,a))$, such as DAC, QSM [1], and FlowQ [2], do not use our formulation and instead rely on noisy estimation.
>
> [1] Psenka et al., Learning a Diffusion Model Policy from Rewards via Q-Score Matching, ICML (2024).
>
>
> [2] Alles et al., FlowQ: Energy-Guided Flow Policies for Offline Reinforcement Learning, [arXiv preprint arXiv:2505.14139](https://arxiv.org/abs/2505.14139) (2025).
>
>
>
> **[R3] Explanation of two-fold (Weakness 3)**
>
>
>
> We would appreciate additional clarification regarding why our original explanation of the two-fold improvement seemed unclear, as this feedback will help us refine the presentation. Our current understanding is that the issue may come from not explicitly stating that each component of the two-fold approach has its own limitations when used on its own. We have revised the explanation accordingly to make these limitations clear and to emphasize that the effectiveness of our method comes from combining both components rather than relying on only one.

---

> > ### Author Response · Authors · 2025-11-24
> >
> > **[R4] Regarding contemporary diffusion and flow policies as baselines (Weakness 4)**
> >
> >
> >
> > We already included several diffusion and flow policy methods as baselines, such as DQL, QGPO, DiffCPS, and DAC for diffusion policies, and FQL, FBRAC, IFQL, and FlowQ for flow policies. Although adding more baselines in the main paper is challenging while keeping comparisons manageable across a wide range of offline RL algorithms, we are happy to include additional diffusion or flow policy methods in the Appendix if you can provide specific examples. We are now considering Diffusion DICE, QIPO (both diffusion and flow), and BDPO as candidates.
> >
> >
> >
> >
> > **[R5] About critic structures of TDP and baselines, and additional ablations (Weakness 5)**
> >
> >
> >
> > Although the LCB objective plays an important role, the more fundamental point is that TDP relies on a temporal-difference (TD) objective. The critic objectives used by the baselines can be grouped into four categories:
> > i) in-sample objectives such as those in IQL,
> > ii) TD objectives,
> > iii) variants of TD, and
> > iv) no critic objective.
> >
> > Under this classification, IQL, QGPO, GC-IQL, GC-POR, and HIQL fall into the first category. TDP, DQL, DiffCPS, and DAC fall into the second. ReBRAC, SVR, SAC-N, and EDAC fall into the third. GCPC and DT belong to the last category.
> >
> > In-sample objectives make the critic independent of the policy. This reduces the risk of severe failure caused by immature policies and is especially helpful for navigation tasks such as Antmaze. However, this independence also limits the critic’s ability to benefit from a strong policy, which is more restrictive in locomotion tasks and still visible in Antmaze when compared with TDP.
> >
> > TD objectives, in contrast, couple the critic with the policy. This enables the critic to improve beyond the in-sample policy and, importantly, strengthens its stitching ability, which is crucial for combining trajectories with overlapping segments to learn long-horizon behavior.
> >
> > Variants of TD are designed to reduce overestimation by adding explicit penalties to the Q target. This direction lies outside our current scope because it introduces additional hyperparameters, and empirically, TDP does not suffer from such overestimation issues.
> >
> > Since GCPC and DT are transformer-based methods, they do not use critics and instead rely on return-to-go or goal conditioning.
> >
> > In short, the TD objective aligns best with TDP, as it allows the Q function to improve iteratively and guide the policy toward better performance through repeated cycles of critic and policy updates. While using an IQL-like critic target with our two-fold improvement is possible in principle, we expect it to degrade performance.
> >
> > Among your suggestions for additional ablations, we conducted the ablation with the IQL critic variant of TDP and added the results to Appendix F.1. Ablations using an IQL critic combined with LCB will not be conducted, since the IQL critic target does not incorporate LCB and therefore does not meaningfully reflect the intended objective.

---

### Official Review · Reviewer_XBd4 · 2025-11-01

**Soundness:** 4
**Presentation:** 3
**Contribution:** 4
**Rating:** 8
**Confidence:** 3

**Summary:**

This manuscript proposes the  constrained diffusion policy optimization (CDPO) framework and a Two-fold improved Diffusion Policy for offline RL. The CDPO combines the explicit and implicit policy constraint methods, while TDP even harnesses the full potential of CDPO. The authors show the properties of TDP theoretically and provide practical method design to implement TDP. Experiments on D4RL benchmark demonstrates the superiority of TDP.

**Strengths:**

* The idea of CDPO is interesting. It unifies explicit and implicit  policy constraints, and serves as foundation for combining diffusion-based and constraint policy optimization methods.

* The method of TDP is quite novel, starting from $\pi_{\eta}^*$ is appealing while avoiding exploring in OOD areas. In addition, it is appreciated that the authors provide rigorous prove for policy improvement.

* The generlization ability of TDP looks pretty good. It can be easily generlized to other variants such as TFP, which also serves as a competitive alternative.

**Weaknesses:**

1. Some related references are missing, and it is suggested to consider the related work in the manuscript.

* https://arxiv.org/abs/2303.15810

* https://arxiv.org/abs/2202.06239

* https://arxiv.org/abs/1911.11361

* https://arxiv.org/abs/2301.12130

* https://arxiv.org/abs/2405.16173


2. Though the manuscript is novel, the reviewer does not prefer the way of statements in the abstract. The readers may be mislead that the CDPO is a conventional method, while the authors only propose TDP. The authors are suggested to refine the statements in abstract.

3. From the experiments, it seems the TDP delivers the best performance in only 3 out of 9 tasks in D4RL benchmark from Table 1 (Similar in Table 2). The effectiveness of TDP is not fully shown.  It is suggested the authors provide further explanations on this issue.

4. For Actor and Critic Update, if the method selects the Q-ensembles, will it lead to extra cost such as inference multiple times or extra training cost? Have the authors evaluated this issue?

5. Another question is about the starting policy $\pi_{\eta}^*$,  how could we easily get this initial policy in practice? Or could we try any other initial policies for better exploration?

**Questions:**

See the weakness above

---

> ### Author Response · Authors · 2025-11-24
>
> We appreciate your quality comments. Below, we offer detailed responses to each of the weaknesses you brought up.
>
>
>
>
>
> **[R1] Additional related works (Weakness 1)**
>
>
>
>
> Among the five suggested references, the first four fall under policy-constraint methods, and we have incorporated them accordingly. The final reference is QVPO, a well-known online diffusion policy algorithm. Since online diffusion policy methods are also relevant to our scope, we added a corresponding paragraph and included QVPO as well.
>
>
>
>
>
> **[R2] Potentially misleading abstract (Weakness 2)**
>
>
>
>
> To resolve this issue, we revised a word in the abstract to make it explicit that CDPO is first shown in our work, thereby reducing any impression that it refers to a pre-existing framework.
>
>
>
>
> **[R3] Empirical effectiveness of TDP (Weakness 3)**
>
>
>
>
>
> First of all, many of the baselines, particularly DAC, DiffCPS, Q ensemble methods, and goal-conditioned algorithms, already achieve strong performance. Because of this, surpassing them across most datasets is difficult, and improving on average is meaningful. More importantly, since these baselines mostly perform well on one of the  different categories of tasks, locomotion and Antmaze, the consistent gains achieved by TDP in both settings indicate a substantial improvement. In addition, the near-perfect results on the Kitchen tasks (Table 3), as noted in the paper, represent an important milestone.
>
>
>
> **[R4] Additional cost for Q ensembles (Weakness 4)**
>
>
>
>
> We already discussed this issue in Appendix F.2. While the Q ensembles in TDP do add computational overhead, the training time has become less than double during training because of the computation-saving techniques we employ. For evaluation, the dominant cost is action sampling rather than Q value computation, so the evaluation time also increases by less than double.
>
>
>
>
> **[R5] The way to obtain $\pi^\ast_\eta$ (Weakness 5)**
>
>
>
> In practice, we use temporal difference (TD) learning with randomly initialized networks, combined with the desired policy losses, i.e., to increase the Q objective while keeping the policy close to $\pi^\ast_\eta$. We also explored an alternative approach that begins directly from $\pi^\ast_\eta$, omitting the Q loss at the start and introducing it only after convergence, but this did not provide performance gains. Our current procedure therefore maintains the intended optimization objective while remaining simpler and equally effective.

---

### Official Review · Reviewer_tumZ · 2025-11-01

**Soundness:** 2
**Presentation:** 2
**Contribution:** 2
**Rating:** 2
**Confidence:** 4

**Summary:**

This paper introduces Constrained Diffusion Policy Optimization (CDPO), a unified framework that generalizes both explicit and implicit policy constraint methods in offline reinforcement learning (RL). Building on this framework, it proposes the Two-Fold Improved Diffusion Policy (TDP), a practical algorithm that leverages the advantages of both approaches. Specifically, TDP initializes from the implicit reverse KL–constrained policy $\pi^*_\eta$, which surpasses the behavior policy while remaining in-distribution, and subsequently refines it through Q-guided optimization under the generalized CDPO loss. Theoretically, the paper establishes a Policy Enhancement Theorem that guarantees monotonic policy improvement under mild assumptions, and further provides a bounded in-distribution property for diffusion-based policies.

**Strengths:**

CDPO unifies explicit and implicit policy constraint methods within a single formulation by generalizing constrained policy optimization (CPO) to incorporate a flexible anchor policy $\pi_0 \neq \pi_\beta$. The two-fold improvement strategy—anchoring to $\pi^*_\eta$ and refining via the Q-loss—is both conceptually elegant and empirically effective. Moreover, the paper establishes formal policy enhancement and in-distribution theorems for diffusion policies, providing theoretical grounding for results that were previously heuristic. Empirically, TDP attains state-of-the-art performance across the D4RL benchmark.

**Weaknesses:**

1.	The formulation of the surrogate loss involves several theoretically unsound steps. For instance, the “noise-free estimation” replaces a theoretically exact expression with an ad-hoc gradient approximation ($f \approx \frac{1}{\eta} Q_\phi(s,a_0)$), introducing bias that is not rigorously analyzed.
2.	Although the “noise-free estimation” technique is proposed, the subsequent issues of exploding or vanishing gradients in the guidance process are neither resolved nor thoroughly discussed.
3.	The theoretical results are limited to expected monotonic improvement and approximate dominance, without providing formal convergence or finite-sample guarantees. Moreover, the theoretical gains assume that $\pi^*_\eta$ is a “strong” initialization, yet its practical computation still relies on accurate Q-value estimation—introducing potential circularity. This limitation may restrict the applicability of the proposed algorithm, especially when dealing with medium- or low-quality datasets.
4.	While the paper claims to present a unified framework generalizing existing explicit and implicit policy constraint methods, several important diffusion-policy baselines [1,2,3,4] are omitted.
5.	Despite employing computational optimizations (e.g., delayed updates, half-batch gradients), the diffusion-based actor remains computationally heavy, and the discussion on training efficiency is relatively brief.

[1] Diffusion Policies as an Expressive Policy Class for Offline Reinforcement Learning (Wang et al., 2022)

[2] Efficient Diffusion Policies for Offline Reinforcement Learning (Kang et al., 2023)

[3] IDQL: Implicit Q Learning as an Actor Critic Method with Diffusion Policies (Hansen-Estruch et al., 2023)

[4] Preferred Action Optimized Diffusion Policies for Offline Reinforcement Learning (Zhang et al., 2024)

**Questions:**

1.	Does the iterative refinement of TDP converge to a fixed point or optimal constrained policy? Are there stability guarantees under function approximation?
2.	How large is the bias introduced by approximating $f(a_k,k,s) \approx \frac{1}{\eta} Q(s,a_0)$? Does this bias accumulate during denoising steps?
3.	Diffusion models are expressive but unstable. How sensitive is TDP to diffusion step count $K$ or noise schedule choices?

---

> ### Author Response · Authors · 2025-11-24
>
> Thank you for such careful and insightful feedback. We address each of the concerns in detail and respond to the questions.
>
>
>
> **[R1] Concerning the bias of noise-free estimation (Weakness 1 and Question 2)**
>
>
>
> During the revision of Section 3.2, we obtained a closed-form expression for the bias introduced by noise-free estimation, $-D_{\text{KL}}(\pi^\ast_\eta(a^0|a^k,s)\parallel \pi_\beta(a^0|a^k,s))$ (see Section 3.2 of the revised PDF for details). When the true guidance strongly shifts noisy actions toward high Q regions, the gap between the two posteriors increases, and this negative KL term acts to suppress such movement. This yields a conservative correction that naturally helps prevent OOD action generation.
>
>
>
> **[R2] Addressing exploding or vanishing gradients in the guidance process (Weakness 2)**
>
>
>
> Our noise-free estimation helps prevent exploding or vanishing gradients by avoiding the log-sum-exp operation required for the unbiased computation of $f$. The remaining gradient behavior comes from the form $\hat{\epsilon} \approx \epsilon - \frac{1}{\eta}\sqrt{\frac{1-\bar{\alpha}^k}{\bar{\alpha}^k}} \nabla_{a^0} Q_\phi(s,a^0)$,
> which can be obtained by using $a^k = \sqrt{\bar{\alpha}^k}a^0 + \sqrt{1-\bar{\alpha}^k}\epsilon$ and thus $\frac{\partial a^0}{\partial a^k} = \frac{1}{\sqrt{\bar{\alpha}^k}}$.
>
> To stabilize $\nabla_{a^0} Q_\phi(s,a^0)$, we apply layer normalization to the Q network. In all of our experiments, this consistently prevented exploding or vanishing gradients, as layer normalization is known to stabilize first-order gradient and indirectly controls gradient scale.
>
> The coefficient $\frac{1}{\eta}$ is bounded in a reasonable scale by the definition of $\pi^\ast_\eta$, and the remaining factor $\sqrt{\frac{1-\bar{\alpha}^k}{\bar{\alpha}^k}}$ depends only on the noise schedule. For our main setting (VP schedule, 5 timesteps, $\beta_{\min}=0.1$, $\beta_{\max}=10$), this factor takes values
> $[0.494, 1.14, 2.31, 4.98, 12.5]$ for $k = 1,\cdots,5$,
> all of which have reasonable magnitudes. Together, these elements prevent gradient explosion or collapse in practice.
>
>
>
> **[R3] Convergence guarantee of the algorithm (Weakness 3 and Question 1)**
>
>
>
> Let us consider CDPO in its primal form together with the Bellman backup update
> $Q(s,a) \leftarrow r + \gamma \mathbb{E}_{\pi}[Q(s',a')]$,
> which is realized through temporal-difference (TD) learning. Since the Bellman backup operator is a $\gamma$-contraction mapping, and TDP can be viewed as repeatedly applying a Bellman backup followed by a projection onto the constraint set induced by the primal CDPO formulation, the resulting composite operator admits a fixed point by the fixed-point theorem, ensuring convergence.
>
> However, this fixed point is with respect to the learned Q function rather than the **true** return. Although this provides a much stronger guarantee than typical arguments used in offline RL, it does not imply convergence to the exact constrained optimal solution, which is an inherent limitation of offline RL itself.

---

> > ### Author Response · Authors · 2025-11-24
> >
> > **[R4] Dealing with medium or low-quality datasets (Weakness 3)**
> >
> >
> >
> > In contrary to this concern, TDP performs best on medium-quality datasets, where high-quality trajectories can be obtained through stitching. Both the two-fold improvement and temporal-difference (TD) learning contribute strongly to this. Tasks that require long-horizon stitching, such as Antmaze ultra, are therefore particularly suitable for TDP, supported by empirical results.
> >
> >
> >
> > **[R5] Classification of existing diffusion policy algorithms within the CDPO framework (Weakness 4)**
> >
> >
> >
> > DQL [1] is already discussed in the paper and corresponds to the case where $\zeta_t \equiv \zeta$ and $\pi_0 = \pi_\beta$.
> > For EDP [2], its objective combines the diffusion loss with a Q loss, similar in spirit to DQL, although the form of the Q loss differs. Under the CDPO framework, it therefore falls into the same category as DQL.
> >
> > For IDQL [3], the training procedure itself becomes standard BC, corresponding to $\zeta_t \equiv 0$ with $\pi_0 = \pi_\beta$.
> >
> > PAO-DP [4] appears to adopt a DPO-like approach to approximate $\pi^\ast_\eta$ by using a linear combination of the diffusion loss and a Q loss–equivalent term, which aligns with explicit methods based on a reverse-KL constraint, an unusual case of CDPO. However, such methods require evaluating log probabilities, which are generally infeasible for diffusion models, and the paper does not describe how this evaluation is carried out. For this reason, we defer incorporating PAO-DP into the main categorization. Instead, we included PAO-DP in the related works section.
> >
> >
> > We appreciate your helpful suggestions regarding additional diffusion policy algorithms and have updated the corresponding section of the main paper accordingly.
> >
> >
> >
> > **[R6] Training and evaluation time analysis (Weakness 5)**
> >
> >
> >
> > We have already included relative training and evaluation times in Appendix F.2. For reference, the absolute training time is approximately 20 hours for three parallel runs on an RTX 3090 GPU.
> >
> >
> >
> >
> > **[R7] Sensitivity of TDP with respect to the number of diffusion timesteps or noise schedules (Question 3)**
> >
> >
> >
> > Regarding the number of diffusion timesteps, we already reported the corresponding ablation results in Appendix F.1 on page 48 (previously page 46).
> > For the cosine noise schedule, we added new results in Appendix F.1 on page 49.
> >
> > As shown in these tables, the performance does not degrade significantly when varying either the diffusion timesteps or the noise schedule.

---

### Author Response · Authors · 2025-11-24
**Common Response**

We thank all reviewers for their insightful and constructive feedback.
Because many of the revisions address concerns shared across multiple reviews, we provide this common comment to clearly summarize the updates made to the paper.


## **List of Revised Sections**

- **Abstract**
- **Section 2 (Preliminaries)**
- **Section 3.1 (Constrained Diffusion Policy Optimization (CDPO))**
- **Section 3.2 (Two-Fold Improved Diffusion Policy (TDP))**
- **Section 5 (Related Work)**
- **Appendix D.7 (Proofs for Section 3.2)**
- **Appendix F.1 (Ablations)**
---

Among these revisions, the updates to **Section 3.2** are the most critical, as they directly address the core conceptual and technical issues raised by the reviewers.
We also made substantial improvements to **Section 5** and **Appendix F.1** to enhance clarity and the level of completion.

Below, we summarize the most important modifications in these sections.  (While all listed sections were revised, we highlight only the major changes here.)



## **Summaries of Revised Sections**

### **1. Section 3.2**

As mentioned above, the changes applied in this section is the most important throughout all the changes. Because this section was the source of many concerns, we rewrote it to provide a clearer and more intuitive structure.
Key changes include:

- We added the CDPO dual form of TDP and clarified that a conversion into the practical $L(\theta)$ formulation is required for implementation.

- The surrogate form of $L_\eta^*(\theta)$ is introduced earlier, so its purpose is clear before deriving the final TDP loss.

- The target structure of the noise update
  $\hat{\epsilon} = \epsilon + \epsilon_\eta^*(a^k,k,s) - \epsilon_\beta(a^k,k,s)$
  is stated immediately after presenting the surrogate objective, and the derivation now proceeds forward rather than backward.

- The score function identities for $\epsilon_\eta^*$ and $\epsilon_\beta$ are presented in direct connection to the suggested form of $\hat{\epsilon}$, providing a clear motivation for introducing the function $f(a^k,k,s)$.

- The function $f(a^k,k,s)$ is defined *after* establishing the minimizer form, making its role explicit. Its final expression is fixed to match (14) (previously (12)).

- Noise-free estimation is now motivated through its interpretation as a **first-order Taylor approximation** of $f(a^k,k,s)$, yielding $\tilde{f}(a^k,k,s)$, and the approximation $\hat{\epsilon} \approx \epsilon - \frac{1}{\eta}\sqrt{1-\bar{\alpha}^k}\nabla_{a^k} Q_\phi(s,a^0)$  is explained via the MMSE estimator property.

- Using the closed-form expression of $\tilde{f}$, we fully derived the bias of noise-free estimation as
  $-D_{\mathrm{KL}}(\pi_\beta(a^0|a^k,s)\parallel\pi_\eta^*(a^0|a^k,s))$,
  and we qualitatively analyzed how this bias introduces a conservative correction that suppresses overly aggressive movement toward high Q, low-density actions and thus mitigates OOD action generation.

These revisions make **Section 3.2** significantly more transparent, self-contained, and logically structured, while preserving all theoretical correctness.


### **2. Section 5**

We incorporated additional related works suggested by the reviewers into both **Policy Constraint Methods for Offline RL** and **Diffusion Models for Offline RL**, improving the completeness of the discussion.  We also added a new subsection on **Diffusion Models for Online RL** to broaden the context and clarify how diffusion models are used beyond offline RL.



### **3. Appendix G.1**

We added ablation studies examining the effect of varying the noise schedule, as well as an ablation of a TDP variant using an IQL critic.



---

We hope this summary helps all reviewers quickly identify the relevant updates, and we welcome further suggestions for improvement.

---

### Note · Authors · 2026-01-09

I have read and agree with the venue's withdrawal policy on behalf of myself and my co-authors.